# On the Certified Robustness for Ensemble Models and Beyond

**Zhuolin Yang**[1]* **Linyi Li**[1]* **Xiaojun Xu**[1] **Bhavya Kailkhura**[2] **Tao Xie**[3] **Bo Li**[1]
[1]University of Illinois Urbana-Champaign
[2]Lawrence Livermore National Laboratory [3]Peking University
`{zhuolin5,linyi2,xiaojun3,lbo}@illinois.edu`
`kailkhura1@llnl.gov taoxie@pku.edu.cn`
* Equal contribution

## Abstract

Recent studies show that deep neural networks (DNN) are vulnerable to adversarial examples, which aim to mislead DNNs by adding perturbations with small magnitude. To defend against such attacks, both empirical and theoretical defense approaches have been extensively studied for a *single ML model*. In this work, we aim to analyze and provide the certified robustness for *ensemble ML models*, together with the sufficient and necessary conditions of robustness for different ensemble protocols. Although ensemble models are shown more robust than a single model empirically; surprisingly, we find that in terms of the certified robustness the standard ensemble models only achieve marginal improvement compared to a single model. Thus, to explore the conditions that guarantee to provide certifiably robust ensemble ML models, we first prove that *diversified gradient* and *large confidence margin* are sufficient and necessary conditions for certifiably robust ensemble models under the model-smoothness assumption. We then provide the bounded model-smoothness analysis based on the proposed *Ensemble-before-Smoothing* strategy. We also prove that an ensemble model can *always* achieve higher certified robustness than a single base model under mild conditions. Inspired by the theoretical findings, we propose the lightweight Diversity Regularized Training (DRT) to train certifiably robust ensemble ML models. Extensive experiments show that our DRT enhanced ensembles can consistently achieve higher certified robustness than existing single and ensemble ML models, demonstrating the state-of-the-art certified $L_2$-robustness on MNIST, CIFAR-10, and ImageNet datasets.

## 1 Introduction

Deep neural networks (DNN) have been widely applied in various applications, such as image classification (Krizhevsky, 2012; He et al., 2016), face recognition (Sun et al., 2014), and natural language processing (Vaswani et al., 2017; Devlin et al., 2019). However, it is well-known that DNNs are vulnerable to adversarial examples (Szegedy et al., 2013; Carlini & Wagner, 2017; Xiao et al., 2018a;b; Bhattad et al., 2020; Bulusu et al., 2020), and it has raised great concerns especially when DNNs are deployed in safety-critical applications such as autonomous driving and facial recognition.

To defend against such attacks, several empirical defenses have been proposed (Papernot et al., 2016b; Madry et al., 2018); however, many of them have been attacked again by strong adaptive attackers (Athalye et al., 2018; Tramer et al., 2020). To end such repeated game between the attackers and defenders, *certified* defenses (Wong & Kolter, 2018; Cohen et al., 2019) have been proposed to provide the robustness guarantees for given ML models, so that no additional attack can break the model under certain adversarial constraints. For instance, randomized smoothing has been proposed as an effective defense providing certified robustness (Lecuyer et al., 2019; Cohen et al., 2019; Yang et al., 2020a). Among different certified robustness approaches (Weng et al., 2018; Xu et al., 2020; Li et al., 2020a; Zhang et al., 2022), randomized smoothing provides a model-independent way to smooth a given ML model and achieves state-of-the-art certified robustness on large-scale datasets such as ImageNet.

Currently, all the existing certified defense approaches focus on the robustness of a single ML model. Given the observations that ensemble ML models are able to bring additional benefits in standard

learning (Opitz & Maclin, 1999; Rokach, 2010), in this work we aim to ask: *Can an ensemble ML model provide additional benefits in terms of the certified robustness compared with a single model? If so, what are the sufficient and necessary conditions to guarantee such certified robustness gain?*

Empirically, we first find that *standard* ensemble models only achieve marginally higher certified robustness by directly appling randomized smoothing: with $L_2$ perturbation radius 1.5, a single model achieves certified accuracy as 21.9%, while the average aggregation based ensemble of three models achieves certified accuracy as 24.2% on CIFAR-10 (Table 2). Given such observations, next we aim to answer: *How to improve the certified robustness of ensemble ML models? What types of conditions are required to improve the certified robustness for ML ensembles?*

In particular, from the theoretical perspective, we analyze the standard Weighted Ensemble (WE) and Max-Margin Ensemble (MME) protocols, and prove the sufficient and necessary conditions for the certifiably robust ensemble models under model-smoothness assumption. Specifically, we prove that: (1) an ensemble ML model is more certifiably robust than each single base model; (2) *diversified gradients* and *large confidence margins* of base models are the sufficient and necessary conditions for the certifiably robust ML ensembles. We show that these two key factors would lead to higher certified robustness for ML ensembles. We further propose *Ensemble-before-Smoothing* as the model smoothing strategy and prove the bounded model-smoothness with such strategy, which realizes our model-smoothness assumption.

Inspired by our theoretical analysis, we propose **D**iversity-**R**egularized **T**raining (DRT), a lightweight regularization-based ensemble training approach. DRT is composed of two simple yet effective and general regularizers to promote the diversified gradients and large confidence margins respectively. DRT can be easily combined with existing ML approaches for training smoothed models, such as Gaussian augmentation (Cohen et al., 2019) and adversarial smoothed

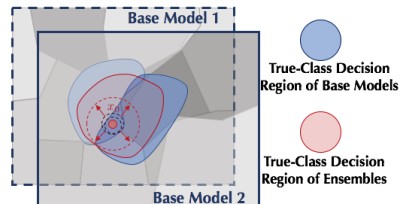

Figure 1: Illustration of a robust ensemble.

training (Salman et al., 2019), with negligible training time overhead while achieves *significantly* higher certified robustness than state-of-the-art approaches consistently.

We conduct extensive experiments on a wide range of datasets including MNIST, CIFAR-10, and ImageNet. The experimental results show that DRT can achieve significantly higher certified robustness compared to baselines with similar training cost as training a single model. Furthermore, as DRT is flexible to integrate any base models, by using the pretrained robust single ML models as base models, DRT achieves the highest certified robustness so far to our best knowledge. For instance, on CIFAR-10 under $L_2$ radius 1.5, the DRT-trained ensemble with three base models improves the certified accuracy from SOTA 24.2% to 30.3%; and under $L_2$ radius 2.0, DRT improves the certified accuracy from SOTA 16.0% to 20.3%.

**Technical Contributions.** In this paper, we conduct the *first* study for the sufficient and necessary conditions of certifiably robust ML ensembles and propose an efficient training algorithm DRT to achieve the state-of-the-art certified robustness. We make contributions on both theoretical and empirical fronts.

- We provide the *necessary* and *sufficient* conditions for robust ensemble ML models including Weighted Ensemble (WE) and Max-Margin Ensemble (MME) under the model-smoothness assumption. In particular, we prove that the *diversified gradients* and *large confidence margins* of base models are the sufficient and necessary conditions of certifiably robust ensembles. We also prove the bounded model-smoothness via proposed *Ensemble-before-Smoothing* strategy, which realizes our model-smoothness assumption.
- To analyze different ensembles, we prove that when the adversarial transferability among base models is low, WE is more robust than MME. We also prove that the ML ensemble is more robust than a single base model under the model-smoothness assumption.
- Based on the theoretical analysis of the sufficient and necessary conditions, we propose DRT, a lightweight regularization-based training approach that can be easily combined with different training approaches and ensemble protocols with small training cost overhead.
- We conduct extensive experiments to evaluate the effectiveness of DRT on various datasets, and we show that to the best of your knowledge, DRT can achieve the *highest* certified robustness, outperforming all existing baselines.

**Related work.** DNNs are known vulnerable to adversarial examples (Szegedy et al., 2013). To defend against such attacks, several empirical defenses have been proposed (Papernot et al., 2016b; Madry et al., 2018). For ensemble models, existing work mainly focuses on empirical robustness (Pang et al., 2019; Li et al., 2020b; Cheng et al., 2021) where the robustness is measured by accuracy under existing attacks and no certified robustness guarantee could be provided or enhanced; or certify the robustness for a standard weighted ensemble (Zhang et al., 2019; Liu et al., 2020) using either LP-based (Zhang et al., 2018) verification or randomized smoothing without considering the model diversity (Liu et al., 2020) to boost their certified robustness. In this paper, we aim to prove that the diversified gradient and large confidence margin are the sufficient and necessary conditions for certifiably robust ensemble ML models. Moreover, to our best knowledge, we propose the *first* training approach to boost the *certified* robustness of ensemble ML models.

Randomized smoothing (Lecuyer et al., 2019; Cohen et al., 2019) has been proposed to provide certified robustness for a single ML model. It achieved the state-of-the-art certified robustness on large-scale dataset such as ImageNet and CIFAR-10 under $L_2$ norm. Several approaches have been proposed to further improve it by: (1) choosing different smoothing distributions for different $L_p$ norms (Dvijotham et al., 2019; Zhang et al., 2020; Yang et al., 2020a), and (2) training more robust smoothed classifiers, using data augmentation (Cohen et al., 2019), unlabeled data (Carmon et al., 2019), adversarial training (Salman et al., 2019), regularization (Li et al., 2019; Zhai et al., 2019), and denoising (Salman et al., 2020). In this paper, we compare and propose a suitable smoothing strategy to improve the certified robustness of ML ensembles.

## 2 CHARACTERIZING ML ENSEMBLE ROBUSTNESS

In this section, we prove the sufficient and necessary robustness conditions for both general and smoothed ML ensemble models. Based on these robustness conditions, we discuss the key factors for improving the certified robustness of an ensemble, compare the robustness of ensemble models with single models, and outline several findings based on additional theoretical analysis.

### 2.1 PRELIMINARIES

**Notations.** Throughout the paper, we consider the classification task with $C$ classes. We first define the classification scoring function $f : \mathbb{R}^d \to \mathbf{\Delta}^C$, which maps the input to a *confidence vector*, and $f(\boldsymbol{x})_i$ represents the confidence for the $i$th class. We mainly focus on the confidence after normalization, i.e., $f(\boldsymbol{x}) \in \mathbf{\Delta}^C = \{\boldsymbol{p} \in \mathbb{R}_{\geq 0}^C : \|\boldsymbol{p}\|_1 = 1\}$ in the probability simplex. To characterize the *confidence margin* between two classes, we define $f^{y_1/y_2}(\boldsymbol{x}) := f(\boldsymbol{x})_{y_1} - f(\boldsymbol{x})_{y_2}$. The corresponding *prediction* $F : \mathbb{R}^d \to [C]$ is defined by $F(\boldsymbol{x}) := \arg\max_{i \in [C]} f(\boldsymbol{x})_i$. We are also interested in the *runner-up prediction* $F^{(2)}(\boldsymbol{x}) := \arg\max_{i \in [C]:i \neq F(\boldsymbol{x})} f(\boldsymbol{x})_i$.

$r$**-Robustness.** For brevity, we consider the model's certified robustness, against the $L_2$-bounded perturbations as defined below. Our analysis can be generalizable for $L_1$ and $L_\infty$ perturbations, leveraging existing work (Li et al., 2019; Yang et al., 2020a; Levine & Feizi, 2021).

**Definition 1** ($r$-Robustness). For a prediction function $F : \mathbb{R}^d \to [C]$ and input $\boldsymbol{x}_0$, if all instance $\boldsymbol{x} \in \{\boldsymbol{x}_0 + \boldsymbol{\delta} : \|\boldsymbol{\delta}\|_2 < r\}$ satisfies $F(\boldsymbol{x}) = F(\boldsymbol{x}_0)$, we say model $F$ is $r$-*robust* (at point $\boldsymbol{x}_0$).

**Ensemble Protocols.** An ensemble model contains $N$ *base models* $\{F_i\}_{i=1}^N$, where $F_i(\boldsymbol{x})$ and $F_i^{(2)}(\boldsymbol{x})$ are their top and runner-up predictions for given input $\boldsymbol{x}$ respectively. The ensemble prediction is denoted by $\mathcal{M} : \mathbb{R}^d \to [C]$, which is computed based on outputs of base models following certain ensemble protocols. In this paper, we consider both Weighted Ensemble (WE) and Maximum Margin Ensemble (MME).

**Definition 2** (Weighted Ensemble (WE)). Given $N$ base models $\{F_i\}_{i=1}^N$, and the weight vector $\{w_i\}_{i=1}^N \in \mathbb{R}_+^N$, the weighted ensemble $\mathcal{M}_{\mathrm{WE}}: \mathbb{R}^d \to [C]$ is defined by

$$\mathcal{M}_{\mathrm{WE}}(\boldsymbol{x}_0) := \arg\max_{i \in [C]} \sum_{j=1}^N w_j f_j(\boldsymbol{x}_0)_i. \tag{1}$$

**Definition 3** (Max-Margin Ensemble (MME)). Given $N$ base models $\{F_i\}_{i=1}^N$, for input $\boldsymbol{x}_0$, the max-margin ensemble model $\mathcal{M}_{\mathrm{MME}} : \mathbb{R}^d \to [C]$ is defined by

$$\mathcal{M}_{\mathrm{MME}}(\boldsymbol{x}_0) := F_c(\boldsymbol{x}_0) \quad \text{where} \quad c = \arg\max_{i \in [N]} \left( f_i(\boldsymbol{x}_0)_{F_i(\boldsymbol{x}_0)} - f_i(\boldsymbol{x}_0)_{F_i^{(2)}(\boldsymbol{x}_0)} \right). \tag{2}$$

The commonly-used WE (Zhang et al., 2019; Liu et al., 2020) sums up the weighted confidence of base models $\{F_i\}_{i=1}^N$ with weight vector $\{w_i\}_{i=1}^N$, and predicts the class with the highest weighted confidence. The standard average ensemble can be viewed as a special case of WE (where all $w_i$'s are equal). MME chooses the base model with the largest confidence margin between the top and the runner-up classes, which is a direct extension from max-margin training (Huang et al., 2008).

**Randomized Smoothing.**    Randomized smoothing (Lecuyer et al., 2019; Cohen et al., 2019) provides certified robustness by constructing a smoothed model from a given model. Formally, let $\varepsilon \sim \mathcal{N}(0, \sigma^2 \boldsymbol{I}_d)$ be a Gaussian random variable, for any given model $F : \mathbb{R}^d \to [C]$ (can be an ensemble), we define *smoothed confidence* function $g_F^\varepsilon : \mathbb{R}^d \to \boldsymbol{\Delta}^C$ such that

$$g_F^\varepsilon(\boldsymbol{x})_j := \mathbb{E}_{\varepsilon \sim \mathcal{N}(0, \sigma^2 \boldsymbol{I}_d)} \mathbb{I}[F(\boldsymbol{x} + \varepsilon) = j] = \Pr_{\varepsilon \sim \mathcal{N}(0, \sigma^2 \boldsymbol{I}_d)}(F(\boldsymbol{x} + \varepsilon) = j). \tag{3}$$

Intuitively, $g_F^\varepsilon(\boldsymbol{x})_j$ is the probability of base model $F$'s prediction on the $j$th class given Gaussian smoothed input. The smoothed classifier $G_F^\varepsilon : \mathbb{R}^d \to [C]$ outputs the class with highest smoothed confidence: $G_F^\varepsilon(\boldsymbol{x}) := \arg\max_{j \in [C]} g_F^\varepsilon(\boldsymbol{x})_j$. Let $c_A$ be the predicted class for input $\boldsymbol{x}_0$, i.e., $c_A := G_F^\varepsilon(\boldsymbol{x}_0)$. Cohen et al. show that $G_F^\varepsilon$ is $(\sigma \Phi^{-1}(g_F^\varepsilon(\boldsymbol{x}_0)_{c_A}))$-robust at input $\boldsymbol{x}_0$, i.e., the certified radius is $\sigma \Phi^{-1}(g_F^\varepsilon(\boldsymbol{x}_0)_{c_A})$ where $\Phi^{-1}$ is the inverse cumulative distribution function of standard normal distribution. In practice, we will leverage the smoothing strategy together with Monte-Carlo sampling to certify ensemble robustness. More details can be found in Appendix A.

## 2.2    Robustness Conditions for General Ensemble Models

We will first provide sufficient and necessary conditions for robust ensembles under the model-smoothness assumption.

**Definition 4** ($\beta$-Smoothness). A differentiable function $f : \mathbb{R}^d \mapsto \mathbb{R}^C$ is $\beta$-smooth, if for any $\boldsymbol{x}_1, \boldsymbol{x}_2 \in \mathbb{R}^d$ and any output dimension $j \in [C]$, $\frac{\|\nabla_{\boldsymbol{x}_1} f(\boldsymbol{x}_1)_j - \nabla_{\boldsymbol{x}_2} f(\boldsymbol{x}_2)_j\|_2}{\|\boldsymbol{x}_1 - \boldsymbol{x}_2\|_2} \leq \beta$.

The definition of $\beta$-smoothness is inherited from optimization theory literature, and it is equivalent to the curvature bound in certified robustness literature (Singla & Feizi, 2020). $\beta$ quantifies the non-linearity of function $f$, where higher $\beta$ indicates more rigid functions/models and smaller $\beta$ indicates smoother ones. When $\beta = 0$ the function/model is linear.

For Weighted Ensemble (WE), we have the following robustness conditions.

**Theorem 1** (Gradient and Confidence Margin Conditions for WE Robustness). *Given input $\boldsymbol{x}_0 \in \mathbb{R}^d$ with ground-truth label $y_0 \in [C]$, and $\mathcal{M}_{\mathrm{WE}}$ as a WE defined over base models $\{F_i\}_{i=1}^N$ with weights $\{w_i\}_{i=1}^N$. $\mathcal{M}_{\mathrm{WE}}(\boldsymbol{x}_0) = y_0$. All base models $F_i$'s are $\beta$-smooth.*

- *(Sufficient Condition) The $\mathcal{M}_{\mathrm{WE}}$ is $r$-robust at point $\boldsymbol{x}_0$ if for any $y_i \neq y_0$,*

$$\Big\| \sum_{j=1}^N w_j \nabla_{\boldsymbol{x}} f_j^{y_0/y_i}(\boldsymbol{x}_0) \Big\|_2 \leq \frac{1}{r} \sum_{j=1}^N w_j f_j^{y_0/y_i}(\boldsymbol{x}_0) - \beta r \sum_{j=1}^N w_j, \tag{4}$$

- *(Necessary Condition) If $\mathcal{M}_{\mathrm{WE}}$ is $r$-robust at point $\boldsymbol{x}_0$, for any $y_i \neq y_0$,*

$$\Big\| \sum_{j=1}^N w_j \nabla_{\boldsymbol{x}} f_j^{y_0/y_i}(\boldsymbol{x}_0) \Big\|_2 \leq \frac{1}{r} \sum_{j=1}^N w_j f_j^{y_0/y_i}(\boldsymbol{x}_0) + \beta r \sum_{j=1}^N w_j. \tag{5}$$

The proof follows from Taylor expansion at $\boldsymbol{x}_0$ and we leave the detailed proof in Appendix B.2. When it comes to Max-Margin Ensemble (MME), the derivation of robust conditions is more involved. In Theorem 3 (Appendix B.1.1) we derive the robustness conditions for MME composed of two base models. The robustness conditions have highly similar forms as those for WE in Theorem 1. Thus, for brevity, we focus on discussing Theorem 1 for WE hereinafter and similar conclusions can be drawn for MME (details are in Appendix B.1.1).

To analyze Theorem 1, we define *Ensemble Robustness Indicator* (ERI) as such:

$$I_{y_i} := \Big\| \sum_{j=1}^N w_j \nabla_{\boldsymbol{x}} f_j^{y_0/y_i}(\boldsymbol{x}_0) \Big\|_2 \Big/ \|\boldsymbol{w}\|_1 - \frac{1}{r \|\boldsymbol{w}\|_1} \sum_{j=1}^N w_j f_j^{y_0/y_i}(\boldsymbol{x}_0). \tag{6}$$

ERI appears in both sufficient (Equation (4)) and necessary (Equation (5)) conditions. In both conditions, *smaller* ERI means *more* certifiably robust ensemble. Note that we can analyze the

robustness under different attack radius $r$ by directly varying $r$ in Equations (4) and (5). When $r$ becomes larger, the gap between the RHS of two inequalities ($2\beta r \sum_{j=1}^{N} w_j$) also becomes larger, and thus it becomes harder to determine robustness via Theorem 1. This is because the first-order condition implied by Theorem 1 becomes coarse when $r$ is large. However, due to bounded $\beta$ as we will show, the training approach motivated by the theorem still empirically works well under large $r$.

**Diversified Gradients.** The core of first term in ERI is the magnitude of the vector sum of gradients: $\|\sum_{j=1}^{N} w_j \nabla_{\boldsymbol{x}} f_j^{y_0/y_i}(\boldsymbol{x}_0)\|_2$. According to the law of cosines: $\|\boldsymbol{a} + \boldsymbol{b}\|_2 = \sqrt{\|\boldsymbol{a}\|_2^2 + \|\boldsymbol{b}\|_2^2 + 2\|\boldsymbol{a}\|_2\|\boldsymbol{b}\|_2 \cos\langle\boldsymbol{a}, \boldsymbol{b}\rangle}$, to reduce this term, we could either reduce the base models' gradient magnitude or diversify their gradients (in terms of cosine similarity). Since simply reducing base models' gradient magnitude would hurt model expressivity (Huster et al., 2018), during regularization the main functionality of this term would be promoting diversified gradients.

**Large Confidence Margins.** The core of second term in ERI is the confidence margin: $\sum_{j=1}^{N} w_j f_j^{y_0/y_i}(\boldsymbol{x}_0)$. Due to the negative sign of second term in ERI, we need to increase this term, i.e., we need to increase confidence margins to achieve higher ensemble robustness.

In summary, the diversified gradients and large confidence margins are the sufficient and necessary conditions for high certified robustness of ensembles. In Section 3, we will directly regularize these two key factors to promote certified robustness of ensembles.

**Impact of Model-Smoothness Bound $\beta$.** From Theorem 1, we observe that: (1) if $\min_{y_i \neq y_0} I_{y_i} \leq -\beta r$, $\mathcal{M}_{\mathrm{WE}}$ is guaranteed to be $r$-robust (sufficient condition); and (2) if $\min_{y_i \neq y_0} I_{y_i} > \beta r$, $\mathcal{M}_{\mathrm{WE}}$ cannot be $r$-robust (necessary condition). However, if $\min_{y_i \neq y_0} I_{y_i} \in (-\beta r, \beta r]$, we only know $\mathcal{M}_{\mathrm{WE}}$ is possibly $r$-robust. As a result, the model-smoothness bound $\beta$ decides the correlation strength between $\min_{y_i \neq y_0} I_{y_i}$ and the robustness of $\mathcal{M}_{\mathrm{WE}}$: if $\beta$ becomes larger, $\min_{y_i \neq y_0} I_{y_i}$ is more likely to fall in $(-\beta r, \beta r]$, inducing an undetermined robustness status from Theorem 1, vice versa. Specifically, when $\beta = 0$, i.e., all base models are linear, the gap is closed and we can always certify the robustness of $\mathcal{M}_{\mathrm{WE}}$ via comparing $\min_{y_i \neq y_0}$ with $0$. Similar observations can be drawn for MME. Therefore, to strengthen the correlation between $I_{y_i}$ and ensemble robustness, we would need model-smoothness bound $\beta$ to be small.

## 2.3 ROBUSTNESS CONDITIONS FOR SMOOTHED ENSEMBLE MODELS

Typically neural networks are nonsmooth or admit only coarse smoothness bounds (Sinha et al., 2018), i.e., $\beta$ is large. Therefore, applying Theorem 1 for normal nonsmooth models would lead to near-zero certified radius. Therefore, we propose soft smoothing to enforce the smoothness of base models. However, with the soft smoothed base models, directly applying Theorem 1 to certify robustness is still practically challenging, since the LHS of Equations (4) and (5) involves gradient of the soft smoothed confidence. A precise computation of such gradient requires high-confidence estimation of high-dimensional vectors via sampling, which requires linear number of samples with respect to input dimension (Mohapatra et al., 2020; Salman et al., 2019) and is thus too expensive in practice. To solve this issue, we then propose *Ensemble-before-Smoothing* as the practical smoothing protocol, which serves as an approximation of soft smoothing, so as to leverage the randomized smoothing based techniques for certification.

**Soft Smoothing.** To impose base models' smoothness, we now introduce soft smoothing (Kumar et al., 2020), which applies randomized smoothing over *the confidence scores*. Given base model's confidence function $f : \mathbb{R}^d \to \boldsymbol{\Delta}^C$ (see Section 2.1), we define *soft smoothed confidence* by $\bar{g}_f^\varepsilon : \boldsymbol{x} \mapsto \mathbb{E}_\varepsilon f(\boldsymbol{x} + \varepsilon)$. Note that *soft* smoothed confidence is different from smoothed confidence $g_F^\varepsilon$ defined in Equation (3). We consider soft smoothing instead of classical smoothing in Equation (3) since soft smoothing reveals differentiable and thus practically regularizable training objectives. The following theorem shows the smoothness bound for $\bar{g}_f^\varepsilon$.

**Theorem 2** (Model-Smoothness Upper Bound for $\bar{g}_f^\varepsilon$). *Let $\varepsilon \sim \mathcal{N}(0, \sigma^2 \boldsymbol{I}_d)$ be a Gaussian random variable, then the soft smoothed confidence function $\bar{g}_f^\varepsilon$ is $(2/\sigma^2)$-smooth.*

We defer the proof to Appendix B.4. The proof views the Gaussian smoothing as the Weierstrass transform (Weierstrass, 1885) of a function from $\mathbb{R}^d$ to $[0, 1]^C$, leverages the symmetry property, and bounds the absolute value of diagonal elements of the Hessian matrix. Note that a Lipschitz constant $\sqrt{2/(\pi\sigma^2)}$ is derived for smoothed confidence in previous work (Salman et al., 2019, Lemma 1),

which characterizes only the first-order smoothness property; while our bound in addition shows the second-order smoothness property. In Appendix B.4, we further show that our smoothness bound in Theorem 2 is tight up to a constant factor.

Now, we apply WE and MME protocols with these soft smoothness confidence $\{\bar{g}_i^\varepsilon(\boldsymbol{x}_0)\}_{i=1}^N$ as base models' confidence scores, and obtain soft ensemble $\bar{G}_{\mathcal{M}_{\mathrm{WE}}}^\varepsilon$ and $\bar{G}_{\mathcal{M}_{\mathrm{MME}}}^\varepsilon$ respectively. Since each $\bar{g}_i^\varepsilon$ is $(2/\sigma^2)$-smooth, take WE as an example, we can study the ensemble robustness with Theorem 1. We state the full statement in Corollary 2 (and in Corollary 3 for MME) in Appendix B.1.3. From the corollary, we observe that the corresponding ERI for the soft smoothed WE can be written as

$$\bar{I}_{y_i} := \left\| \mathbb{E}_\varepsilon \nabla_{\boldsymbol{x}} \sum_{j=1}^N w_j f_j^{y_0/y_i}(\boldsymbol{x}_0 + \varepsilon) \right\|_2 \Big/ \|\boldsymbol{w}\|_1 - \frac{1}{r\|\boldsymbol{w}\|_1} \mathbb{E}_\varepsilon \sum_{j=1}^N w_j f_j^{y_0/y_i}(\boldsymbol{x}_0 + \varepsilon). \tag{7}$$

We have following observations: (1) unlike for standard models with unbounded $\beta$, for the smoothed ensemble models, this ERI (Equation (7)) would have *guaranteed* correlation with the model robustness since $\beta = \Theta(1/\sigma^2)$ is bounded and can be controlled by tuning $\sigma$ for smoothing. (2) we can still control ERI by diversifying gradients and ensuring large confidence margins as discussed in Section 2.2, but need to compute on the noise augmented input $\boldsymbol{x}_0 + \varepsilon$ instead of original input $\boldsymbol{x}_0$.

**Towards Practical Certification.** As outlined at the beginning of this subsection, even with smoothed base models, certifying robustness using Theorem 1 is practically difficult. Therefore, we introduce *Ensemble-before-Smoothing* strategy as below to construct $G_{\mathcal{M}_{\mathrm{WE}}}^\varepsilon$ and $G_{\mathcal{M}_{\mathrm{MME}}}^\varepsilon$ as approximations of soft ensemble $\bar{G}_{\mathcal{M}_{\mathrm{WE}}}^\varepsilon$ and $\bar{G}_{\mathcal{M}_{\mathrm{MME}}}^\varepsilon$ respectively.

**Definition 5** (*Ensemble-before-Smoothing* (**EBS**)). Let $\mathcal{M}$ be an ensemble model over base models $\{F_i\}_{i=1}^N$ and $\varepsilon$ be a random variable. The EBS strategy construct smoothed classifier $G_{\mathcal{M}}^\varepsilon : \mathbb{R}^d \to [C]$ that picks the class with highest smoothed confidence of $\mathcal{M}$: $G_{\mathcal{M}}^\varepsilon(\boldsymbol{x}) := \arg\max_{j \in [C]} g_{\mathcal{M}}^\varepsilon(\boldsymbol{x})_j$.

Here $\mathcal{M}$ could be either $\mathcal{M}_{\mathrm{WE}}$ or $\mathcal{M}_{\mathrm{MME}}$. EBS aims to approximate the soft smoothed ensemble. Formally, use WE as an example, we let $f_{\mathcal{M}_{\mathrm{WE}}} := \frac{\sum_{j=1}^N w_j f_j}{\|\boldsymbol{w}\|_1}$ to be WE ensemble's confidence, then

$$g_{\mathcal{M}_{\mathrm{WE}}}^\varepsilon(\boldsymbol{x})_i = \mathbb{E}_\varepsilon \mathbb{I}[\mathcal{M}_{\mathrm{WE}}(\boldsymbol{x} + \varepsilon) = i] \approx \mathbb{E}_\varepsilon f_{\mathcal{M}_{\mathrm{WE}}}(\boldsymbol{x} + \varepsilon)_i = \frac{\sum_{j=1}^N w_j (\bar{g}_{f_j}^\varepsilon)_i}{\sum_{j=1}^N w_j} = \bar{g}_{\mathcal{M}_{\mathrm{WE}}}^\varepsilon(\boldsymbol{x})_i \tag{8}$$

where LHS is the smoothed confidence of EBS ensemble and RHS is the soft smoothed ensemble's confidence. Such approximation is also adopted in existing work (Salman et al., 2019; Zhai et al., 2019; Kumar et al., 2020) and shown effective and useful. Therefore, our robustness analysis of soft smoothed ensemble still applies with EBS and we can control ERI in Equation (7) to improve the certified robustness of EBS ensemble. For EBS ensemble, we can leverage randomized smoothing based techniques to compute the robustness certification (see Proposition C.1 in Appendix C).

### 2.4 ADDITIONAL PROPERTIES OF ML ENSEMBLES

**Comparison between Ensemble and Single-Model Robustness.** In Appendix B.1, we show Corollary 1, a corollary of Theorem 1, which indicates that when the base models are smooth enough, both WE and MME ensemble models are more certifiably robust than the base models. This aligns with our empirical observations (see Table 1 and Table 2), though *without* advanced training approaches such as DRT, the improvement of robustness brought by ensemble itself is marginal. In Appendix B.1, we also show larger number of base models $N$ can lead to better certified robustness.

**Comparison between WE and MME Robustness.** Since in actual computing, the certified radius of a smoothed model is directly correlated with the probability of correct prediction under smoothed input (see Equation (11) in Appendix A), we study the robustness of both WE and MME along with single models from the statistical robustness perspective in Appendix D. From the study, we have the following theoretical observations verified by numerical experiments: (1) MME is more robust when the adversarial transferability is high; while WE is more robust when the adversarial transferability is low. (2) If we further assume that $f_i(x_0 + \varepsilon)_{y_0}$ follows marginally uniform distribution, when the number of base models $N$ is sufficiently large, MME is always more certifiably robust. Appendix D.5 entails the numerical evaluations that verify our theoretical conclusions.

### 3 DIVERSITY-REGULARIZED TRAINING

Inspired by the above key factors in the sufficient and necessary conditions for the certifiably robust ensembles, we propose the **D**iversity-**R**egularized **T**raining (DRT). In particular, let $\boldsymbol{x}_0$ be a training sample, DRT contains the following two regularization terms in the objective function to minimize:

- Gradient Diversity Loss (GD Loss): $\mathcal{L}_{\text{GD}}(\boldsymbol{x}_0)_{ij} = \left\| \nabla_{\boldsymbol{x}} f_i^{y_0/y_i^{(2)}}(\boldsymbol{x}_0) + \nabla_{\boldsymbol{x}} f_j^{y_0/y_j^{(2)}}(\boldsymbol{x}_0) \right\|_2.$  (9)

- Confidence Margin Loss (CM Loss): $\mathcal{L}_{\text{CM}}(\boldsymbol{x}_0)_{ij} = f_i^{y_i^{(2)}/y_0}(\boldsymbol{x}_0) + f_j^{y_j^{(2)}/y_0}(\boldsymbol{x}_0).$  (10)

In Equations (9) and (10), $y_0$ is the ground-truth label of $\boldsymbol{x}_0$, and $y_i^{(2)}$ (or $y_j^{(2)}$) is the runner-up class of base model $F_i$ (or $F_j$). Intuitively, for each model pair $(F_i, F_j)$ where $i, j \in [N]$ and $i \neq j$, the GD loss promotes the *diversity of gradients* between the base model $F_i$ and $F_j$. Note that the gradient computed here is actually the gradient difference between different labels. As our theorem reveals, it is the gradient difference between different labels instead of pure gradient itself that matters, which improves existing understanding of gradient diversity (Pang et al., 2019; Demontis et al., 2019). Specifically, the GD loss encourages both large gradient diversity and small base models' gradient magnitude in a naturally balanced way, and encodes the interplay between gradient magnitude and direction diversity. In contrast, solely regularizing the base models' gradient would hurt the model's benign accuracy, and solely regularizing gradient diversity is hard to realize due to the boundedness of cosine similarity. The CM loss encourages the *large margin* between the true and runner-up classes for base models. Both regularization terms are directly motivated by theoretical analysis in Section 2.

For each input $\boldsymbol{x}_0$ with ground truth $y_0$, we use $\boldsymbol{x}_0 + \varepsilon$ with $\varepsilon \sim \mathcal{N}(0, \sigma^2 \boldsymbol{I}_d)$ as training input for each base model (i.e., Gaussian augmentation). We call two base models $(F_i, F_j)$ a *valid* model pair at $(\boldsymbol{x}_0, y_0)$ if both $F_i(\boldsymbol{x}_0 + \varepsilon)$ and $F_j(\boldsymbol{x}_0 + \varepsilon)$ equal to $y_0$. For every valid model pair, we apply DRT: GD Loss and CM Loss with $\rho_1$ and $\rho_2$ as the weight hyperparameters as below.

$$\mathcal{L}_{\text{train}} = \sum_{i \in [N]} \mathcal{L}_{\text{std}}(\boldsymbol{x}_0 + \varepsilon, y_0)_i + \rho_1 \sum_{\substack{i,j \in [N], i \neq j \\ (F_i, F_j) \text{ is valid}}} \mathcal{L}_{\text{GD}}(\boldsymbol{x}_0 + \varepsilon)_{ij} + \rho_2 \sum_{\substack{i,j \in [N], i \neq j \\ (F_i, F_j) \text{ is valid}}} \mathcal{L}_{\text{CM}}(\boldsymbol{x}_0 + \varepsilon)_{ij}.$$

The standard training loss $\mathcal{L}_{\text{std}}(\boldsymbol{x}_0 + \varepsilon, y_0)_i$ of each base model $F_i$ is either cross-entropy loss (Cohen et al., 2019), or adversarial training loss (Salman et al., 2019). This standard training loss will help to produce sufficient valid model pairs with high benign accuracy for robustness regularization. Specifically, as discussed in Section 2.3, we compute $\mathcal{L}_{\text{GD}}$ and $\mathcal{L}_{\text{CM}}$ on the noise augmented inputs $(\boldsymbol{x}_0 + \varepsilon)$ instead of $\boldsymbol{x}_0$ to improve the certified robustness for the *smoothed* ensemble.

**Discussion.**    To our best knowledge, this is the *first* training approach that is able to promote the certified robustness of ML ensembles, while existing work either only provide empirical robustness without guarantees (Pang et al., 2019; Kariyappa & Qureshi, 2019; Yang et al., 2020b; 2021), or tries to only optimize the weights of Weighted Ensemble (Zhang et al., 2019; Liu et al., 2020). We should notice that, though concepts similar with the gradient diversity have been explored in empirically robust ensemble training (e.g., ADP (Pang et al., 2019), GAL (Kariyappa & Qureshi, 2019)), directly applying these regularizers cannot train models with high *certified robustness* due to the lack of theoretical guarantees in their design. We indicate this through ablation studies in Appendix G.4. For the design of DRT, we also find that there exist some variations. We analyze them and show that the current design is usually better based on the analysis in Appendix E. Our approach is generalizable for other $L_p$-bounded perturbations such as $L_1$ and $L_\infty$ leveraging existing work (Li et al., 2019; Lecuyer et al., 2019; Yang et al., 2020a; Levine & Feizi, 2021).

## 4    EXPERIMENTAL EVALUATION

To make a thorough comparison with existing certified robustness approaches, we evaluate DRT on different datasets including MNIST (LeCun et al., 2010), CIFAR-10 (Krizhevsky, 2012), and ImageNet (Deng et al., 2009), based on both MME and WE protocols. Overall, we show that the DRT enabled ensemble outperforms all baselines in terms of certified robustness under different settings.

### 4.1    EXPERIMENTAL SETUP

**Baselines.** We consider the following state-of-the-art baselines for certified robustness: **Gaussian smoothing** (Cohen et al., 2019), **SmoothAdv** (Salman et al., 2019), **MACER** (Zhai et al., 2019), **Stability** (Li et al., 2019), and **SWEEN** (Liu et al., 2020). Detail description of these baselines can be found in Appendix F. We follow the configurations of baselines, and compare DRT-based ensemble with Gaussian Smoothing, SmoothAdv, and MACER on all datasets, and in addition compare it with other baselines on MNIST and CIFAR-10 considering the training efficiency. There are other baselines, e.g., (Jeong & Shin, 2020). However, SmoothAdv performs consistently better across different datasets, so we mainly consider SmoothAdv as our strong baseline.

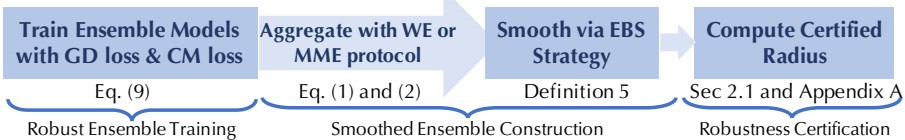

Figure 2: Pipeline for DRT-based ensemble.

Table 1: Certified accuracy under different radii on MNIST dataset. The grey rows present the performance of the proposed DRT approach. The brackets show the base models we use.

| Radius $r$ | 0.00 | 0.25 | 0.50 | 0.75 | 1.00 | 1.25 | 1.50 | 1.75 | 2.00 | 2.25 | 2.50 |
|---|---|---|---|---|---|---|---|---|---|---|---|
| Gaussian (Cohen et al., 2019) | 99.1 | 97.9 | 96.6 | 94.7 | 90.0 | 83.0 | 68.2 | 46.6 | 33.0 | 20.5 | 11.5 |
| SmoothAdv (Salman et al., 2019) | 99.1 | 98.4 | 97.0 | 96.3 | 93.0 | 87.7 | 80.2 | 66.3 | 43.2 | 34.3 | 24.0 |
| MACER (Zhai et al., 2019) | 99.2 | 98.5 | 97.4 | 94.6 | 90.2 | 83.5 | 72.4 | 54.4 | 36.6 | 26.4 | 16.5 |
| Stability (Li et al., 2019) | 99.3 | **98.6** | 97.1 | 93.8 | 90.7 | 83.2 | 69.2 | 46.8 | 33.1 | 20.0 | 11.2 |
| SWEEN (Gaussian) (Liu et al., 2020) | 99.2 | 98.4 | 96.9 | 94.9 | 90.5 | 84.4 | 71.1 | 48.9 | 35.3 | 23.7 | 12.8 |
| SWEEN (SmoothAdv) (Liu et al., 2020) | 99.2 | 98.2 | 97.4 | 96.3 | 93.4 | 88.1 | 81.0 | 67.2 | 44.5 | 34.9 | 25.0 |
| MME (Gaussian) | 99.2 | 98.4 | 96.8 | 94.9 | 90.5 | 84.3 | 69.8 | 48.8 | 34.7 | 23.4 | 12.7 |
| DRT + MME (Gaussian) | **99.5** | **98.6** | 97.5 | 95.5 | 92.6 | 86.8 | 76.5 | 60.2 | 43.9 | 36.0 | 29.1 |
| MME (SmoothAdv) | 99.2 | 98.2 | 97.3 | 96.4 | 93.2 | 88.1 | 80.6 | 67.9 | 44.8 | 35.0 | 25.2 |
| DRT + MME (SmoothAdv) | 99.2 | 98.4 | **97.6** | **96.7** | 93.1 | **88.5** | 83.2 | 68.9 | 48.2 | **40.3** | 34.7 |
| WE (Gaussian) | 99.2 | 98.4 | 96.9 | 94.9 | 90.6 | 84.5 | 70.4 | 49.0 | 35.2 | 23.7 | 12.9 |
| DRT + WE (Gaussian) | **99.5** | **98.6** | 97.4 | 95.6 | 92.6 | 86.7 | 76.7 | 60.2 | 43.9 | 35.8 | 29.0 |
| WE (SmoothAdv) | 99.1 | 98.2 | 97.4 | 96.4 | **93.4** | 88.2 | 81.1 | 67.9 | 44.7 | 35.2 | 24.9 |
| DRT + WE (SmoothAdv) | 99.1 | 98.4 | **97.6** | **96.7** | **93.4** | **88.5** | 83.3 | 69.6 | 48.3 | 40.2 | **34.8** |

**Models.** For base models in our ensemble, we follow the configurations used in baselines: LeNet (Le-Cun et al., 1998), ResNet-110, and ResNet-50 (He et al., 2016) for MNIST, CIFAR-10, and ImageNet datasets respectively. Throughout the experiments, we use $N = 3$ base models to construct the ensemble for demonstration. We expect more base models would yield higher ensemble robustness.

**Training Details.** We follow Section 3 to train the base models. We combine DRT with Gaussian smoothing and SmoothAdv (i.e., instantiating $\mathcal{L}_{std}$ by either cross-entropy loss (Cohen et al., 2019; Yang et al., 2020a) or adversarial training loss (Salman et al., 2019)). We leave training details along with hyperparametes in Appendix F.

**Pipeline.** After the base models are trained with DRT, we aggregate them to form the ensemble $\mathcal{M}$, using either WE or MME protocol (see Definitions 2 and 3). If we use WE, to filter out the effect of different weights, we adopt the average ensemble where all weights are equal. We also studied how optimizing weights can further improve the certified robustness in Appendix G.3. Then, we leverage *Ensemble-before-Smoothing* strategy to form a smoothed ensemble (see Definition 5). Finally, we compute the certified robustness for the smoothed ensemble based on Monte-Carlo sampling with high-confidence (99.9%). The training pipeline is shown in Figure 2.

**Evaluation Metric.** We report the standard *certified accuracy* under different $L_2$ radii $r$'s as our evaluation metric following existing work (Cohen et al., 2019; Yang et al., 2020b; Zhai et al., 2019; Jeong & Shin, 2020). More evaluation details are in Appendix F.

## 4.2 EXPERIMENTAL RESULTS

Here we consider ensemble models consisting of three base models. We show that 1) DRT-based ensembles outperform the SOTA baselines significantly especially under large perturbation radii; 2) smoothed ensembles are always more certifiably robust than each base model (Corollary 1 in Appendix B.1); 3) applying DRT for either MME or WE ensemble protocols achieves similar and consistent improvements on certified robustness.

**Certified Robustness of DRT with Different Ensemble Protocols.** The evaluation results on MNIST, CIFAR-10, ImageNet are shown in Tables 1, 2, 3 respectively. It is clear that though the certified accuracy of a single model can be improved by directly applying either MME or WE ensemble training (proved in Corollary 1), such improvements are usually negligible (usually less than 2%). In contrast, in all tables we find DRT provides significant gains on certified robustness for both MME and WE (up to over 16% as Table 1 shows).

From Tables 1 and 2 on MNIST and CIFAR-10, we find that compared with all baselines, DRT-based ensemble achieves the highest robust accuracy, and the performance gap is more pronounced on large radii (over 8% for $r = 2.50$ on MNIST and 6% for $r = 1.50$ on CIFAR-10). We also demonstrate the scalability of DRT by training on ImageNet, and Table 3 shows that DRT achieves the highest certified robustness under large radii. It is clear that DRT can be easily combined with existing training approaches (e.g. Gaussian smoothing or SmoothAdv), boost their certified robustness, and set the state-of-the-art results to the best of our knowledge.

Table 2: Certified accuracy under different radii on CIFAR-10 dataset. The grey rows present the performance of the proposed DRT approach. The brackets show the base models we use.

| Radius $r$ | 0.00 | 0.25 | 0.50 | 0.75 | 1.00 | 1.25 | 1.50 | 1.75 | 2.00 |
|---|---|---|---|---|---|---|---|---|---|
| Gaussian (Cohen et al., 2019) | 78.9 | 64.4 | 47.4 | 33.7 | 23.1 | 18.3 | 13.6 | 10.5 | 7.3 |
| SmoothAdv (Salman et al., 2019) | 68.9 | 61.0 | 54.4 | 45.7 | 34.8 | 28.5 | 21.9 | 18.2 | 15.7 |
| MACER (Zhai et al., 2019) | 79.5 | 68.8 | 55.6 | 42.3 | 35.0 | 27.5 | 23.4 | 20.4 | 17.5 |
| Stability (Li et al., 2019) | 72.4 | 58.2 | 43.4 | 27.5 | 23.9 | 16.0 | 15.6 | 11.4 | 7.8 |
| SWEEN (Gaussian) (Liu et al., 2020) | 81.2 | 68.7 | 54.4 | 38.1 | 28.3 | 19.6 | 15.2 | 11.5 | 8.6 |
| SWEEN (SmoothAdv) (Liu et al., 2020) | 69.5 | 62.3 | 55.0 | 46.2 | 35.2 | 29.5 | 22.4 | 19.3 | 16.6 |
| MME (Gaussian) | 80.8 | 68.2 | 53.4 | 38.4 | 29.0 | 19.6 | 15.6 | 11.6 | 8.8 |
| DRT + MME (Gaussian) | 81.4 | **70.4** | 57.8 | 43.8 | 34.4 | 29.6 | 24.9 | 20.9 | 16.6 |
| MME (SmoothAdv) | 71.4 | 64.5 | 57.6 | 48.4 | 36.2 | 29.8 | 23.9 | 19.5 | 16.2 |
| DRT + MME (SmoothAdv) | 72.6 | 67.2 | **60.2** | 50.4 | 39.4 | 35.8 | **30.4** | 24.0 | 20.1 |
| WE (Gaussian) | 80.8 | 68.4 | 53.6 | 38.4 | 29.2 | 19.7 | 15.9 | 11.8 | 8.9 |
| DRT + WE (Gaussian) | **81.5** | **70.4** | 57.9 | 44.0 | 34.2 | 29.6 | 24.9 | 20.8 | 16.4 |
| WE (SmoothAdv) | 71.8 | 64.6 | 57.8 | 48.5 | 36.2 | 29.6 | 24.2 | 19.6 | 16.0 |
| DRT + WE (SmoothAdv) | 72.6 | 67.0 | **60.2** | 50.5 | **39.5** | **36.0** | 30.3 | **24.1** | **20.3** |

To evaluate the computational cost of DRT, we analyze the theoretical complexity in Appendix E and compare the efficiency of different methods in practice in Appendices F.1 and F.2. In particular, we show that DRT with Gaussian Smoothing base models even achieves around two times speedup compared with SmoothAdv with comparable or even higher certified robustness, since DRT does not require adversarial training. More discussions about hyper-parameters settings for DRT can be found in Appendix F. In Appendix G.4, we also show that our proposed DRT approach could achieve $6\% \sim 10\%$ higher certified accuracy compared to adapted ADP (Pang et al., 2019) and GAL (Kariyappa & Qureshi, 2019) training on large radii for both MNIST and CIFAR-10 datasets.

**Certified Accuracy with Different Perturbation Radius.** We visualize the trend of certified accuracy along with different perturbation radii in Figure 3. For each radius $r$, we present the best certified accuracy among different smoothing parameters $\sigma \in \{0.25, 0.50, 1.00\}$. We notice that while simply applying MME or WE protocol could slightly improve the certified accuracy, DRT could significantly boost the certified accuracy under different radii. We also present the trends of different smoothing parameters separately in Appendix F which lead to similar conclusions.

Table 3: Certified accuracy under different radii on ImageNet dataset. The grey rows present the performance of the proposed DRT approach. The brackets show the base models we use.

| Radius $r$ | 0.00 | 0.50 | 1.00 | 1.50 | 2.00 | 2.50 | 3.00 |
|---|---|---|---|---|---|---|---|
| Gaussian (Cohen et al., 2019) | 57.2 | 46.2 | 37.0 | 29.2 | 19.6 | 15.2 | 12.4 |
| SmoothAdv (Salman et al., 2019) | 54.6 | 49.0 | 43.8 | 37.2 | 27.0 | 25.2 | 20.4 |
| MACER (Zhai et al., 2019) | **68.0** | **57.0** | 43.0 | 31.0 | 25.0 | 18.0 | 14.0 |
| SWEEN (Gaussian) (Liu et al., 2020) | 58.4 | 47.0 | 37.4 | 29.8 | 20.2 | 15.8 | 12.8 |
| SWEEN (SmoothAdv) (Liu et al., 2020) | 55.2 | 50.0 | 44.2 | 37.8 | 27.6 | 26.6 | 21.6 |
| MME (Gaussian) | 58.0 | 47.2 | 38.8 | 31.2 | 21.4 | 16.4 | 14.2 |
| DRT + MME (Gaussian) | 52.2 | 46.8 | 42.4 | 34.2 | 24.0 | 19.6 | 18.0 |
| MME (SmoothAdv) | 55.0 | 50.2 | 44.2 | 38.6 | 27.4 | 26.4 | 21.6 |
| DRT + MME (SmoothAdv) | 49.8 | 46.8 | **44.4** | **39.8** | 30.2 | 28.2 | **23.4** |
| WE (Gaussian) | 58.2 | 47.2 | 38.6 | 31.2 | 21.6 | 17.0 | 14.4 |
| DRT + WE (Gaussian) | 52.2 | 46.8 | 41.8 | 33.6 | 24.2 | 19.8 | 18.4 |
| WE (SmoothAdv) | 55.2 | 50.2 | **44.4** | 38.6 | 28.2 | 26.2 | 22.0 |
| DRT + WE (SmoothAdv) | 49.8 | 46.6 | **44.4** | 38.8 | **30.4** | **29.0** | 23.2 |

**Effects of GD and CM Losses in DRT.** To explore the effects of individual Gradient Diversity and Confidence Margin Losses in DRT, we set $\rho_1$ or $\rho_2$ to 0 separately and tune the other for evaluation on MNIST and CIFAR-10. The full results are shown in Appendix G.1. We observe that both GD and CM losses have positive effects on improving the certified accuracy, and GD plays a major role on larger radii. By combining these two regularization losses as DRT does, the ensemble model achieves the highest certified accuracy under all radii.

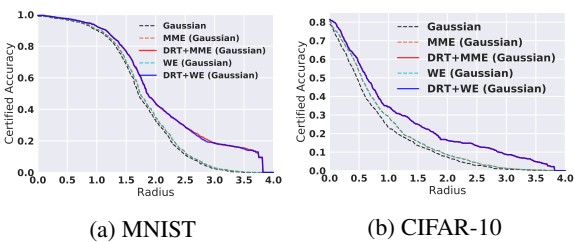

(a) MNIST        (b) CIFAR-10

Figure 3: Certified accuracy for ML ensembles with Gaussian smoothed base models, under smoothing parameter $\sigma \in \{0.25, 0.50, 1.00\}$ on (Left) MNIST; (Right) CIFAR-10.

## 5 CONCLUSION

In this paper, we explored and characterized the robustness conditions for certifiably robust ensemble ML models theoretically, and proposed DRT for training a robust ensemble. Our analysis provided the justification of the regularization-based training approach DRT. Extensive experiments showed that DRT-enhanced ensembles achieve the highest certified robustness compared with existing baselines.

**Ethics Statement.** In this paper, we characterized the robustness conditions for certifying ML ensemble robustness. Based on the analysis, we propose DRT to train a certifiably robust ensemble. On the one hand, the training approach boosts the certified robustness of ML ensemble, thus significantly reducing the security vulnerabilities of ML ensemble. On the other hand, the trained ML ensemble can only guarantee its robustness under specific conditions of the attack. Specifically, we evaluate the trained ML ensemble on the held-out test set and constrain the attack to be within predefined $L_2$ distance from the original input. We cannot provide robustness guarantee for all possible real-world inputs. Therefore, users should be aware of such limitations of DRT-trained ensembles, and should not blindly rely on the ensembles when the attack can cause large deviations measured by $L_2$ distance. As a result, we encourage researchers to understand the potential risks, and evaluate whether our attack constraints align with their usage scenarios when applying our DRT approach to real-world applications. We do not expect any ethics issues raised by our work.

**Reproducibility Statement.** All the theorem statements are substantiated with rigorous proofs in Appendices B to D. In Appendix F, we list the details and hyperparameters for reproducing all experimental results. Our evaluation is conducted on commonly accessible MNIST, CIFAR-10, and ImageNet datasets. Finally, we upload the source code as the supplementary material for reproducibility purpose.

## ACKNOWLEDGEMENTS

This work was performed under the auspices of the U.S. Department of Energy by the Lawrence Livermore National Laboratory under Contract No. DE-AC52-07NA27344 and LLNL LDRD Program Project No. 20-ER-014. This work is partially supported by the NSF grant No.1910100, NSF CNS 20-46726 CAR, Alfred P. Sloan Fellowship, and Amazon Research Award.

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

In Appendix A, we provide more background knowledge about randomized smoothing. In Appendix B, we first discuss the direct connection between the definition of $r$-robustness and the robustness certification of randomized smoothing, then prove the robustness conditions and the comparison results presented in Section 2. In Appendix C, we formally define, discuss, and theoretically compare the smoothing strategies for ensembles. In Appendix D, we characterize the robustness of smoothed ML ensembles from statistical robustness perspective which is directly related to the robustness certification of randomized smoothing. In Appendix E, we present and analyze some alternative designs of DRT. In Appendix F, we show the detailed experimental setup and full experiment results. Finally, in Appendix G, we conduct abalation studies on the effects of Gradient Diversity and Confidence Margin Losses in DRT in Appendix G.1, certified robustness of single base model within DRT-trained ensemble in Appendix G.2, and investigate how optimizing ensemble weights can further improve the certified robustness of DRT-trained ensemble in Appendix G.3. We also analyze other gradient diversity promoted regularizers' performance and compare them with DRT in Appendix G.4.

## A    BACKGROUND: RANDOMIZED SMOOTHING

Cohen et al. (Cohen et al., 2019) leverage Neyman-Pearson Lemma Neyman & Pearson (1933) to provide a computable robustness certification for the smoothed classifier.

**Lemma A.1** (Robustness Certificate of Randomized Smoothing; (Cohen et al., 2019)). *At point $\boldsymbol{x}_0$, let random variable $\varepsilon \sim \mathcal{N}(0, \sigma^2 \boldsymbol{I}_d)$, a smoothed model $G_F^\varepsilon$ is $r$-robust where*

$$r = \frac{\sigma}{2} \left( \Phi^{-1} \left( g_F^\varepsilon(\boldsymbol{x}_0)_{c_A} \right) - \Phi^{-1} \left( g_F^\varepsilon(\boldsymbol{x}_0)_{c_B} \right) \right), \tag{11}$$

*where $c_A = G_F^\varepsilon(\boldsymbol{x}_0)$ and $c_B = G_F^{\varepsilon(2)}(\boldsymbol{x}_0)$ are the top and runner-up class respectively, and $\Phi^{-1}$ is the inverse cumulative distribution function (CDF) of standard normal distribution.*

In practice, for ease of sampling, the standard way of computing the certified radius is using the lower bound of Equation (11): $r = \sigma \Phi^{-1}(g_F^\varepsilon(\boldsymbol{x}_0)_{c_A})$. Now we only need to figure out $g_F^\varepsilon(\boldsymbol{x}_0)_{c_A}$. The common way is to use Monte-Carlo sampling together with binomial confidence interval Cohen et al. (2019); Yang et al. (2020a); Zhai et al. (2019); Jeong & Shin (2020). Concretely, from definition $g_F^\varepsilon(\boldsymbol{x}_0)_{c_A} = \Pr_{\varepsilon \sim \mathcal{N}(0,\sigma^2 \boldsymbol{I}_d)}(F(\boldsymbol{x}_0 + \varepsilon) = c_A)$, we sample $n$ Gaussian noises: $\varepsilon_1, \varepsilon_2, \ldots, \varepsilon_n \sim \mathcal{N}(0, \sigma^2 \boldsymbol{I}_d)$ and compute the empirical mean: $\hat{g}_F^\varepsilon(\boldsymbol{x}_0)_{c_A} = \frac{1}{n} \sum_{i=1}^n \mathbb{I}[F(\boldsymbol{x}_0 + \varepsilon_i) = c_A]$. The binomial testing Clopper & Pearson (1934) then gives a high-confidence lower bound of $g_F^\varepsilon(\boldsymbol{x}_0)_{c_A}$ based on $\hat{g}_F^\varepsilon(\boldsymbol{x}_0)_{c_A}$. We follow the setting in the literature: sample $n = 10^5$ samples and set confidence level be $99.9\%$ Cohen et al. (2019); Yang et al. (2020a); Zhai et al. (2019); Jeong & Shin (2020). More details are available in Appendix F. Note that $F$ could be either single model or any ensemble models with *Ensemble-before-Smoothing* strategy.

## B    DETAILED ANALYSIS AND PROOFS IN SECTION 2

In this appendix, we first show the omitted theoretical results in Section 2, which are the robustness conditions for Max-Margin Ensemble (MME) and the comparison between the robustness of ensemble model and single model. We then present all proofs for these theoretical results.

### B.1    DETAILED THEORETICAL RESULTS AND DISCUSSION

Here we present the theoretical results omitted from Section 2 along with some discussions.

### B.1.1    ROBUSTNESS CONDITION OF MME

For MME, we have the following robustness condition.

**Theorem 3** (Gradient and Confidence Margin Condition for MME Robustness). *Given input $\boldsymbol{x}_0 \in \mathbb{R}^d$ with ground-truth label $y_0 \in [C]$, and $\mathcal{M}_{\mathrm{MME}}$ as an MME defined over base models $\{F_1, F_2\}$. $\mathcal{M}_{\mathrm{MME}}(\boldsymbol{x}_0) = y_0$. Both $F_1$ and $F_2$ are $\beta$-smooth.*

- *(Sufficient Condition) If for any $y_1, y_2 \in [C]$ such that $y_1 \neq y_0$ and $y_2 \neq y_0$,*

$$\|\nabla_{\boldsymbol{x}} f_1^{y_0/y_1}(\boldsymbol{x}_0) + \nabla_{\boldsymbol{x}} f_2^{y_0/y_2}(\boldsymbol{x}_0)\|_2 \leq \frac{1}{r}(f_1^{y_0/y_1}(\boldsymbol{x}_0) + f_2^{y_0/y_2}(\boldsymbol{x}_0)) - 2\beta r, \quad (12)$$

*then $\mathcal{M}_{\mathrm{MME}}$ is $r$-robust at point $\boldsymbol{x}_0$.*

- *(Necessary Condition) Suppose for any $\boldsymbol{x} \in \{\boldsymbol{x}_0 + \boldsymbol{\delta} : ||\boldsymbol{\delta}||_2 \leq r\}$, for any $i \in \{1, 2\}$, either $F_i(\boldsymbol{x}) = y_0$ or $F_i^{(2)}(\boldsymbol{x}) = y_0$. If $\mathcal{M}_{\mathrm{MME}}$ is $r$-robust at point $\boldsymbol{x}_0$, then for any $y_1, y_2 \in [C]$ such that $y_1 \neq y_0$ and $y_2 \neq y_0$,*

$$\|\nabla_{\boldsymbol{x}} f_1^{y_0/y_1}(\boldsymbol{x}_0) + \nabla_{\boldsymbol{x}} f_2^{y_0/y_2}(\boldsymbol{x}_0)\|_2 \leq \frac{1}{r}(f_1^{y_0/y_1}(\boldsymbol{x}_0) + f_2^{y_0/y_2}(\boldsymbol{x}_0)) + 2\beta r. \quad (13)$$

Comparing with the robustness conditions of MME (Theorem 1), the conditions for MME have highly similar forms. Thus, the discussion for ERI in main text (Equation (6)) still applies here, including the positive impact of diversified gradients and large confidence margins towards MME ensemble robustness in both sufficient and necessary conditions and the implication of small model-smoothness bound $\beta$. A major distinction is that the condition for MME is limited to two base models. This is because the "maximum" operator in MME protocol poses difficulties for expressing the robust conditions in succinct continuous functions of base models' confidence. Therefore, Taylor expansion cannot be applied. We leave the extension to $N > 2$ base models as future work, and we conjecture the tendency would be similar as Equation (5). The theorem is proved in Appendix B.2.

### B.1.2 COMPARISON BETWEEN ENSEMBLE ROBUSTNESS AND SINGLE-MODEL ROBUSTNESS

To compare the robustness of ensemble models and single models, we have the following corollary that is extended from Theorem 1 and Theorem 3.

**Corollary 1** (Comparison of Ensemble and Single-Model Robustness). *Given an input $\boldsymbol{x}_0 \in \mathbb{R}^d$ with ground-truth label $y_0 \in [C]$. Suppose we have two $\beta$-smooth base models $\{F_1, F_2\}$, which are both $r$-robust at point $\boldsymbol{x}_0$. For any $\Delta \in [0, 1)$:*

- *(Weighted Ensemble) Define Weighted Ensemble $\mathcal{M}_{\mathrm{WE}}$ with base models $\{F_1, F_2\}$. Suppose $\mathcal{M}_{\mathrm{WE}}(\boldsymbol{x}_0) = y_0$. If for any label $y_i \neq y_0$, the base models' smoothness $\beta \leq \Delta \cdot \min\{f_1^{y_0/y_i}(\boldsymbol{x}_0), f_2^{y_0/y_i}(\boldsymbol{x}_0)\}/(c^2 r^2)$, and the gradient cosine similarity $\cos\langle\nabla_{\boldsymbol{x}} f_1^{y_0/y_i}(\boldsymbol{x}_0), \nabla_{\boldsymbol{x}} f_2^{y_0/y_i}(\boldsymbol{x}_0)\rangle \leq \cos\theta$, then the $\mathcal{M}_{\mathrm{WE}}$ with weights $\{w_1, w_2\}$ is at least $R$-robust at point $\boldsymbol{x}_0$ with*

$$R = r \cdot \frac{1 - \Delta}{1 + \Delta} \left(1 - C_{\mathrm{WE}}(1 - \cos\theta)\right)^{-1/2}, where \quad (14)$$

$$C_{\mathrm{WE}} = \min_{y_i: y_i \neq y_0} \frac{2 w_1 w_2 f_1^{y_0/y_i}(\boldsymbol{x}_0) f_2^{y_0/y_i}(\boldsymbol{x}_0)}{(w_1 f_1^{y_0/y_i}(\boldsymbol{x}_0) + w_2 f_2^{y_0/y_i}(\boldsymbol{x}_0))^2}, c = \max\{\frac{1-\Delta}{1+\Delta}\left(1 - C_{\mathrm{WE}}(1 - \cos\theta)\right)^{-1/2}, 1\}.$$

- *(Max-Margin Ensemble) Define Max-Margin Ensemble $\mathcal{M}_{\mathrm{MME}}$ with the base models $\{F_1, F_2\}$. Suppose $\mathcal{M}_{\mathrm{MME}}(\boldsymbol{x}_0) = y_0$. If for any label $y_1 \neq y_0$ and $y_2 \neq y_0$, the base models' smoothness $\beta \leq \Delta \cdot \min\{f_1^{y_0/y_1}(\boldsymbol{x}_0), f_2^{y_0/y_2}(\boldsymbol{x}_0)\}/(c^2 r^2)$, and the gradient cosine similarity $\cos\langle\nabla_{\boldsymbol{x}} f_1^{y_0/y_1}(\boldsymbol{x}_0), \nabla_{\boldsymbol{x}} f_2^{y_0/y_2}(\boldsymbol{x}_0)\rangle \leq \cos\theta$, then the $\mathcal{M}_{\mathrm{MME}}$ is at least $R$-robust at point $\boldsymbol{x}_0$ with*

$$R = r \cdot \frac{1 - \Delta}{1 + \Delta} \left(1 - C_{\mathrm{MME}}(1 - \cos\theta)\right)^{-1/2}, where \quad (15)$$

$$C_{\mathrm{MME}} = \min_{\substack{y_1, y_2: \\ y_1, y_2 \neq y_0}} \frac{2 f_1^{y_0/y_1}(\boldsymbol{x}_0) f_2^{y_0/y_2}(\boldsymbol{x}_0)}{(f_1^{y_0/y_1}(\boldsymbol{x}_0) + f_2^{y_0/y_2}(\boldsymbol{x}_0))^2}, c = \max\{\frac{1-\Delta}{1+\Delta}\left(1 - C_{\mathrm{MME}}(1 - \cos\theta)\right)^{-1/2}, 1\}.$$

The proof is given in Appendix B.3.

**Optimizing Weighted Ensemble.** As we can observe from Corollary 1, we can adjust the weights $\{w_1, w_2\}$ for Weighted Ensemble to change $C_{\mathrm{WE}}$ and the certified robust radius (Equation (14)).

Then comes the problem of which set of weights can achieve the highest certified robust radius. Since larger $C_{\mathrm{WE}}$ results in higher radius, we need to choose

$$(w_1^{OPT}, w_2^{OPT}) = \arg \max_{w_1, w_2} \min_{y_i : y_i \neq y_0} \frac{2w_1 w_2 f_1^{y_0/y_i}(\boldsymbol{x}_0) f_2^{y_0/y_i}(\boldsymbol{x}_0)}{(w_1 f_1^{y_0/y_i}(\boldsymbol{x}_0) + w_2 f_2^{y_0/y_i}(\boldsymbol{x}_0))^2}.$$

Since this quantity is scale-invariant, we can fix $w_1$ and optimize over $w_2$ to get the optimal weights. In particular, if there are only two classes, we have a closed-form solution

$$\begin{aligned} (w_1^{OPT}, w_2^{OPT}) &= \arg \max_{w_1, w_2} \frac{2w_1 w_2 f_1^{y_0/y_1}(\boldsymbol{x}_0) f_2^{y_0/y_1}(\boldsymbol{x}_0)}{(w_1 f_1^{y_0/y_1}(\boldsymbol{x}_0) + w_2 f_2^{y_0/y_1}(\boldsymbol{x}_0))^2} \\ &= \{k \cdot f_2^{y_0/y_1}(\boldsymbol{x}_0), k \cdot f_1^{y_0/y_1}(\boldsymbol{x}_0) : k \in \mathbb{R}_+\}, \end{aligned}$$

and corresponding $C_{\mathrm{WE}}$ achieves the maximum $1/2$.

For a special case—average weighted ensemble, we get the corresponding certified robust radius by setting $w_1 = w_2$ and plug the yielded

$$C_{\mathrm{WE}} = \min_{y_i : y_i \neq y_0} \frac{2 f_1^{y_0/y_i}(\boldsymbol{x}_0) f_2^{y_0/y_i}(\boldsymbol{x}_0)}{(f_1^{y_0/y_i}(\boldsymbol{x}_0) + f_2^{y_0/y_i}(\boldsymbol{x}_0))^2} \in (0, 1/2].$$

into Equation (14).

**Comparison between ensemble and single-model robustness.** The similar forms of $R$ in the corollary allow us to discuss the Weighted Ensemble and Max-Margin Ensemble together. Specifically, we let $C$ be either $C_{\mathrm{WE}}$ or $C_{\mathrm{MME}}$, then

$$R = r \cdot \frac{1 - \Delta}{1 + \Delta} \left(1 - C(1 - \cos\theta)\right)^{-1/2}.$$

Since when $R > r$, both ensembles have higher certified robustness than the base models, we solve this condition for $\cos\theta$:

$$R > r \iff \left(\frac{1 - \Delta}{1 + \Delta}\right)^2 > 1 - C(1 - \cos\theta) \iff \cos\theta \leq 1 - \frac{4\Delta}{C(1 + \Delta)^2}.$$

Notice that $C \in (0, 1/2]$. From this condition, we can easily observe that when the gradient cosine similarity is smaller, it is more likely that the ensemble has higher certified robustness than the base models. When the model is smooth enough, according to the condition on $\beta$, we can notice that $\Delta$ could be close to zero. As a result, $1 - \frac{4\Delta}{C(1+\Delta)^2}$ is close to 1. *Thus, unless the gradient of base models is (or close to) colinear, it always holds that the ensemble (either WE or MME) has higher certified robustness than the base models.*

**Larger certified radius with larger number of base models $N$.** Following the same methodology, we can further observe that larger number of base models $N$ can lead to larger certified radius as the following proposition shows.

**Proposition B.1** (More Base Models Lead to Higher Certified Robustness of Weighted Ensemble). *At clean input $\boldsymbol{x}_0 \in \mathbb{R}^d$ with ground-truth label $y_0 \in [C]$, suppose all base models $\{f_i\}_{i=1}^{N+M}$ are $\beta$-smooth. Suppose the Weighted Ensemble $\mathcal{M}_1$ of base models $\{f_i\}_{i=1}^{N}$ and $\mathcal{M}_2$ of base models $\{f_i\}_{i=N+1}^{N+M}$ are both $r$-robust according to the sufficient condition in Theorem 1, and for any $y_i \neq y_0$ the $\mathcal{M}_1$ and $\mathcal{M}_2$'s ensemble gradients ($\sum_{j=1}^{N} w_j \nabla_{\boldsymbol{x}} f_j^{y_0/y_i}(\boldsymbol{x}_0)$ and $\sum_{j=N+1}^{N+M} w_j \nabla_{\boldsymbol{x}} f_j^{y_0/y_i}(\boldsymbol{x}_0)$) are non-zero and not colinear, then the Weighted Ensemble $\mathcal{M}$ of $\{f_i\}_{i=1}^{N+M}$ is $r'$-robust for some $r' > r$.*

*Proof of Proposition B.1.* For any $y_i \neq y_0$, since both $\mathcal{M}_1$ and $\mathcal{M}_2$ are $r$-robust according to the sufficient condition of Theorem 1, we have

$$\left\| \sum_{j=1}^{N} w_j \nabla_{\boldsymbol{x}} f_j^{y_0/y_i}(\boldsymbol{x}_0) \right\|_2 \leq \frac{1}{r} \sum_{j=1}^{N} f_j^{y_0/y_i}(\boldsymbol{x}_0) - \beta r \sum_{j=1}^{N} w_j, \tag{16}$$

$$\Big\| \sum_{j=N+1}^{N+M} w_j \nabla_{\boldsymbol{x}} f_j^{y_0/y_i}(\boldsymbol{x}_0) \Big\|_2 \le \frac{1}{r} \sum_{j=N+1}^{N+M} f_j^{y_0/y_i}(\boldsymbol{x}_0) - \beta r \sum_{j=N+1}^{N+M} w_j. \tag{17}$$

Adding above two inequalties we get

$$\Big\| \sum_{j=1}^{N} w_j \nabla_{\boldsymbol{x}} f_j^{y_0/y_i}(\boldsymbol{x}_0) \Big\|_2 + \Big\| \sum_{j=N+1}^{N+M} w_j \nabla_{\boldsymbol{x}} f_j^{y_0/y_i}(\boldsymbol{x}_0) \Big\|_2 \le \frac{1}{r} \sum_{j=1}^{N+M} f_j^{y_0/y_i}(\boldsymbol{x}_0) - \beta r \sum_{j=1}^{N+M} w_j. \tag{18}$$

Since gradients of ensemble are not colinear and non-zero, from the triangle inequality,

$$\Big\| \sum_{j=1}^{N+M} w_j \nabla_{\boldsymbol{x}} f_j^{y_0/y_i}(\boldsymbol{x}_0) \Big\|_2 < \Big\| \sum_{j=1}^{N} w_j \nabla_{\boldsymbol{x}} f_j^{y_0/y_i}(\boldsymbol{x}_0) \Big\|_2 + \Big\| \sum_{j=N+1}^{N+M} w_j \nabla_{\boldsymbol{x}} f_j^{y_0/y_i}(\boldsymbol{x}_0) \Big\|_2 \tag{19}$$

and thus

$$\Big\| \sum_{j=1}^{N+M} w_j \nabla_{\boldsymbol{x}} f_j^{y_0/y_i}(\boldsymbol{x}_0) \Big\|_2 < \frac{1}{r} \sum_{j=1}^{N+M} f_j^{y_0/y_i}(\boldsymbol{x}_0) - \beta r \sum_{j=1}^{N+M} w_j, \tag{20}$$

which means we can increase $r$ to $r'$ and still keep the inequality hold with "$\le$", and in turn certify a larger radius $r'$ according to Theorem 1. $\qquad\square$

Since DRT imposes diversified gradients via GD Loss, the "not colinear" condition easily holds for DRT ensemble, and therefore the proposition implies larger number of base models $N$ lead to higher certified robustness of WE. For MME, we empirically observe similar trends.

### B.1.3 ROBUSTNESS CONDITIONS FOR SMOOTHED ENSEMBLE MODELS

Following the discussion in Section 2.3, for *smoothed* WE and MME, with the model-smoothness bound (Theorem 2), we can concretize the general robustness conditions in this way.

We define the *soft* smoothed confidence function $\bar{g}_f^\varepsilon(\boldsymbol{x}) := \mathbb{E}_\varepsilon f(\boldsymbol{x} + \varepsilon)$. This definition is also used in the literature (Salman et al., 2019; Zhai et al., 2019; Kumar et al., 2020). As a result, we revise the ensemble protocols of WE and MME by replacing the original confidences $\{f_i(\boldsymbol{x}_0)\}_{i=1}^N$ with these soft smoothed confidences $\{\bar{g}_i^\varepsilon(\boldsymbol{x}_0)\}_{i=1}^N$. These protocols then choose the predicted class by treating these $\{\bar{g}_i^\varepsilon(\boldsymbol{x}_0)\}_{i=1}^N$ as the base models' confidence scores. In the experiments, we did not actually evaluate these revised protocols since their robustness performance are expected to be similar as original ones (Salman et al., 2019; Zhai et al., 2019; Kumar et al., 2020). The derived results connect smoothed ensemble robustness with the confidence scores.

**Corollary 2** (Gradient and Confidence Margin Conditions for Smoothed WE Robustness). *Given input $\boldsymbol{x}_0 \in \mathbb{R}^d$ with ground-truth label $y_0 \in [C]$. Let $\varepsilon \sim \mathcal{N}(0, \sigma^2 \boldsymbol{I}_d)$ be a Gaussian random variable. Define soft smoothed confidence $\bar{g}_i^\varepsilon(\boldsymbol{x}) := \mathbb{E}_\varepsilon f_i(\boldsymbol{x} + \varepsilon)$ for each base model $F_i$ ($1 \le i \le N$). The $\bar{G}_{\mathcal{M}_{\mathrm{WE}}}^\varepsilon$ is a WE defined over soft smoothed base models $\{\bar{g}_i^\varepsilon\}_{i=1}^N$ with weights $\{w_i\}_{i=1}^N$. $\bar{G}_{\mathcal{M}_{\mathrm{WE}}}^\varepsilon(\boldsymbol{x}_0) = y_0$.*

- *(Sufficient Condition) The $\bar{G}_{\mathcal{M}_{\mathrm{WE}}}^\varepsilon$ is $r$-robust at point $\boldsymbol{x}_0$ if for any $y_i \ne y_0$,*

$$\Big\| \sum_{j=1}^{N} w_j \nabla_{\boldsymbol{x}} (\bar{g}_j^\varepsilon)^{y_0/y_i}(\boldsymbol{x}_0) \Big\|_2 \le \frac{1}{r} \sum_{j=1}^{N} w_j (\bar{g}_j^\varepsilon)^{y_0/y_i}(\boldsymbol{x}_0) - \frac{2r}{\sigma^2} \sum_{j=1}^{N} w_j, \tag{21}$$

- *(Necessary Condition) If $\bar{G}_{\mathcal{M}_{\mathrm{WE}}}^\varepsilon$ is $r$-robust at point $\boldsymbol{x}_0$, for any $y_i \ne y_0$,*

$$\Big\| \sum_{j=1}^{N} w_j \nabla_{\boldsymbol{x}} (\bar{g}_j^\varepsilon)^{y_0/y_i}(\boldsymbol{x}_0) \Big\|_2 \le \frac{1}{r} \sum_{j=1}^{N} w_j (\bar{g}_j^\varepsilon)^{y_0/y_i}(\boldsymbol{x}_0) + \frac{2r}{\sigma^2} \sum_{j=1}^{N} w_j. \tag{22}$$

**Corollary 3** (Gradient and Confidence Margin Condition for Smoothed MME Robustness). *Given input $\boldsymbol{x}_0 \in \mathbb{R}^d$ with ground-truth label $y_0 \in [C]$. Let $\varepsilon \sim \mathcal{N}(0, \sigma^2 \boldsymbol{I}_d)$ be a Gaussian random variable. Define soft smoothed confidence $\bar{g}_i^\varepsilon(\boldsymbol{x}) := \mathbb{E}_\varepsilon f_i(\boldsymbol{x} + \varepsilon)$ for either base model $F_1$ or $F_2$. The $\bar{G}_{\mathcal{M}_{\mathrm{MME}}}^\varepsilon$ is a MME defined over soft smoothed base models $\{\bar{g}_1^\varepsilon, \bar{g}_2^\varepsilon\}$. $\bar{G}_{\mathcal{M}_{\mathrm{MME}}}^\varepsilon(\boldsymbol{x}_0) = y_0$.*

- *(Sufficient Condition) If for any $y_1, y_2 \in [C]$ such that $y_1 \neq y_0$ and $y_2 \neq y_0$,*

$$\|\nabla_{\boldsymbol{x}}(\bar{g}_1^\varepsilon)^{y_0/y_1}(\boldsymbol{x}_0) + \nabla_{\boldsymbol{x}}(\bar{g}_2^\varepsilon)^{y_0/y_2}(\boldsymbol{x}_0)\|_2 \leq \frac{1}{r}((\bar{g}_1^\varepsilon)^{y_0/y_1}(\boldsymbol{x}_0) + (\bar{g}_2^\varepsilon)^{y_0/y_2}(\boldsymbol{x}_0)) - \frac{4r}{\sigma^2},$$
(23)

  *then $\bar{G}^\varepsilon_{\mathcal{M}_{\mathrm{MME}}}$ is $r$-robust at point $\boldsymbol{x}_0$.*

- *(Necessary Condition) Suppose for any $\boldsymbol{x} \in \{\boldsymbol{x}_0 + \boldsymbol{\delta} : \|\boldsymbol{\delta}\|_2 \leq r\}$, for any $i \in \{1, 2\}$, either $G_{F_i}(\boldsymbol{x}) = y_0$ or $G_{F_i}^{(2)}(\boldsymbol{x}) = y_0$. If $\bar{G}^\varepsilon_{\mathcal{M}_{\mathrm{MME}}}$ is $r$-robust at point $\boldsymbol{x}_0$, then for any $y_1, y_2 \in [C]$ such that $y_1 \neq y_0$ and $y_2 \neq y_0$,*

$$\|\nabla_{\boldsymbol{x}}(\bar{g}_1^\varepsilon)^{y_0/y_1}(\boldsymbol{x}_0) + \nabla_{\boldsymbol{x}}(\bar{g}_2^\varepsilon)^{y_0/y_2}(\boldsymbol{x}_0)\|_2 \leq \frac{1}{r}((\bar{g}_1^\varepsilon)^{y_0/y_1}(\boldsymbol{x}_0) + (\bar{g}_2^\varepsilon)^{y_0/y_2}(\boldsymbol{x}_0)) + \frac{4r}{\sigma^2}.$$
(24)

*Remark.* The above two corollaries are extended from Theorem 1 and Theorem 3 respectively, and correspond to our discussion in Section 2.3. We defer the proofs to Appendix B.5. From these two corollaries, we can explicit see that the *Ensemble-before-Smoothing* (see Definition 5) provides smoothed classifiers $\bar{G}^\varepsilon_{\mathcal{M}_{\mathrm{WE}}}$ and $\bar{G}^\varepsilon_{\mathcal{M}_{\mathrm{MME}}}$ with bounded model-smoothness; and we can see the correlation between the robustness conditions and gradient diversity/confidence margin for smoothed ensembles.

## B.2 Proofs of Robustness Conditions for General Ensemble Models

This subsection contains the proofs of robustness conditions. First, we connect the prediction of ensemble models with the arithmetic relations of confidence scores of base models. This connection is straightforward to establish for Weighted Ensemble (shown in Proposition B.2), but nontrivial for Max-Margin Ensemble (shown in Theorem 4). Then, we prove the desired robustness conditions using Taylor expansion with Lagrange reminder.

**Proposition B.2** (Robustness Condition for WE). *Consider an input $\boldsymbol{x}_0 \in \mathbb{R}^d$ with ground-truth label $y_0 \in [C]$, and an ensemble model $\mathcal{M}_{\mathrm{WE}}$ constructed by base models $\{F_i\}_{i=1}^N$ with weights $\{w_i\}_{i=1}^N$. Suppose $\mathcal{M}_{\mathrm{WE}}(\boldsymbol{x}_0) = y_0$. Then, the ensemble $\mathcal{M}_{\mathrm{WE}}$ is $r$-robust at point $\boldsymbol{x}_0$ if and only if for any $\boldsymbol{x} \in \{\boldsymbol{x}_0 + \boldsymbol{\delta} : \|\boldsymbol{\delta}\|_2 \leq r\}$,*

$$\min_{y_i \in [C]:y_i \neq y_0} \sum_{j=1}^N w_j f_j^{y_0/y_i}(\boldsymbol{x}) \geq 0.$$
(25)

*Proof of Proposition B.2.* According the the definition of $r$-robust, we know $\mathcal{M}_{\mathrm{WE}}$ is $r$-robust if and only if for any point $\boldsymbol{x} := \boldsymbol{x}_0 + \boldsymbol{\delta}$ where $\|\boldsymbol{\delta}\|_2 \leq r$, $\mathcal{M}_{\mathrm{WE}}(\boldsymbol{x}_0 + \boldsymbol{\delta}) = y_0$, which means that for any other label $y_i \neq y_0$, the confidence score for label $y_0$ is larger or equal than the confidence score for label $y_i$. It means that

$$\sum_{j=1}^N w_j f_j(\boldsymbol{x})_{y_0} \geq \sum_{j=1}^N w_j f_j(\boldsymbol{x})_{y_i}$$

for any $\boldsymbol{x} \in \{\boldsymbol{x}_0 + \boldsymbol{\delta} : \|\boldsymbol{\delta}\|_2 \leq r\}$. Since this should hold for any $y_i \neq y_0$, we have the sufficient and necessary condition

$$\min_{y_i \in [C]:y_i \neq y_0} \sum_{j=1}^N w_j f_j^{y_0/y_i}(\boldsymbol{x}) \geq 0.$$
(25)

$\square$

**Theorem 4** (Robustness Condition for MME). *Consider an input $\boldsymbol{x}_0 \in \mathbb{R}^d$ with ground-truth label $y_0 \in [C]$. Let $\mathcal{M}_{\mathrm{MME}}$ be an MME defined over base models $\{F_i\}_{i=1}^N$. Suppose: (1) $\mathcal{M}_{\mathrm{MME}}(\boldsymbol{x}_0) = y_0$; (2) for any $\boldsymbol{x} \in \{\boldsymbol{x}_0 + \boldsymbol{\delta} : \|\boldsymbol{\delta}\|_2 \leq r\}$, given any base model $i \in [N]$, either $F_i(\boldsymbol{x}) = y_0$ or $F_i^{(2)}(\boldsymbol{x}) = y_0$. Then, the ensemble $\mathcal{M}_{\mathrm{MME}}$ is $r$-robust at point $\boldsymbol{x}_0$ if and only if for any $\boldsymbol{x} \in \{\boldsymbol{x}_0 + \boldsymbol{\delta} : \|\boldsymbol{\delta}\|_2 \leq r\}$,*

$$\max_{i \in [N]} \min_{y_i \in [C]:y_i \neq y_0} f_i^{y_0/y_i}(\boldsymbol{x}) \geq \max_{i \in [N]} \min_{y_i' \in [C]:y_i' \neq y_0} f_i^{y_i'/y_0}(\boldsymbol{x}).$$
(26)

The theorem states the sufficient and necessary robustness condition for MME. We divide the two directions into the following two lemmas and prove them separately. We mainly use the alternative form of Equation (26) as such in the following lemmas and their proofs:

$$\max_{i\in[N]} \min_{y_i\in[C]:y_i\neq y_0} f_i^{y_0/y_i}(\boldsymbol{x}) + \min_{i\in[N]} \min_{y_i'\in[C]:y_i'\neq y_0} f_i^{y_0/y_i'}(\boldsymbol{x}) \geq 0. \tag{26}$$

**Lemma B.1** (Sufficient Condition for MME). *Let $\mathcal{M}_{\mathrm{MME}}$ be an MME defined over base models $\{F_i\}_{i=1}^N$. For any input $\boldsymbol{x}_0 \in \mathbb{R}^d$, the Max-Margin Ensemble $\mathcal{M}_{\mathrm{MME}}$ predicts $\mathcal{M}_{\mathrm{MME}}(\boldsymbol{x}_0) = y_0$ if*

$$\max_{i\in[N]} \min_{y_i\in[C]:y_i\neq y_0} f_i^{y_0/y_i}(\boldsymbol{x}_0) + \min_{i\in[N]} \min_{y_i'\in[C]:y_i'\neq y_0} f_i^{y_0/y_i'}(\boldsymbol{x}_0) \geq 0. \tag{26}$$

*Proof of Lemma B.1.* For brevity, for $i \in [N]$, we denote $y_i := F_i(\boldsymbol{x}_0), y_i' := F_i^{(2)}(\boldsymbol{x}_0)$ for each base model's top class and runner-up class at point $\boldsymbol{x}_0$.

Suppose $\mathcal{M}_{\mathrm{MME}}(\boldsymbol{x}_0) \neq y_0$, then according to ensemble definition (see Definition 3), there exists $c \in [N]$, such that $\mathcal{M}_{\mathrm{MME}}(\boldsymbol{x}_0) = F_c(\boldsymbol{x}_0) = y_c$, and

$$\forall i \in [N],\ i \neq c,\ f_c(\boldsymbol{x}_0)^{y_c/y_c'} > f_i(\boldsymbol{x}_0)^{y_i/y_i'}. \tag{27}$$

Because $y_c \neq y_0$, we have $f_c(\boldsymbol{x}_0)_{y_0} \leq f_c(\boldsymbol{x}_0)_{y_c'}$, so that $f_c(\boldsymbol{x}_0)^{y_c/y_0} \geq f_c(\boldsymbol{x}_0)^{y_c/y_c'}$. Now consider any model $F_i$ where $i \in [N]$, we would like to show that there exists $y^* \neq y_0$, such that $f_i(\boldsymbol{x}_0)^{y_i/y_i'} \geq f_i(\boldsymbol{x}_0)^{y_0/y^*}$:

- If $y_i = y_0$, let $y^* := y_i'$, trivially $f_i(\boldsymbol{x}_0)^{y_i/y_i'} = f_i(\boldsymbol{x}_0)^{y_0/y^*}$;

- If $y_i \neq y_0$, and $y_i' \neq y_0$, we let $y^* := y_i'$, then $f_i(\boldsymbol{x}_0)^{y_i/y_i'} = f_i(\boldsymbol{x}_0)^{y_i/y^*} \geq f_i(\boldsymbol{x}_0)^{y_0/y^*}$;

- If $y_i \neq y_0$, but $y_i' = y_0$, we let $y^* := y_i$, then $f_i(\boldsymbol{x}_0)^{y_i/y_i'} = f_i(\boldsymbol{x}_0)^{y_i/y_0} \geq f_i(\boldsymbol{x}_0)^{y_0/y_i} = f_i(\boldsymbol{x}_0)^{y_0/y^*}$.

Combine the above findings with Equation 27, we have:

$$\forall i \in [N],\ i \neq c,\ \exists y_c^* \in [C] \text{ and } y_c^* \neq y_0,\ \exists y_i^* \in [C] \text{ and } y_i^* \neq y_0,\ f_c(\boldsymbol{x}_0)^{y_c^*/y_0} > f_i(\boldsymbol{x}_0)^{y_0/y_i^*}.$$

Therefore, its negation

$$\exists i \in [N],\ i \neq c,\ \forall y_c^* \in [C] \text{ and } y_c^* \neq y_0,\ \forall y_i^* \in [C] \text{ and } y_i^* \neq y_0,\ f_c(\boldsymbol{x}_0)^{y_0/y_c^*} + f_i(\boldsymbol{x}_0)^{y_0/y_i^*} \geq 0 \tag{28}$$

implies $\mathcal{M}(\boldsymbol{x}_0) = y_0$. Since Equation (28) holds for any $y_c^*$ and $y_i^*$, the equation is equivalent to

$$\exists i \in [N],\ i \neq c,\ \min_{y_c\in[C]:y_c\neq y_0} f_c(\boldsymbol{x}_0)^{y_0/y_c}(\boldsymbol{x}_0) + \min_{y_i'\in[C]:y_i'\neq y_0} f_i(\boldsymbol{x}_0)^{y_0/y_i'}(\boldsymbol{x}_0) \geq 0.$$

The existence qualifier over $i$ can be replaced by maximum:

$$\min_{y_c\in[C]:y_c\neq y_0} f_c(\boldsymbol{x}_0)^{y_0/y_c}(\boldsymbol{x}_0) + \max_{i\in[N]} \min_{y_i'\in[C]:y_i'\neq y_0} f_i(\boldsymbol{x}_0)^{y_0/y_i'}(\boldsymbol{x}_0) \geq 0.$$

It is implied by

$$\max_{i\in[N]} \min_{y_i\in[C]:y_i\neq y_0} f_i^{y_0/y_i}(\boldsymbol{x}_0) + \min_{i\in[N]} \min_{y_i'\in[C]:y_i'\neq y_0} f_i^{y_0/y_i'}(\boldsymbol{x}_0) \geq 0. \tag{26}$$

Thus, Equation (26) is a sufficient condition for $\mathcal{M}_{\mathrm{MME}}(\boldsymbol{x}_0) = y_0$. $\square$

**Lemma B.2** (Necessary Condition for MME). *For any input $\boldsymbol{x}_0 \in \mathbb{R}^d$, if for any base model $i \in [N]$, either $F_i(\boldsymbol{x}_0) = y_0$ or $F_i^{(2)}(\boldsymbol{x}_0) = y_0$, then Max-Margin Ensemble $\mathcal{M}_{\mathrm{MME}}$ predicting $\mathcal{M}_{\mathrm{MME}}(\boldsymbol{x}_0) = y_0$ implies*

$$\max_{i\in[N]} \min_{y_i\in[C]:y_i\neq y_0} f_i^{y_0/y_i}(\boldsymbol{x}_0) + \min_{i\in[N]} \min_{y_i'\in[C]:y_i'\neq y_0} f_i^{y_0/y_i'}(\boldsymbol{x}_0) \geq 0. \tag{26}$$

*Proof of Lemma B.2.* Similar as before, for brevity, for $i \in [N]$, we denote $y_i := F_i(\boldsymbol{x}_0), y_i' := F_i^{(2)}(\boldsymbol{x}_0)$ for each base model's top class and runner-up class at point $\boldsymbol{x}_0$.

Suppose Equation (26) is not satisfied, it means that

$$\exists c \in [N], \exists y_c^* \in [C] \text{ and } y_c^* \neq y_0, \forall i \in [N], \exists y_i^* \in [C] \text{ and } y_i^* \neq y_0, f_c^{y_c^*/y_0}(\boldsymbol{x}_0) > f_i^{y_0/y_i^*}(\boldsymbol{x}_0).$$

- If $y_c = y_0$, then $f_c^{y_c^*/y_0}(\boldsymbol{x}_0) \leq 0$, which implies that $f_i^{y_0/y_i^*}(\boldsymbol{x}_0) < 0$, and hence $F_i(\boldsymbol{x}_0) \neq y_0$. Moreover, we know that $f_i^{y_i/y_i'}(\boldsymbol{x}_0) = f_i^{y_i/y_0}(\boldsymbol{x}_0) \geq f_i^{y_i^*/y_0}(\boldsymbol{x}_0) > f_c^{y_0/y_c^*}(\boldsymbol{x}_0) \geq f_c^{y_0/y_c'}(\boldsymbol{x}_0) = f_c^{y_c/y_c'}(\boldsymbol{x}_0)$ so $\mathcal{M}(\boldsymbol{x}_0) \neq F_c(\boldsymbol{x}_0) = y_0$.

- If $y_c \neq y_0$, i.e., $y_c' = y_0$, then $f_c^{y_c/y_0}(\boldsymbol{x}_0) \geq f_c^{y_c^*/y_0}(\boldsymbol{x}_0) > f_i^{y_0/y_1^*}(\boldsymbol{x}_0)$. If $F_i(\boldsymbol{x}_0) = y_0$, then $f_i^{y_0/y_i^*}(\boldsymbol{x}_0) \geq f_i^{y_0/y_i'}(\boldsymbol{x}_0) = f_i^{y_i/y_i'}(\boldsymbol{x}_0)$. Thus, $f_c^{y_c/y_c'}(\boldsymbol{x}_0) = f_c^{y_c/y_0}(\boldsymbol{x}_0) > f_i^{y_i/y_i'}(\boldsymbol{x}_0)$. As the result, $\mathcal{M}(\boldsymbol{x}_0) = F_c(\boldsymbol{x}_0) \neq y_0$.

For both cases, we show that $\mathcal{M}_{\mathrm{MME}}(\boldsymbol{x}_0) \neq y_0$, i.e., Equation (26) is a necessary condition for $\mathcal{M}(\boldsymbol{x}_0) = y_0$. $\qquad\square$

*Proof of Theorem 4.* Lemmas B.1 and B.2 are exactly the two directions (necessary and sufficient condition) of $\mathcal{M}_{\mathrm{MME}}$ predicting label $y_0$ at point $\boldsymbol{x}$. Therefore, if the condition (Equation (26)) holds for any $\boldsymbol{x} \in \{\boldsymbol{x}_0 + \boldsymbol{\delta} : \|\boldsymbol{\delta}\|_2 \leq r\}$, the ensemble $\mathcal{M}_{\mathrm{MME}}$ is $r$-robust at point $\boldsymbol{x}_0$; vice versa. $\qquad\square$

For comparison, here we list the trivial robustness condition for single model.

**Fact B.1** (Robustness Condition for Single Model). *Consider an input $\boldsymbol{x}_0 \in \mathbb{R}^d$ with ground-truth label $y_0 \in [C]$. Suppose a model $F$ satisfies $F(\boldsymbol{x}_0) = y_0$. Then, the model $F$ is $r$-robust at point $\boldsymbol{x}_0$ if and only if for any $\boldsymbol{x} \in \{\boldsymbol{x}_0 + \boldsymbol{\delta} : \|\boldsymbol{\delta}\|_2 \leq r\}$,*

$$\min_{y_i \in [C]: y_i \neq y_0} f^{y_0/y_i}(\boldsymbol{x}) \geq 0.$$

The fact is apparent given that the model predicts the class with the highest confidence.

Now we are ready to apply Taylor expansion to derive the robustness conditions shown in main text.

**Theorem 1** (Gradient and Confidence Margin Condition for WE Robustness). *Given input $\boldsymbol{x}_0 \in \mathbb{R}^d$ with ground-truth label $y_0 \in [C]$, and $\mathcal{M}_{\mathrm{WE}}$ as a WE defined over base models $\{F_i\}_{i=1}^N$ with weights $\{w_i\}_{i=1}^N$. $\mathcal{M}_{\mathrm{WE}}(\boldsymbol{x}_0) = y_0$. All base model $F_i$'s are $\beta$-smooth.*

- *(Sufficient Condition) $\mathcal{M}_{\mathrm{WE}}$ is $r$-robust at point $\boldsymbol{x}_0$ if for any $y_i \neq y_0$,*

$$\left\| \sum_{j=1}^N w_j \nabla_{\boldsymbol{x}} f_j^{y_0/y_i}(\boldsymbol{x}_0) \right\|_2 \leq \frac{1}{r} \sum_{j=1}^N w_j f_j^{y_0/y_i}(\boldsymbol{x}_0) - \beta r \sum_{j=1}^N w_j. \tag{4}$$

- *(Necessary Condition) If $\mathcal{M}_{\mathrm{WE}}$ is $r$-robust at point $\boldsymbol{x}_0$, then for any $y_i \neq y_0$,*

$$\left\| \sum_{j=1}^N w_j \nabla_{\boldsymbol{x}} f_j^{y_0/y_i}(\boldsymbol{x}_0) \right\|_2 \leq \frac{1}{r} \sum_{j=1}^N w_j f_j^{y_0/y_i}(\boldsymbol{x}_0) + \beta r \sum_{j=1}^N w_j. \tag{5}$$

*Proof of Theorem 1.* From Taylor expansion with Lagrange remainder and the $\beta$-smoothness assumption on the base models, we have

$$\sum_{j=1}^N w_j f_j^{y_0/y_i}(\boldsymbol{x}_0) - r \left\| \sum_{j=1}^N w_j \nabla_{\boldsymbol{x}} f_j^{y_0/y_i}(\boldsymbol{x}_0) \right\|_2 - \frac{1}{2} r^2 \sum_{j=1}^N (2\beta w_j) \leq \min_{\boldsymbol{x}: \|\boldsymbol{x} - \boldsymbol{x}_0\|_2 \leq r} \sum_{j=1}^N w_j f_j^{y_0/y_i}(\boldsymbol{x})$$

$$\leq \sum_{j=1}^N w_j f_j^{y_0/y_i}(\boldsymbol{x}_0) - r \left\| \sum_{j=1}^N w_j \nabla_{\boldsymbol{x}} f_j^{y_0/y_i}(\boldsymbol{x}_0) \right\|_2 + \frac{1}{2} r^2 \sum_{j=1}^N (2\beta w_j),$$

$$\tag{29}$$

where the term $-\frac{1}{2}r^2 \sum_{j=1}^N (2\beta w_j)$ and $\frac{1}{2}r^2 \sum_{j=1}^N (2\beta w_j)$ are bounded from Lagrange remainder. Note that the difference $f_j^{y_0/y_i}$ is $(2\beta)$-smooth instead of $\beta$-smooth since it is the difference of two $\beta$-smooth function, and thus $\sum_{j=1}^N w_j f_j^{y_0/y_i}$ is $\sum_{j=1}^N (2\beta w_j)$-smooth. From Proposition B.2, the sufficient and necessary condition of WE's $r$-robustness is $\sum_{j=1}^N w_j f_j^{y_0/y_i}(\boldsymbol{x}) \geq 0$ for any $y_i \in [C]$ such that $y_i \neq y_0$, and any $\boldsymbol{x} = \boldsymbol{x}_0 + \boldsymbol{\delta}$ where $\|\boldsymbol{\delta}\|_2 \leq r$. Plugging this term into Equation (29) we get the theorem. $\qquad\square$

**Theorem 3** (Gradient and Confidence Margin Condition for MME Robustness). *Given input $\boldsymbol{x}_0 \in \mathbb{R}^d$ with ground-truth label $y_0 \in [C]$, and $\mathcal{M}_{\mathrm{MME}}$ as an MME defined over base models $\{F_1, F_2\}$. $\mathcal{M}_{\mathrm{MME}}(\boldsymbol{x}_0) = y_0$. Both $F_1$ and $F_2$ are $\beta$-smooth.*

- *(Sufficient Condition) If for any $y_1, y_2 \in [C]$ such that $y_1 \neq y_0$ and $y_2 \neq y_0$,*

$$\|\nabla_{\boldsymbol{x}} f_1^{y_0/y_1}(\boldsymbol{x}_0) + \nabla_{\boldsymbol{x}} f_2^{y_0/y_2}(\boldsymbol{x}_0)\|_2 \leq \frac{1}{r}(f_1^{y_0/y_1}(\boldsymbol{x}_0) + f_2^{y_0/y_2}(\boldsymbol{x}_0)) - 2\beta r, \qquad (12)$$

  *then $\mathcal{M}_{\mathrm{MME}}$ is $r$-robust at point $\boldsymbol{x}_0$.*

- *(Necessary Condition) Suppose for any $\boldsymbol{x} \in \{\boldsymbol{x}_0 + \boldsymbol{\delta} : \|\boldsymbol{\delta}\|_2 \leq r\}$, for any $i \in \{1, 2\}$, either $F_i(\boldsymbol{x}) = y_0$ or $F_i^{(2)}(\boldsymbol{x}) = y_0$. If $\mathcal{M}_{\mathrm{MME}}$ is $r$-robust at point $\boldsymbol{x}_0$, then for any $y_1, y_2 \in [C]$ such that $y_1 \neq y_0$ and $y_2 \neq y_0$,*

$$\|\nabla_{\boldsymbol{x}} f_1^{y_0/y_1}(\boldsymbol{x}_0) + \nabla_{\boldsymbol{x}} f_2^{y_0/y_2}(\boldsymbol{x}_0)\|_2 \leq \frac{1}{r}(f_1^{y_0/y_1}(\boldsymbol{x}_0) + f_2^{y_0/y_2}(\boldsymbol{x}_0)) + 2\beta r. \qquad (13)$$

*Proof of Theorem 3.* We prove the sufficient condition and necessary condition separately.

- (Sufficient Condition)
  From Lemma B.1, since there are only two base models, we can simplify the sufficient condition for $\mathcal{M}_{\mathrm{MME}}(\boldsymbol{x}) = y_0$ as

$$\min_{y_i \in [C]: y_i \neq y_0} f_1^{y_0/y_i}(\boldsymbol{x}) + \min_{y_i' \in [C]: y_i' \neq y_0} f_2^{y_0/y_i'}(\boldsymbol{x}) \geq 0.$$

  In other words, for any $y_1 \neq y_0$ and $y_2 \neq y_0$,

$$f_1^{y_0/y_1}(\boldsymbol{x}) + f_2^{y_0/y_2}(\boldsymbol{x}) \geq 0. \qquad (30)$$

  With Taylor expansion and model-smoothness assumption, we have

$$\min_{\boldsymbol{x}: \|\boldsymbol{x} - \boldsymbol{x}_0\|_2 \leq r} f_1^{y_0/y_1}(\boldsymbol{x}) + f_2^{y_0/y_2}(\boldsymbol{x})$$

$$\geq f_1^{y_0/y_1}(\boldsymbol{x}_0) + f_2^{y_0/y_2}(\boldsymbol{x}_0) - r\|\nabla_{\boldsymbol{x}} f_1^{y_0/y_1}(\boldsymbol{x}_0) + \nabla_{\boldsymbol{x}} f_2^{y_0/y_2}(\boldsymbol{x}_0)\|_2 - \frac{1}{2} \cdot 4\beta r^2.$$

  Plugging this into Equation (30) yields the sufficient condition.

  In the above equation, the term $-\frac{1}{2} \cdot 4\beta r^2$ is bounded from Lagrange remainder. Here, the $4\beta$ term comes from the fact that $f_1^{y_0/y_1}(\boldsymbol{x}) + f_2^{y_0/y_2}(\boldsymbol{x})$ is $(4\beta)$-smooth since it is the sum of difference of $\beta$-smooth function.

- (Necessary Condition)
  From Lemma B.2, similarly, the necessary condition for $\mathcal{M}_{\mathrm{MME}}(\boldsymbol{x}) = y_0$ is simplified to: for any $y_1 \neq y_0$ and $y_2 \neq y_0$,

$$f_1^{y_0/y_1}(\boldsymbol{x}) + f_2^{y_0/y_2}(\boldsymbol{x}) \geq 0. \qquad (30)$$

  Again, from Taylor expansion, we have

$$\min_{\boldsymbol{x}: \|\boldsymbol{x} - \boldsymbol{x}_0\|_2 \leq r} f_1^{y_0/y_1}(\boldsymbol{x}) + f_2^{y_0/y_2}(\boldsymbol{x})$$

$$\leq f_1^{y_0/y_1}(\boldsymbol{x}_0) + f_2^{y_0/y_2}(\boldsymbol{x}_0) - r\|\nabla_{\boldsymbol{x}} f_1^{y_0/y_1}(\boldsymbol{x}_0) + \nabla_{\boldsymbol{x}} f_2^{y_0/y_2}(\boldsymbol{x}_0)\|_2 + \frac{1}{2} \cdot 4\beta r^2.$$

Plugging this into Equation (30) yields the necessary condition.

In the above equation, the term $+\frac{1}{2} \cdot 4\beta r^2$ is bounded from Lagrange remainder. The $4\beta$ term appears because of the same reason as before.

$\square$

Since we will compare the robustness of ensemble models and the single model, we show the corresponding conditions for single-model robustness.

**Proposition B.3** (Gradient and Confidence Margin Conditions for Single-Model Robustness). *Given input $\boldsymbol{x}_0 \in \mathbb{R}^d$ with ground-truth label $y_0 \in [C]$. Model $F(\boldsymbol{x}_0) = y_0$, and it is $\beta$-smooth.*

- *(Sufficient Condition) If for any $y_1 \in [C]$ such that $y_1 \neq y_0$,*

$$\|\nabla_{\boldsymbol{x}} f^{y_0/y_1}(\boldsymbol{x}_0)\|_2 \leq \frac{1}{r} f^{y_0/y_1}(\boldsymbol{x}_0) - \beta r, \tag{31}$$

  *$F$ is $r$-robust at point $\boldsymbol{x}_0$.*

- *(Necessary Condition) If $F$ is $r$-robust at point $\boldsymbol{x}_0$, for any $y_1 \in [C]$ such that $y_1 \neq y_0$,*

$$\|\nabla_{\boldsymbol{x}} f^{y_0/y_1}(\boldsymbol{x}_0)\|_2 \leq \frac{1}{r} f^{y_0/y_1}(\boldsymbol{x}_0) + \beta r. \tag{32}$$

*Proof of Proposition B.3.* This proposition is apparent given the following inequality from Taylor expansion

$$f^{y_0/y_1}(\boldsymbol{x}_0) - r\|\nabla_{\boldsymbol{x}} f^{y_0/y_1}(\boldsymbol{x}_0)\|_2 - \beta r^2 \leq \min_{\boldsymbol{x}: \|\boldsymbol{x}-\boldsymbol{x}_0\|_2 \leq r} f^{y_0/y_1}(\boldsymbol{x}) \leq f^{y_0/y_1}(\boldsymbol{x}_0) - r\|\nabla_{\boldsymbol{x}} f^{y_0/y_1}(\boldsymbol{x}_0)\|_2 + \beta r^2$$

and the sufficient and necessary robust condition in Fact B.1. $\square$

### B.3 PROOF OF ROBUSTNESS COMPARISON RESULTS BETWEEN ENSEMBLE MODELS AND SINGLE MODELS

**Corollary 1** (Comparison of Ensemble and Single-Model Robustness). *Given an input $\boldsymbol{x}_0 \in \mathbb{R}^d$ with ground-truth label $y_0 \in [C]$. Suppose we have two $\beta$-smooth base models $\{F_1, F_2\}$, which are both $r$-robust at point $\boldsymbol{x}_0$. For any $\Delta \in [0, 1)$:*

- *(Weighted Ensemble) Define Weighted Ensemble $\mathcal{M}_{\mathrm{WE}}$ with base models $\{F_1, F_2\}$. Suppose $\mathcal{M}_{\mathrm{WE}}(\boldsymbol{x}_0) = y_0$. If for any label $y_i \neq y_0$, the base models' smoothness $\beta \leq \Delta \cdot \min\{f_1^{y_0/y_i}(\boldsymbol{x}_0), f_2^{y_0/y_i}(\boldsymbol{x}_0)\}/(c^2 r^2)$, and the gradient cosine similarity $\cos\langle \nabla_{\boldsymbol{x}} f_1^{y_0/y_i}(\boldsymbol{x}_0), \nabla_{\boldsymbol{x}} f_2^{y_0/y_i}(\boldsymbol{x}_0)\rangle \leq \cos\theta$, then the $\mathcal{M}_{\mathrm{WE}}$ with weights $\{w_1, w_2\}$ is at least $R$-robust at point $\boldsymbol{x}_0$ with*

$$R = r \cdot \frac{1-\Delta}{1+\Delta} \left(1 - C_{\mathrm{WE}}(1 - \cos\theta)\right)^{-1/2}, where \tag{14}$$

$$C_{\mathrm{WE}} = \min_{y_i : y_i \neq y_0} \frac{2w_1 w_2 f_1^{y_0/y_i}(\boldsymbol{x}_0) f_2^{y_0/y_i}(\boldsymbol{x}_0)}{(w_1 f_1^{y_0/y_i}(\boldsymbol{x}_0) + w_2 f_2^{y_0/y_i}(\boldsymbol{x}_0))^2}, c = \max\{\tfrac{1-\Delta}{1+\Delta}\left(1 - C_{\mathrm{WE}}(1-\cos\theta)\right)^{-1/2}, 1\}.$$

- *(Max-Margin Ensemble) Define Max-Margin Ensemble $\mathcal{M}_{\mathrm{MME}}$ with the base models $\{F_1, F_2\}$. Suppose $\mathcal{M}_{\mathrm{MME}}(\boldsymbol{x}_0) = y_0$. If for any label $y_1 \neq y_0$ and $y_2 \neq y_0$, the base models' smoothness $\beta \leq \Delta \cdot \min\{f_1^{y_0/y_1}(\boldsymbol{x}_0), f_2^{y_0/y_2}(\boldsymbol{x}_0)\}/(c^2 r^2)$, and the gradient cosine similarity $\cos\langle \nabla_{\boldsymbol{x}} f_1^{y_0/y_1}(\boldsymbol{x}_0), \nabla_{\boldsymbol{x}} f_2^{y_0/y_2}(\boldsymbol{x}_0)\rangle \leq \cos\theta$, then the $\mathcal{M}_{\mathrm{MME}}$ is at least $R$-robust at point $\boldsymbol{x}_0$ with*

$$R = r \cdot \frac{1-\Delta}{1+\Delta} \left(1 - C_{\mathrm{MME}}(1 - \cos\theta)\right)^{-1/2}, where \tag{15}$$

$$C_{\mathrm{MME}} = \min_{\substack{y_1, y_2: \\ y_1, y_2 \neq y_0}} \frac{2 f_1^{y_0/y_1}(\boldsymbol{x}_0) f_2^{y_0/y_2}(\boldsymbol{x}_0)}{(f_1^{y_0/y_1}(\boldsymbol{x}_0) + f_2^{y_0/y_2}(\boldsymbol{x}_0))^2}, c = \max\{\tfrac{1-\Delta}{1+\Delta}\left(1 - C_{\mathrm{MME}}(1-\cos\theta)\right)^{-1/2}, 1\}.$$

*Proof of Corollary 1.* We first prove the theorem for Weighted Ensemble. For arbitrary $y_i \neq y_0$, we have

$$\|w_1 \nabla_{\boldsymbol{x}} f_1^{y_0/y_i}(\boldsymbol{x}_0) + w_2 \nabla_{\boldsymbol{x}} f_2^{y_0/y_i}(\boldsymbol{x}_0)\|_2$$

$$= \sqrt{w_1^2 \|\nabla_{\boldsymbol{x}} f_1^{y_0/y_i}(\boldsymbol{x}_0)\|_2^2 + w_2^2 \|\nabla_{\boldsymbol{x}} f_2^{y_0/y_i}(\boldsymbol{x}_0)\|_2^2 + 2 w_1 w_2 \langle \nabla_{\boldsymbol{x}} f_1^{y_0/y_i}(\boldsymbol{x}_0), f_2^{y_0/y_i}(\boldsymbol{x}_0)\rangle}$$

$$\leq \sqrt{w_1^2 \|\nabla_{\boldsymbol{x}} f_1^{y_0/y_i}(\boldsymbol{x}_0)\|_2^2 + w_2^2 \|\nabla_{\boldsymbol{x}} f_2^{y_0/y_i}(\boldsymbol{x}_0)\|_2^2 + 2 w_1 w_2 \|\nabla_{\boldsymbol{x}} f_1^{y_0/y_i}(\boldsymbol{x}_0)\|_2 \|\nabla_{\boldsymbol{x}} f_2^{y_0/y_i}(\boldsymbol{x}_0)\|_2 \cos\theta}$$

$$\overset{(i.)}{\leq} \sqrt{w_1^2 \left(\frac{1}{r} f_1^{y_0/y_i}(\boldsymbol{x}_0) + \beta r\right)^2 + w_2^2 \left(\frac{1}{r} f_2^{y_0/y_i}(\boldsymbol{x}_0) + \beta r\right)^2 + 2 w_1 w_2 \left(\frac{1}{r} f_1^{y_0/y_i}(\boldsymbol{x}_0) + \beta r\right) \left(\frac{1}{r} f_2^{y_0/y_i}(\boldsymbol{x}_0) + \beta r\right) \cos\theta}$$

$$= \frac{1}{r} \sqrt{w_1^2 \left(f_1^{y_0/y_i}(\boldsymbol{x}_0) + \beta r^2\right)^2 + w_2^2 \left(f_2^{y_0/y_i}(\boldsymbol{x}_0) + \beta r^2\right)^2 + 2 w_1 w_2 \left(f_1^{y_0/y_i}(\boldsymbol{x}_0) + \beta r^2\right) \left(f_2^{y_0/y_i}(\boldsymbol{x}_0) + \beta r^2\right) \cos\theta}$$

$$\overset{(ii.)}{\leq} \frac{1}{r} \cdot \left(1 + \frac{\Delta}{c^2}\right) \sqrt{w_1^2 f_1^{y_0/y_i}(\boldsymbol{x}_0)^2 + w_2^2 f_2^{y_0/y_i}(\boldsymbol{x}_0)^2 + 2 w_1 w_2 f_1^{y_0/y_i}(\boldsymbol{x}_0) f_2^{y_0/y_i}(\boldsymbol{x}_0) \cos\theta}$$

$$= \frac{1}{r} \cdot \left(1 + \frac{\Delta}{c^2}\right) \sqrt{\left(w_1 f_1^{y_0/y_i}(\boldsymbol{x}_0) + w_2 f_2^{y_0/y_i}(\boldsymbol{x}_0)\right)^2 - 2(1 - \cos\theta) w_1 f_1^{y_0/y_i}(\boldsymbol{x}_0) w_2 f_2^{y_0/y_i}(\boldsymbol{x}_0)}$$

$$\overset{(iii.)}{\leq} \frac{1}{r} \cdot \left(1 + \frac{\Delta}{c^2}\right) \sqrt{1 - (1 - \cos\theta) C_{\text{WE}}} \left(w_1 f_1^{y_0/y_i}(\boldsymbol{x}_0) + w_2 f_2^{y_0/y_i}(\boldsymbol{x}_0)\right)$$

where $(i.)$ follows from the necessary condition in Proposition B.3; $(ii.)$ uses the condition on $\beta$; and $(iii.)$ replaces $2 w_1 w_2 f_1^{y_0/y_i}(\boldsymbol{x}_0) f_2^{y_0/y_i}(\boldsymbol{x}_0)$ leveraging $C_{\text{WE}}$. Now, we define

$$K := \frac{1 - \Delta}{1 + \Delta} \left(1 - C_{\text{WE}}(1 - \cos\theta)\right)^{-1/2}.$$

All we need to do is to prove that $\mathcal{M}_{\text{WE}}$ is robust within radius $Kr$. To do so, from Equation (4), we upper bound $\|w_1 \nabla_{\boldsymbol{x}} f_1^{y_0/y_i}(\boldsymbol{x}_0) + w_2 \nabla_{\boldsymbol{x}} f_2^{y_0/y_i}(\boldsymbol{x}_0)\|_2$ by $\frac{1}{Kr}\left(w_1 f_1^{y_0/y_i}(\boldsymbol{x}_0) + w_2 f_2^{y_0/y_i}(\boldsymbol{x}_0)\right) - \beta Kr(w_1 + w_2)$:

$$\|w_1 \nabla_{\boldsymbol{x}} f_1^{y_0/y_i}(\boldsymbol{x}_0) + w_2 \nabla_{\boldsymbol{x}} f_2^{y_0/y_i}(\boldsymbol{x}_0)\|_2$$

$$\leq \frac{1}{r} \cdot \left(1 + \frac{\Delta}{c^2}\right) \sqrt{1 - (1 - \cos\theta) C_{\text{WE}}} \left(w_1 f_1^{y_0/y_i}(\boldsymbol{x}_0) + w_2 f_2^{y_0/y_i}(\boldsymbol{x}_0)\right)$$

$$\leq \frac{1}{r}(1 + \Delta) \sqrt{1 - (1 - \cos\theta) C_{\text{WE}}} \left(w_1 f_1^{y_0/y_i}(\boldsymbol{x}_0) + w_2 f_2^{y_0/y_i}(\boldsymbol{x}_0)\right)$$

$$= \frac{1}{r} \cdot \frac{1 - \Delta}{\frac{1-\Delta}{1+\Delta}\left(1 - (1 - \cos\theta) C_{\text{WE}}\right)^{-1/2}} \left(w_1 f_1^{y_0/y_i}(\boldsymbol{x}_0) + w_2 f_2^{y_0/y_i}(\boldsymbol{x}_0)\right)$$

$$= \frac{1}{Kr}(1 - \Delta) \left(w_1 f_1^{y_0/y_i}(\boldsymbol{x}_0) + w_2 f_2^{y_0/y_i}(\boldsymbol{x}_0)\right)$$

$$\leq \frac{1}{Kr} \left(w_1 f_1^{y_0/y_i}(\boldsymbol{x}_0) + w_2 f_2^{y_0/y_i}(\boldsymbol{x}_0) - \Delta \min\{f_1^{y_0/y_i}(\boldsymbol{x}_0), f_2^{y_0/y_i}(\boldsymbol{x}_0)\}(w_1 + w_2)\right).$$

Notice that $\Delta \min\{f_1^{y_0/y_i}(\boldsymbol{x}_0), f_2^{y_0/y_i}(\boldsymbol{x}_0)\} \geq \beta c^2 r^2$ from $\beta$'s condition, so

$$\|w_1 \nabla_{\boldsymbol{x}} f_1^{y_0/y_i}(\boldsymbol{x}_0) + w_2 \nabla_{\boldsymbol{x}} f_2^{y_0/y_i}(\boldsymbol{x}_0)\|_2$$

$$\leq \frac{1}{Kr} \left(w_1 f_1^{y_0/y_i}(\boldsymbol{x}_0) + w_2 f_2^{y_0/y_i}(\boldsymbol{x}_0) - \beta c^2 r^2 (w_1 + w_2)\right)$$

$$= \frac{1}{Kr} \left(w_1 f_1^{y_0/y_i}(\boldsymbol{x}_0) + w_2 f_2^{y_0/y_i}(\boldsymbol{x}_0)\right) - \beta Kr(w_1 + w_2) \cdot \frac{c^2}{K^2}$$

$$\leq \frac{1}{Kr} \left(w_1 f_1^{y_0/y_i}(\boldsymbol{x}_0) + w_2 f_2^{y_0/y_i}(\boldsymbol{x}_0)\right) - \beta Kr(w_1 + w_2).$$

From Equation (4), the theorem for Weighted Ensemble is proved.

Now we prove the theorem for Max-Margin Ensemble. Similarly, for any arbitrary $y_1, y_2$ such that $y_1 \neq y_0, y_2 \neq y_0$, we have

$$\|\nabla_{\boldsymbol{x}} f_1^{y_0/y_1}(\boldsymbol{x}_0) + \nabla_{\boldsymbol{x}} f_2^{y_0/y_2}(\boldsymbol{x}_0)\|_2$$
$$\leq \frac{1}{r} \cdot \left(1 + \frac{\Delta}{c^2}\right) \sqrt{1 - (1 - \cos\theta)C_{\mathrm{MME}}} \left(f_1^{y_0/y_1}(\boldsymbol{x}_0) + f_2^{y_0/y_2}(\boldsymbol{x}_0)\right).$$

Now we define

$$K' := \frac{1 - \Delta}{1 + \Delta} \left(1 - C_{\mathrm{MME}}(1 - \cos\theta)\right)^{-1/2}.$$

Again, from $\beta$'s condition we have $\Delta \min\{f_1^{y_0/y_1}(\boldsymbol{x}_0), f_2^{y_0/y_2}(\boldsymbol{x}_0)\} \geq \beta c^2 r^2$ and

$$\|\nabla_{\boldsymbol{x}} f_1^{y_0/y_1}(\boldsymbol{x}_0) + \nabla_{\boldsymbol{x}} f_2^{y_0/y_2}(\boldsymbol{x}_0)\|_2 \leq \frac{1}{K'r} \left(f_1^{y_0/y_i}(\boldsymbol{x}_0) + f_2^{y_0/y_i}(\boldsymbol{x}_0)\right) - 2\beta K'r.$$

From Equation (12), the ensemble is $(K'r)$-robust at point $\boldsymbol{x}_0$, i.e., the theorem for Max-Margin Ensemble is proved. $\square$

### B.4 PROOFS OF MODEL-SMOOTHNESS BOUNDS FOR RANDOMIZED SMOOTHING

**Theorem 2** (Model-Smoothness Upper Bound for $\bar{g}_f^\varepsilon$). *Let $\varepsilon \sim \mathcal{N}(0, \sigma^2 \boldsymbol{I}_d)$ be a Gaussian random variable, then the soft smoothed confidence function $\bar{g}_f^\varepsilon$ is $(2/\sigma^2)$-smooth.*

*Proof of Theorem 2.* Recall that $\bar{g}_f^\varepsilon(\boldsymbol{x})_j = \mathbb{E}_{\varepsilon \sim \mathcal{N}(0,\sigma^2 \boldsymbol{I}_d)} f(\boldsymbol{x} + \varepsilon)_j$, where $f(\boldsymbol{x} + \varepsilon)_j$ is a function from $\mathbb{R}^d$ to $\{0, 1\}$. Therefore, to prove that $g_{\mathcal{M}}^\varepsilon$ is $(2/\sigma^2)$-smooth, we only need to show that for any function $f : \mathbb{R}^d \to [0, 1]$, the function $\bar{f} := f * \mathcal{N}(0, \sigma^2 \boldsymbol{I}_d)$ is $(2/\sigma^2)$-smooth.

According to (Salman et al., 2019, Lemma 1), we have

$$\bar{f}(\boldsymbol{x}) = \frac{1}{(2\pi\sigma^2)^{d/2}} \int_{\mathbb{R}^d} f(\boldsymbol{t}) \exp\left(-\frac{\|\boldsymbol{x} - \boldsymbol{t}\|_2^2}{2\sigma^2}\right) \mathrm{d}\boldsymbol{t}, \tag{33}$$

$$\nabla \bar{f}(\boldsymbol{x}) = \frac{1}{(2\pi\sigma^2)^{d/2}\sigma^2} \int_{\mathbb{R}^d} f(\boldsymbol{t})(\boldsymbol{x} - \boldsymbol{t}) \exp\left(-\frac{\|\boldsymbol{x} - \boldsymbol{t}\|_2^2}{2\sigma^2}\right) \mathrm{d}\boldsymbol{t}. \tag{34}$$

To show $\bar{f}$ is $(2/\sigma^2)$-smooth, we only need to show that $\nabla\bar{f}$ is $(2/\sigma^2)$-Lipschitz. Let $\boldsymbol{H}_{\bar{f}}(\boldsymbol{x})$ be the Hessian matrix of $\bar{f}$. Thus, we only need to show that for any unit vector $\boldsymbol{u}$, $|\boldsymbol{u}^\intercal \boldsymbol{H}_{\bar{f}}(\boldsymbol{x})\boldsymbol{u}| \leq 2/\sigma^2$. By the isotropy of $\boldsymbol{H}_{\bar{f}}(\boldsymbol{x})$, it is sufficient to consider $\boldsymbol{u} = (1, 0, 0, \ldots, 0)^\intercal$, where $\boldsymbol{u}^\intercal \boldsymbol{H}_{\bar{f}}(\boldsymbol{x})\boldsymbol{u} = \boldsymbol{H}_{\bar{f}}(\boldsymbol{x})_{11}$. Now we only need to bound the absolute value of $\boldsymbol{H}_{\bar{f}}(\boldsymbol{x})_{11}$:

$$|\boldsymbol{H}_{\bar{f}}(\boldsymbol{x})_{11}| = \left|\frac{1}{(2\pi\sigma^2)^{d/2}\sigma^2} \int_{\mathbb{R}^d} f(\boldsymbol{t}) \cdot \frac{\partial}{\partial \boldsymbol{x}_1}(\boldsymbol{x} - \boldsymbol{t}) \exp\left(-\frac{\|\boldsymbol{x} - \boldsymbol{t}\|_2^2}{2\sigma^2}\right) \mathrm{d}\boldsymbol{t}\right|$$

$$= \frac{1}{(2\pi\sigma^2)^{d/2}\sigma^2} \left|\int_{\mathbb{R}^d} f(\boldsymbol{t}) \cdot \left(1 - \frac{(\boldsymbol{x}_1 - \boldsymbol{t}_1)^2}{\sigma^2}\right) \exp\left(-\frac{\|\boldsymbol{x} - \boldsymbol{t}\|_2^2}{2\sigma^2}\right) \mathrm{d}\boldsymbol{t}\right|$$

$$\leq \frac{1}{(2\pi\sigma^2)^{d/2}\sigma^2} \left|\int_{\mathbb{R}^d} \exp\left(-\frac{\|\boldsymbol{x} - \boldsymbol{t}\|_2^2}{2\sigma^2}\right) \mathrm{d}\boldsymbol{t}\right|$$

$$+ \frac{1}{(2\pi\sigma^2)^{d/2}\sigma^2} \left|\int_{\mathbb{R}^d} \frac{(\boldsymbol{x}_1 - \boldsymbol{t}_1)^2}{\sigma^2} \exp\left(-\frac{\|\boldsymbol{x} - \boldsymbol{t}\|_2^2}{2\sigma^2}\right) \mathrm{d}\boldsymbol{t}\right|$$

$$= \frac{1}{\sigma^2} + \frac{1}{\sqrt{2\pi\sigma^2}\sigma^2} \cdot 2 \int_0^\infty \frac{x^2}{\sigma^2} \exp\left(-\frac{x^2}{2\sigma^2}\right) \mathrm{d}x$$

$$= \frac{1}{\sigma^2} + \sqrt{\frac{2}{\pi}} \cdot \frac{1}{\sigma^2} \int_0^\infty t^2 \exp(-t^2/2) \mathrm{d}t. \tag{35}$$

Let $\Gamma(\cdot)$ be the Gamma function, we note that

$$\int_0^\infty t^2 \exp(-t^2/2) \mathrm{d}t = \int_0^\infty t \exp(-t^2/2) \mathrm{d}(-t^2/2) = \sqrt{2} \int_0^\infty \sqrt{t} \exp(-t) \mathrm{d}t = \sqrt{2}\Gamma(3/2) = \sqrt{\pi/2},$$

and thus

$$|\boldsymbol{H}_{\bar{f}}(\boldsymbol{x})_{11}| \leq \frac{1}{\sigma^2} + \sqrt{\frac{2}{\pi}} \cdot \frac{1}{\sigma^2} \cdot \sqrt{\frac{\pi}{2}} = \frac{2}{\sigma^2}, \tag{36}$$

which concludes the proof. □

*Remark.* The model-smoothness upper bound Theorem 2 is not limited to the ensemble model with *Ensemble-before-Smoothing* strategy. Indeed, for arbitrary classification models, since the confidence score is in range $[0, 1]$, the theorem still holds. If the confidence score is bounded in $[a, b]$, simple scaling yields the model-smoothness upper bound $\beta = \frac{2(b-a)}{\sigma^2}$.

**Proposition B.4** (Model-Smoothness Lower Bound for $\bar{g}_f^\varepsilon$)**.** *There exists a smoothed confidence function $\bar{g}_f^\varepsilon$ that is $\beta$-smooth if and only if $\beta \geq \left( \frac{1}{\sqrt{2\pi e}\sigma^2} \right)$.*

*Proof of Proposition B.4.* We prove by construction. Consider the single dimensional input space $\mathbb{R}$, and a model $f$ that has confidence 1 if and only if input $x \geq 0$. As a result,

$$g_f^\varepsilon(x)_{y_0} = \frac{1}{\sqrt{2\pi}\sigma} \int_0^{+\infty} \exp\left( -\frac{(t-x)^2}{2\sigma^2} \right) \mathrm{d}t = \frac{1}{\sqrt{2\pi}\sigma} \int_{-x}^{+\infty} \exp\left( -\frac{t^2}{2\sigma^2} \right) \mathrm{d}t.$$

Thus,

$$\frac{\mathrm{d}g_f^\varepsilon(x)_{y_0}}{\mathrm{d}x} = \frac{1}{\sqrt{2\pi}\sigma} \exp\left( -\frac{x^2}{2\sigma^2} \right) \quad \text{and} \quad \left| \frac{\mathrm{d}g_f^\varepsilon(x)_{y_0}^2}{\mathrm{d}^2 x} \right| = \frac{1}{\sqrt{2\pi}\sigma^2} \cdot \left| \frac{x}{\sigma} \right| \exp\left( -\frac{x^2}{2\sigma^2} \right).$$

By symmetry, we study the function $h(x) = x\exp(-x^2/2)$ for $x \geq 0$. We have $h'(x) = (1 - x)\exp(-x^2/2)$. Thus, $h(x)$ obtains its maximum at $x_0 = 1$: $h(x_0) = \exp(-1/2)$, which implies that

$$\max\left| \frac{\mathrm{d}g_f^\varepsilon(x)_{y_0}^2}{\mathrm{d}^2 x} \right| = \frac{\exp(-1/2)}{\sqrt{2\pi}\sigma^2} = \frac{1}{\sqrt{2\pi e}\sigma^2}$$

which implies $\beta \geq \frac{1}{\sqrt{2\pi e}\sigma^2}$ for this $\bar{g}_f^\varepsilon$ per smoothness definition (Definition 4). □

## B.5 Proofs of Robustness Conditions for Smoothed Ensemble Models

**Corollary 2** (Gradient and Confidence Margin Conditions for Smoothed WE Robustness)**.** *Given input $\boldsymbol{x}_0 \in \mathbb{R}^d$ with ground-truth label $y_0 \in [C]$. Let $\varepsilon \sim \mathcal{N}(0, \sigma^2 \boldsymbol{I}_d)$ be a Gaussian random variable. Define soft smoothed confidence $\bar{g}_i^\varepsilon(\boldsymbol{x}) := \mathbb{E}_\varepsilon f_i(\boldsymbol{x} + \varepsilon)$ for each base model $F_i$ ($1 \leq i \leq N$). The $\bar{G}_{\mathcal{M}_{\mathrm{WE}}}^\varepsilon$ is a WE defined over soft smoothed base models $\{\bar{g}_i^\varepsilon\}_{i=1}^N$ with weights $\{w_i\}_{i=1}^N$. $\bar{G}_{\mathcal{M}_{\mathrm{WE}}}^\varepsilon(\boldsymbol{x}_0) = y_0$.*

- *(Sufficient Condition) The $\bar{G}_{\mathcal{M}_{\mathrm{WE}}}^\varepsilon$ is $r$-robust at point $\boldsymbol{x}_0$ if for any $y_i \neq y_0$,*

$$\left\| \sum_{j=1}^N w_j \nabla_{\boldsymbol{x}} (\bar{g}_j^\varepsilon)^{y_0/y_i}(\boldsymbol{x}_0) \right\|_2 \leq \frac{1}{r} \sum_{j=1}^N w_j (\bar{g}_j^\varepsilon)^{y_0/y_i}(\boldsymbol{x}_0) - \frac{2r}{\sigma^2} \sum_{j=1}^N w_j, \tag{21}$$

- *(Necessary Condition) If $\bar{G}_{\mathcal{M}_{\mathrm{WE}}}^\varepsilon$ is $r$-robust at point $\boldsymbol{x}_0$, for any $y_i \neq y_0$,*

$$\left\| \sum_{j=1}^N w_j \nabla_{\boldsymbol{x}} (\bar{g}_j^\varepsilon)^{y_0/y_i}(\boldsymbol{x}_0) \right\|_2 \leq \frac{1}{r} \sum_{j=1}^N w_j (\bar{g}_j^\varepsilon)^{y_0/y_i}(\boldsymbol{x}_0) + \frac{2r}{\sigma^2} \sum_{j=1}^N w_j. \tag{22}$$

*Proof of Corollary 2.* Since $\bar{G}_{\mathcal{M}_{\mathrm{WE}}}^\varepsilon$ is a WE defined over $\{\bar{g}_i^\varepsilon\}_{i=1}^N$, we apply Theorem 1 directly for $\bar{G}_{\mathcal{M}_{\mathrm{WE}}}^\varepsilon$. Notice that each $\bar{g}_i^\varepsilon$ has model-smoothness bound $\beta = 2/\sigma^2$ from Theorem 2 and the corollary statement follows. □

**Corollary 3** (Gradient and Confidence Margin Condition for Smoothed MME Robustness)**.** *Given input $\boldsymbol{x}_0 \in \mathbb{R}^d$ with ground-truth label $y_0 \in [C]$. Let $\varepsilon \sim \mathcal{N}(0, \sigma^2 \boldsymbol{I}_d)$ be a Gaussian random variable. Define soft smoothed confidence $\bar{g}_i^\varepsilon(\boldsymbol{x}) := \mathbb{E}_\varepsilon f_i(\boldsymbol{x} + \varepsilon)$ for either base model $F_1$ or $F_2$. The $\bar{G}_{\mathcal{M}_{\mathrm{MME}}}^\varepsilon$ is a MME defined over soft smoothed base models $\{\bar{g}_1^\varepsilon, \bar{g}_2^\varepsilon\}$. $\bar{G}_{\mathcal{M}_{\mathrm{MME}}}^\varepsilon(\boldsymbol{x}_0) = y_0$.*

- *(Sufficient Condition) If for any $y_1, y_2 \in [C]$ such that $y_1 \neq y_0$ and $y_2 \neq y_0$,*

$$\|\nabla_{\boldsymbol{x}}(\bar{g}_1^\varepsilon)^{y_0/y_1}(\boldsymbol{x}_0) + \nabla_{\boldsymbol{x}}(\bar{g}_2^\varepsilon)^{y_0/y_2}(\boldsymbol{x}_0)\|_2 \leq \frac{1}{r}((\bar{g}_1^\varepsilon)^{y_0/y_1}(\boldsymbol{x}_0) + (\bar{g}_2^\varepsilon)^{y_0/y_2}(\boldsymbol{x}_0)) - \frac{4r}{\sigma^2}, \tag{23}$$

  *then $\bar{G}^\varepsilon_{\mathcal{M}_{\mathrm{MME}}}$ is $r$-robust at point $\boldsymbol{x}_0$.*

- *(Necessary Condition) Suppose for any $\boldsymbol{x} \in \{\boldsymbol{x}_0 + \boldsymbol{\delta} : \|\boldsymbol{\delta}\|_2 \leq r\}$, for any $i \in \{1, 2\}$, either $G_{F_i}(\boldsymbol{x}) = y_0$ or $G_{F_i}^{(2)}(\boldsymbol{x}) = y_0$. If $\bar{G}^\varepsilon_{\mathcal{M}_{\mathrm{MME}}}$ is $r$-robust at point $\boldsymbol{x}_0$, then for any $y_1, y_2 \in [C]$ such that $y_1 \neq y_0$ and $y_2 \neq y_0$,*

$$\|\nabla_{\boldsymbol{x}}(\bar{g}_1^\varepsilon)^{y_0/y_1}(\boldsymbol{x}_0) + \nabla_{\boldsymbol{x}}(\bar{g}_2^\varepsilon)^{y_0/y_2}(\boldsymbol{x}_0)\|_2 \leq \frac{1}{r}((\bar{g}_1^\varepsilon)^{y_0/y_1}(\boldsymbol{x}_0) + (\bar{g}_2^\varepsilon)^{y_0/y_2}(\boldsymbol{x}_0)) + \frac{4r}{\sigma^2}. \tag{24}$$

*Proof of Corollary 3.* Since $\bar{G}^\varepsilon_{\mathcal{M}_{\mathrm{MME}}}$ is constructed over confidences $\bar{g}_1^\varepsilon$ and $\bar{g}_2^\varepsilon$, we can directly apply Theorem 1. Again, with the model-smoothness bound $\beta = 2/\sigma^2$ we can easily derive the corollary statement. $\qquad\square$

## C ANALYSIS OF ENSEMBLE SMOOTHING STRATEGIES

In main text we mainly use the adapted randomized model smoothing strategy which is named *Ensemble-before-Smoothing* (EBS). We also consider *Ensemble-after-Smoothing* (*Ensemble-after-Smoothing*). Through the following analysis, we will show *Ensemble-before-Smoothing* generally provides higher certified robust radius than *Ensemble-after-Smoothing* which justifies our choice of the strategy.

The *Ensemble-before-Smoothing* strategy is defined in Definition 5. The *Ensemble-after-Smoothing* strategy is defined as such.

**Definition 6** (*Ensemble-after-Smoothing* (**EAS**)). Let $\mathcal{M}$ be an ensemble model over base models $\{F_i\}_{i=1}^N$. Let $\varepsilon$ be a random variable. The EAS ensemble $H^\varepsilon_{\mathcal{M}} : \mathbb{R}^d \mapsto [C]$ at input $\boldsymbol{x}_0 \in \mathbb{R}^d$ is defined as:

$$H^\varepsilon_{\mathcal{M}}(\boldsymbol{x}_0) := G^\varepsilon_{F_c}(\boldsymbol{x}_0) \quad \text{where} \quad c = \arg\max_{i \in [N]} g^\varepsilon_{F_i}(\boldsymbol{x}_0)_{G^\varepsilon_{F_i}(\boldsymbol{x}_0)}. \tag{37}$$

Here, $c$ is the index of the smoothed base model selected.

*Remark.* In EBS, we first construct a model ensemble $\mathcal{M}$ based on base models using WE or MME protocol, then apply randomized smoothing on top of the ensemble. The resulting smoothed ensemble predicts the most frequent class of $\mathcal{M}$ when the input follows distribution $\boldsymbol{x}_0 + \varepsilon$.

In EAS, we use $\varepsilon$ to construct smoothed classifiers for base models respectively. Then, for given input $\boldsymbol{x}_0$, the ensemble agrees on the base model which has the highest probability for its predicted class.

### C.1 CERTIFIED ROBUSTNESS

In this subsection, we characterize the certified robustness when using both strategies.

#### C.1.1 *Ensemble-before-Smoothing*

The following theorem gives an explicit method (first compute $g^\varepsilon_{\mathcal{M}}(\boldsymbol{x}_0)_{G^\varepsilon_{\mathcal{M}}(\boldsymbol{x}_0)}$ via sampling then compute $r$) to compute the certified robust radius $r$ for EBS protocol. This method is used for computing the certified robust radius in our paper. All other baselines appeared in our paper also use this method.

**Proposition C.1** (Certified Robustness for *Ensemble-before-Smoothing*). *Let $G^\varepsilon_{\mathcal{M}}$ be an ensemble constructed by EBS strategy. The random variable $\varepsilon \sim \mathcal{N}(0, \sigma^2 \boldsymbol{I}_d)$. Then the ensemble $G^\varepsilon_{\mathcal{M}}$ is $r$-robust at point $\boldsymbol{x}_0$ where*

$$r := \sigma \Phi^{-1}\left(g^\varepsilon_{\mathcal{M}}(\boldsymbol{x}_0)_{G^\varepsilon_{\mathcal{M}}(\boldsymbol{x}_0)}\right). \tag{38}$$

*Here, $g^\varepsilon_{\mathcal{M}}(\boldsymbol{x}_0)_j = \Pr_\epsilon(\mathcal{M}(\boldsymbol{x}_0 + \varepsilon) = j)$.*

The proposition is a direct application of Lemma A.1.

### C.1.2 *Ensemble-after-Smoothing*

The following theorem gives an explicit method to compute the certified robust radius $r$ for EAS protocol.

**Theorem 5** (Certified robustness for *Ensemble-after-Smoothing*)**.** *Let* $H_{\mathcal{M}}^{\varepsilon}$ *be an ensemble constructed by EAS strategy over base models* $\{F_i\}_{i=1}^{N}$. *The random variable* $\epsilon \sim \mathcal{N}(0, \sigma^2 \boldsymbol{I}_d)$. *Let* $y_0 = H_{\mathcal{M}}^{\varepsilon}(\boldsymbol{x}_0)$. *For each* $i \in [N]$, *define*

$$
r_i := \begin{cases} \sigma \Phi^{-1} \left( g_{F_i}^{\varepsilon}(\boldsymbol{x}_0)_{G_{F_i}^{\varepsilon}(\boldsymbol{x}_0)} \right), & \text{if } G_{F_i}^{\varepsilon}(\boldsymbol{x}_0) = y_0 \\[2mm] -\sigma \Phi^{-1} \left( g_{F_i}^{\varepsilon}(\boldsymbol{x}_0)_{G_{F_i}^{\varepsilon}(\boldsymbol{x}_0)} \right). & \text{if } G_{F_i}^{\varepsilon}(\boldsymbol{x}_0) \neq y_0 \end{cases}
$$

*Then the ensemble* $H_{\mathcal{M}}^{\varepsilon}$ *is* $r$-*robust at point* $\boldsymbol{x}_0$ *where*

$$
r := \frac{\max_{i \in [N]} r_i + \min_{i \in [N]} r_i}{2}. \tag{39}
$$

*Remark.* The theorem appears to be a bit counter-intuitive — picking the best smoothed model in terms of certified robustness cannot give strong certified robustness for the ensemble. As long as the base models have different certified robust radius (i.e., $r_i$'s are different), the $r$, certified robust radius for the ensemble, is strictly inferior to that of the best base model (i.e., $\max r_i$). Furthermore, if there exists a base model with wrong prediction (i.e., $r_i \leq 0$), the certified robust radius $r$ is strictly smaller than *half* of the best base model.

*Proof of Theorem 5.* Without loss of generality, we assume $r_1 > r_2 > \cdots > r_N$. Let the perturbation added to $\boldsymbol{x}_0$ has $L_2$ length $\delta$.

When $\delta \leq r_N$, since picking any model always gives the right prediction, the ensemble is robust.

When $r_N < \delta \leq \frac{r_1 + r_N}{2}$, the highest robust radius with wrong prediction is $\delta - r_N$, and we can still guarantee that model $F_1$ has robust radius at least $r_1 - \delta$ from the smoothness of function $\boldsymbol{x} \mapsto g_{F_1}^{\varepsilon}(\boldsymbol{x})_{G_{F_1}^{\varepsilon}(\boldsymbol{x}_0)}$ (Salman et al., 2019). Since $r_1 - \delta \geq \frac{r_1 - r_N}{2} \geq \delta - r_N$, the ensemble will agree on $F_1$ or other base model with correct prediction and still gives the right prediction.

When $\delta > \frac{r_1 + r_N}{2}$, suppose $f_N$ is a linear model and only predicts two labels (which achieves the tight robust radius bound according to Cohen et al. (2019)), then $f_N$ can have robust radius $\delta - r_N$ for the wrong prediction. At the same time, for any other model $F_i$ which is linear and predicts correctly, the robust radius is at most $r_i - \delta$. Since $r_i - \delta < r_1 - \delta < \frac{r_1 - r_N}{2} < \delta - r_N$, the ensemble can probably give wrong prediction.

In summary, as we have shown, the certified robust radius can be at most $r$. For any radius $\delta > r$, there exist base models which lead the ensemble $H_{\mathcal{M}}^{\varepsilon}(\boldsymbol{x}_0 + \delta \boldsymbol{e})$ to predict the label other than $y_0$. $\square$

### C.2 COMPARISON OF TWO STRATEGIES

In this subsection, we compare the two ensemble strategies when the ensembles are constructed from two base models.

**Corollary 4** (Smoothing Strategy Comparison)**.** *Given* $\mathcal{M}_{\text{MME}}$, *a Max-Margin Ensemble constructed from base models* $\{f_a, f_b\}$. *Let* $\varepsilon \sim \mathcal{N}(0, \sigma^2 \boldsymbol{I}_d)$. *Let* $G_{\mathcal{M}_{\text{MME}}}^{\varepsilon}$ *be the EBS ensemble, and* $H_{\mathcal{M}_{\text{MME}}}^{\varepsilon}$ *be the EAS ensemble. Suppose at point* $\boldsymbol{x}_0$ *with ground-truth label* $y_0$, $G_{F_a}^{\varepsilon}(\boldsymbol{x}_0) = G_{F_b}^{\varepsilon}(\boldsymbol{x}_0) = y_0$, $g_{F_a}^{\varepsilon}(\boldsymbol{x}_0) > 0.5$, $g_{F_b}^{\varepsilon}(\boldsymbol{x}_0) > 0.5$.

*Let* $\delta$ *be their probability difference for class* $y_0$, *i.e,* $\delta := |g_{F_a}^{\varepsilon}(\boldsymbol{x}_0)_{y_0} - g_{F_b}^{\varepsilon}(\boldsymbol{x}_0)_{y_0}|$,*. Let* $p_{\min}$ *be the smaller probability for class* $y_0$ *between them, i.e.,* $p_{\min} := \min\{g_{F_a}^{\varepsilon}(\boldsymbol{x}_0)_{y_0}, g_{F_b}^{\varepsilon}(\boldsymbol{x}_0)_{y_0}\}$. *We denote* $p$ *to the probability of choosing the correct class when the base models disagree with each other; denote* $p_{ab}$ *to the probability of both base models agreeing on the correct class:*

$$
p := \Pr_{\varepsilon} \left( \mathcal{M}_{\text{MME}}(\boldsymbol{x}_0 + \varepsilon) = y_0 \mid F_a(\boldsymbol{x}_0 + \varepsilon) \neq F_b(\boldsymbol{x}_0 + \varepsilon) \text{ and } (F_a(\boldsymbol{x}_0 + \varepsilon) = y_0 \text{ or } F_b(\boldsymbol{x}_0 + \varepsilon) = y_0) \right),
$$

$$
p_{ab} := \Pr_{\varepsilon} \left( F_a(\boldsymbol{x}_0 + \varepsilon) = F_b(\boldsymbol{x}_0 + \varepsilon) = y_0 \right).
$$

*We have:*

1. *If $p > 1/2 + (2 + 4(p_{\min} - p_{ab})/\delta)^{-1}$, $r_G > r_H$.*

2. *If $p \leq 1/2$, $r_H \geq r_G$.*

*Here, $r_G$ is the certified robust radius of $G^\varepsilon_{\mathcal{M}_{\mathrm{MME}}}$ computed from Equation (38); and $r_H$ is the certified robust radius of $H^\varepsilon_{\mathcal{M}_{\mathrm{MME}}}$ computed from Equation (39).*

*Remark.* Since $p$ is the probability where the ensemble chooses the correct prediction between two base model predictions, with Max-Margin Ensemble, we think $p > 1/2$ with non-trivial margin.

The quantity $p_{\min} - p_{ab}$ and $\delta$ both measure the base model's diversity in terms of predicted label distribution, and generally they should be close. As a result, $1/2 + (2 + 4(p_{\min} - p_{ab})/\delta)^{-1} \approx 1/2 + 1/6 = 2/3$, and case (1) should be much more likely to happen than case (2). Therefore, *EBS usually yields higher robustness guarantee.* We remark that the similar tendency also holds with multiple base models.

*Proof of Corollary 4.* For convenience, define $p_a := g^\varepsilon_{F_a}(\boldsymbol{x}_0)_{y_0}, p_b := g^\varepsilon_{F_b}(\boldsymbol{x}_0)_{y_0}$, where $p_a = p_b + \delta$ and $p_{\min} = p_b$.

From Proposition C.1 and Theorem 5, we have

$$r_G := \frac{\sigma}{2} \cdot 2\Phi^{-1}\left(\Pr_\epsilon(\mathcal{M}_{\mathrm{MME}}(\boldsymbol{x}_0 + \epsilon) = y_0)\right), \quad r_H := \frac{\sigma}{2}\left(\Phi^{-1}(p_a) + \Phi^{-1}(p_b)\right).$$

Notice that $\Pr_\epsilon(\mathcal{M}_{\mathrm{MME}}(\boldsymbol{x}_0 + \epsilon) = y_0) = p_{ab} + p(p_a + p_b - 2p_{ab})$, we can rewrite $r_G$ as

$$r_G = \frac{\sigma}{2} \cdot 2\Phi^{-1}(p_{ab} + p(p_a + p_b - 2p_{ab})).$$

1. When $p > 1/2 + (2 + 4(p_{\min} - p_{ab})/\delta)^{-1}$,
   since

$$p > \frac{1}{2} + \frac{1}{2 + \frac{4(p_{\min} - p_{ab})}{\delta}} = \frac{1}{2} + \frac{\delta}{2\delta + 4(p_b - p_{ab})} = \frac{p_a + p_b + \delta - 2p_{ab}}{2(p_a + p_b - 2p_{ab})} = \frac{p_a - p_{ab}}{p_a + p_b - 2p_{ab}},$$

   we have $p_{ab} + p(p_a + p_b - 2p_{ab}) > p_a$. Therefore, $r_G > \sigma\Phi^{-1}(p_a)$. Whereas, $r_H \leq \sigma/2 \cdot 2\Phi^{-1}(p_a) = \sigma\Phi^{-1}(p_a)$. So $r_G > r_H$.

2. When $p \leq 1/2$,

$$p_{ab} + p(p_a + p_b - 2p_{ab}) \leq p_{ab} + 1/2 \cdot (p_a + p_b - 2p_{ab}) = (p_a + p_b)/2.$$

Therefore, $r_G \leq \sigma\Phi^{-1}((p_a + p_b)/2)$. Notice that $\Phi^{-1}$ is convex in $[1/2, +\infty)$, so $\Phi^{-1}(p_a) + \Phi^{-1}(p_b) \geq 2\Phi^{-1}((p_a + p_b)/2)$, i.e., $r_H \geq r_G$.

$\square$

# D ROBUSTNESS FOR SMOOTHED ML ENSEMBLE: STATISTICAL ROBUSTNESS PERSPECTIVE

In this appendix, we study the robustness of ensemble models from the statistical robustness perspective. This perspective is motivated from Lemma A.1, where the certified robust radius of a model smoothed with Gaussian distribution $\varepsilon \sim \mathcal{N}(0, \sigma^2 \boldsymbol{I}_d)$ is directly proportional to the probability of the original (unsmoothed) model predicting the correct class under such noise.

We first define the notation of statistical robustness in Appendix D.1; then we show and prove the certified robustness guarantees of WE, MME, and single models respectively in Appendix D.2; next we use these results to compare these ensembles under both general assumptions (Appendix D.3) and more specific uniform distribution assumptions (Appendix D.4) where several findings are also discussed; finally, we conduct extensive numerical experiments to verify all these findings in Appendix D.5.

## D.1 DEFINITIONS OF STATISTICAL ROBUSTNESS

**Definition 7** (($\varepsilon$, $p$)-Statistical Robust). Given a random variable $\varepsilon$ and model $F : \mathbb{R}^d \mapsto [C]$, at point $\boldsymbol{x}_0$ with ground truth label $y_0$, we call $F$ is ($\varepsilon$, $p$)-statistical robust if $\Pr_\varepsilon(F(\boldsymbol{x}_0 + \varepsilon) = y_0) \geq p$.

*Remark.* Note that based on Lemma A.1, when $\varepsilon \sim \mathcal{N}(0, \sigma^2 \boldsymbol{I}_d)$, if $F$ is ($\varepsilon$, $p$)-statistical robust at point $\boldsymbol{x}_0$, the smoothed model $G_F^\varepsilon$ over $F$ is $(\sigma \Phi^{-1}(p))$-robust at point $\boldsymbol{x}_0$.

The following three definitions are used in the theorem statements in the following subsections. They can be viewed as the "confidence margins" under noised inputs $\boldsymbol{x}_0 + \varepsilon$ for single model and ensemble respectively.

**Definition 8** (($\varepsilon$, $\lambda$, $p$)-Single Confident). Given a classification model $F$. If at point $\boldsymbol{x}_0$ with ground-truth label $y_0$ and the random variable $\varepsilon$, we have

$$\Pr_\varepsilon \left( \max_{y_j \in [C]: y_j \neq y_0} f(\boldsymbol{x}_0 + \varepsilon)_{y_j} \leq \lambda(1 - f(\boldsymbol{x}_0 + \varepsilon)_{y_0}) \right) = 1 - p,$$

we call $F$ ($\varepsilon$, $\lambda$, $p$)-single confident at point $\boldsymbol{x}_0$.

**Definition 9** (($\varepsilon$, $\lambda$, $p$)-WE Confident). Let $\mathcal{M}_{\mathrm{WE}}$ be a weighted ensemble defined over base models $\{F_i\}_{i=1}^N$ with weights $\{w_i\}_{i=1}^N$. If at point $\boldsymbol{x}_0$ with ground-truth $y_0$ and random variable $\varepsilon$, we have

$$\Pr_\varepsilon \left( \max_{y_j \in [C]: y_j \neq y_0} \left( \sum_{i=1}^N w_i f_i(\boldsymbol{x}_0 + \boldsymbol{\epsilon})_{y_j} \right) \leq \lambda \sum_{i=1}^N w_i \left( 1 - f_i(\boldsymbol{x}_0 + \boldsymbol{\epsilon})_{y_0} \right) \right) = 1 - p, \qquad (40)$$

we call weighted ensemble $\mathcal{M}_{\mathrm{WE}}$ ($\varepsilon$, $\lambda$, $p$)-WE confident at point $\boldsymbol{x}_0$.

**Definition 10** (($\varepsilon$, $\lambda$, $p$)-MME Confident). Let $\mathcal{M}_{\mathrm{MME}}$ be a max-margin ensemble over $\{F_i\}_{i=1}^N$. If at point $\boldsymbol{x}_0$ with ground-truth $y_0$ and random variable $\varepsilon$, we have

$$\Pr_\varepsilon \left( \bigwedge_{i \in [N]} \left( \max_{y_j \in [C]: y_j \neq y_0} f_i(\boldsymbol{x}_0 + \varepsilon)_{y_j} \leq \lambda(1 - f_i(\boldsymbol{x}_0 + \varepsilon)_{y_0}) \right) \right) = 1 - p, \qquad (41)$$

we call max-margin ensemble $\mathcal{M}_{\mathrm{MME}}$ ($\varepsilon$, $\lambda$, $p$)-MME confident at point $\boldsymbol{x}_0$.

Note that the confidence of every single model lies in the probability simplex, and $\lambda$ reflects the confidence portion that a wrong prediction class can take beyond the true class $(1 - f_i(\boldsymbol{x}_0 + \varepsilon))$.

To reduce ambiguity, we usualy use $\lambda_1$ in WE Confident, $\lambda_2$ in MME Confident, and $\lambda_3$ in Single Confident. Note that given $\lambda_1$ is the weighted average and $\lambda_2$ the maximum over $\lambda$'s of all base models, under the same $p$, $\lambda_1/\lambda_2 \leq 1$. Furthermore, *under the same $p$, $\lambda_1/\lambda_2$ reflects the adversarial transferability (Papernot et al., 2016a) among base models*: If the transferability is high, the confidence scores of base models are similar ($\lambda$'s are similar), and thus $\lambda_1$ is large resulting in large $\lambda_1/\lambda_2$. On the other hand, when the transferability is low, the confidence scores are diverse ($\lambda$'s are diverse), and thus $\lambda_1$ is small resulting in small $\lambda_1/\lambda_2$.

The following lemma is frequently used in our following proofs:

**Lemma D.1.** *Suppose the random variable $X$ satisfies $\mathbb{E}X > 0$, $\mathrm{Var}(X) < \infty$ and for any $x \in \mathbb{R}_+$, $\Pr(X \geq \mathbb{E}X + x) = \Pr(X \leq \mathbb{E}X - x)$, then*

$$\Pr(X \leq 0) \leq \frac{\mathrm{Var}(X)}{2(\mathbb{E}X)^2}.$$

*Proof of Lemma D.1.* Apply Chebyshev's inequality on random variable $X$ and notice that $X$ is symmetric, then we can easily observe this lemma. □

Now we are ready to present the certified robustness for different ensemble models.

## D.2 STATISTICAL CERTIFIED ROBUSTNESS GUARANTEES

The main results in this subsection are Theorem 6 and Theorem 7.

### D.2.1 CERTIFIED ROBUSTNESS FOR SINGLE MODEL

As the start point, we first show a direct proposition stating the certified robustness guarantee of the single model.

**Proposition D.1** (Certified Robustness for Single Model). *Let $\varepsilon$ be a random variable. Let $F$ be a classification model, which is $(\varepsilon, \lambda_3, p)$-single confident. Let $\boldsymbol{x}_0 \in \mathbb{R}^d$ be the input with ground-truth $y_0 \in [C]$. Suppose $f(\boldsymbol{x}_0 + \varepsilon)_{y_0}$ follows symmetric distribution with mean $\mu$ and variance $s^2$, where $\mu > (1 + \lambda_3^{-1})^{-1}$. We have*

$$\Pr_{\varepsilon}(F(\boldsymbol{x}_0 + \varepsilon) = y_0) \geq 1 - p - \frac{s^2}{2(\mu - (1 + \lambda_3^{-1})^{-1})^2}.$$

*Proof of Proposition D.1.* We consider the distribution of quantity $Y := f(\boldsymbol{x}_0 + \varepsilon)_{y_0} - \lambda_3(1 - f(\boldsymbol{x}_0 + \varepsilon)_{y_0})$. Since the model $F$ is $(\varepsilon, \lambda_3, p)$-single confident, with probability $1 - p$, $Y \leq f(\boldsymbol{x}_0 + \varepsilon)_{y_0} - \max_{y_j \in [C]:y_j \neq y_0} f(\boldsymbol{x}_0 + \varepsilon)_{y_j}$. We note that since

$$\mathbb{E}Y = (1 + \lambda_3)\mu - \lambda_3, \ \text{Var}(Y) = (1 + \lambda_3)^2 s^2,$$

from Lemma D.1,

$$\Pr(Y \leq 0) \leq \frac{s^2}{2(\mu - (1 + \lambda_3^{-1})^{-1})^2}.$$

Thus,

$$\begin{aligned}
\Pr(F(\boldsymbol{x}_0 + \varepsilon) = y_0) &= 1 - \Pr(F(\boldsymbol{x}_0 + \varepsilon) \neq y_0) \\
&= 1 - \Pr\left( f(\boldsymbol{x}_0 + \varepsilon)_{y_0} - \max_{y_j \in [C]:y_j \neq y_0} f(\boldsymbol{x}_0 + \varepsilon)_{y_j} < 0 \right) \\
&\geq 1 - p - \Pr(Y \leq 0) \\
&\geq 1 - p - \frac{s^2}{2(\mu - (1 + \lambda_3^{-1})^{-1})^2}.
\end{aligned}$$

$\square$

### D.2.2 CERTIFIED ROBUSTNESS FOR ENSEMBLES

Now we are ready to prove the certified robustness of the Weighted Ensemble and Max-Margin Ensemble (Theorems 6 and 7).

In the following text, we first define statistical margins for both WE and MME, and point out their connections to the notion of $(\varepsilon, p)$-Statistical Robust. Then, we reason about the expectation, variance, and tail bounds of the statistical margins. Finally, we derive the certified robustness from the statistical margins.

**Definition D.1** ($\hat{X}_1$; Statistical Margin for WE $\mathcal{M}_{\text{WE}}$). Let $\mathcal{M}_{\text{WE}}$ be Weighted Ensemble defined over base models $\{F_i\}_{i=1}^N$ with weights $\{w_i\}_{i=1}^N$. Suppose $\mathcal{M}_{\text{WE}}$ is $(\varepsilon, \lambda_1, p)$-WE-confident. We define random variable $\hat{X}_1$ which is depended by random variable $\varepsilon$:

$$\hat{X}_1(\boldsymbol{\epsilon}) := (1 + \lambda_1) \sum_{j=1}^N w_j f_j(\boldsymbol{x}_0 + \boldsymbol{\epsilon})_{y_0} - \lambda_1 \|\boldsymbol{w}\|_1. \tag{42}$$

**Definition D.2** ($\hat{X}_2$; Statistical Margin for MME $\mathcal{M}_{\text{MME}}$). Let $\mathcal{M}_{\text{MME}}$ be Max-Margin Ensemble defined over base models $\{F_i\}_{i=1}^N$. Suppose $\mathcal{M}_{\text{MME}}$ is $(\varepsilon, \lambda_2, p)$-MME-confident. We define random variable $\hat{X}_2$ which is depended by random variable $\varepsilon$:

$$\hat{X}_2(\boldsymbol{\epsilon}) := (1 + \lambda_2) \left( \max_{i \in [N]} f_i(\boldsymbol{x}_0 + \boldsymbol{\epsilon})_{y_0} + \min_{i \in [N]} f_i(\boldsymbol{x}_0 + \boldsymbol{\epsilon})_{y_0} \right) - 2\lambda_2. \tag{43}$$

We have the following observation:

**Lemma D.2.** *For Weighted Ensemble,*

$$\Pr_{\varepsilon} \left( \mathcal{M}_{\mathrm{WE}}(\boldsymbol{x}_0 + \varepsilon) = y_0 \right) \geq 1 - p - \Pr_{\varepsilon} \left( \hat{X}_1(\varepsilon) < 0 \right).$$

*For Max-Margin Ensemble,*

$$\Pr_{\varepsilon} \left( \mathcal{M}_{\mathrm{MME}}(\boldsymbol{x}_0 + \varepsilon) = y_0 \right) \geq 1 - p - \Pr_{\varepsilon} \left( \hat{X}_2(\varepsilon) < 0 \right).$$

*Proof of Lemma D.2.* (1) For Weighted Ensemble, we define the random variable $X_1$:

$$X_1(\boldsymbol{\epsilon}) := \min_{y_i \in [C]: y_i \neq y_0} \sum_{j=1}^{N} w_j f_j^{y_0/y_i}(\boldsymbol{x}_0 + \boldsymbol{\epsilon}).$$

Since $\mathcal{M}_{\mathrm{WE}}$ is $(\varepsilon, \lambda_1, p)$-WE-confident, from Definition 9, with probability $1 - p$, we have

$$X_1(\varepsilon) \geq \sum_{j=1}^{N} w_j \left( f_j(\boldsymbol{x}_0 + \varepsilon)_{y_0} - \lambda_2(1 - f_j(\boldsymbol{x}_0 + \varepsilon)_{y_0}) \right)$$

$$= (1 + \lambda_2) \sum_{j=1}^{N} w_j f_j(\boldsymbol{x}_0 + \varepsilon)_{y_0} - \lambda_1 \|\boldsymbol{w}\|_1 = \hat{X}_1(\varepsilon).$$

Therefore,

$$\Pr_{\varepsilon}(\mathcal{M}_{\mathrm{WE}}(\boldsymbol{x}_0 + \varepsilon) = y_0) = \Pr_{\varepsilon}(X_1(\varepsilon) \geq 0) \geq 1 - p - \Pr_{\varepsilon}(\hat{X}_2(\varepsilon) < 0).$$

(2) For Max-Margin Ensemble, we define the random variable $X_2$:

$$X_2(\boldsymbol{\epsilon}) := \max_{i \in [N]} \min_{y_i \in [C]: y_i \neq y_0} f_i^{y_0/y_i}(\boldsymbol{x}_0 + \boldsymbol{\epsilon}) + \min_{i \in [N]} \min_{y_i \in [C]: y_i \neq y_0} f_i^{y_0/y_i}(\boldsymbol{x}_0 + \boldsymbol{\epsilon}).$$

Similarly, since $\mathcal{M}_{\mathrm{MME}}$ is $(\varepsilon, \lambda_2, p)$-MME-confident, from Definition 10, with probability $1 - p$, we have

$$X_2(\boldsymbol{\epsilon}) \geq \max_{i \in [N]} \left( f_i(\boldsymbol{x}_0 + \varepsilon)_{y_0} - \lambda_2(1 - f_i(\boldsymbol{x}_0 + \varepsilon)_{y_0}) \right) + \min_{i \in [N]} \left( f_i(\boldsymbol{x}_0 + \varepsilon)_{y_0} - \lambda_2(1 - f_i(\boldsymbol{x}_0 + \varepsilon)_{y_0}) \right)$$

$$= (1 + \lambda_2) \left( \max_{i \in [N]} f_i(\boldsymbol{x}_0 + \boldsymbol{\epsilon})_{y_0} + \min_{i \in [N]} f_i(\boldsymbol{x}_0 + \boldsymbol{\epsilon})_{y_0} \right) - 2\lambda_2 = \hat{X}_2(\varepsilon).$$

Moreover, from Lemma B.1, we know

$$\Pr_{\varepsilon}(\mathcal{M}(\boldsymbol{x}_0 + \varepsilon) = y_0) \geq \Pr_{\varepsilon}(X_2(\varepsilon) \geq 0) \geq 1 - p - \Pr_{\varepsilon}(\hat{X}_2(\varepsilon) < 0).$$

$\square$

As the result, to quantify the statistical robustness of two types of ensembles, we can analyze the distribution of statistical margins $\hat{X}_1$ and $\hat{X}_2$.

**Lemma D.3** (Expectation and variance of $\hat{X}_1$ and $\hat{X}_2$). *Let $\hat{X}_1$ and $\hat{X}_2$ be defined by Definition D.1 and Definition D.2 respectively. Assume $\{f_i(\boldsymbol{x}_0 + \varepsilon)_{y_0}\}_{i=1}^{N}$ are i.i.d. and follow symmetric distribution with mean $\mu$ and variance $s^2$. Define $s_f^2 = \mathrm{Var}(\min_{i \in [N]} f_i(\boldsymbol{x}_0 + \varepsilon)_{y_0})$. We have*

$$\mathbb{E}\,\hat{X}_1(\varepsilon) = (1 + \lambda_1)\|\boldsymbol{w}\|_1\mu - \lambda_1\|\boldsymbol{w}\|_1, \quad \mathrm{Var}\,\hat{X}_1(\varepsilon) = (1 + \lambda_1)^2 s^2 \|\boldsymbol{w}\|_2^2,$$

$$\mathbb{E}\,\hat{X}_2(\varepsilon) = 2(1 + \lambda_2)\mu - 2\lambda_2, \qquad \mathrm{Var}\,\hat{X}_2(\varepsilon) \leq 4(1 + \lambda_2)^2 s_f^2.$$

*Proof of Lemma D.3.*

$$\mathbb{E}\hat{X}_1(\varepsilon) = (1 + \lambda_1) \sum_{j=1}^{N} \mathbb{E}w_j f_j(\boldsymbol{x}_0 + \boldsymbol{\epsilon})_{y_0} - \lambda_1\|\boldsymbol{w}\|_1 = (1 + \lambda_1)\|\boldsymbol{w}\|_1\mu - \lambda_1\|\boldsymbol{w}\|_1;$$

$$\mathrm{Var}\hat{X}_1(\varepsilon) = (1 + \lambda_1)^2 \sum_{j=1}^{N} w_j^2 \mathrm{Var}(f_j(\boldsymbol{x}_0 + \boldsymbol{\epsilon})_{y_0}) = (1 + \lambda_1)^2 s^2 \|\boldsymbol{w}\|_2^2.$$

According to the symmetric distribution property of $\{f_i(\boldsymbol{x}_0 + \varepsilon)_{y_0}\}_{i=1}^N$, we have

$$\mathbb{E}\,\hat{X}_2(\varepsilon) = \mathbb{E}(1 + \lambda_2)\left(\max_{i\in[N]} f_i(\boldsymbol{x}_0 + \boldsymbol{\epsilon})_{y_0} + \min_{i\in[N]} f_i(\boldsymbol{x}_0 + \boldsymbol{\epsilon})_{y_0}\right) - 2\lambda_2$$
$$= 2(1 + \lambda_2)\mu - 2\lambda_2.$$

Also, due the symmetry, we have

$$\mathrm{Var}\left(\min_{i\in[N]} f_i(\boldsymbol{x}_0 + \varepsilon)_{y_0}\right) = \mathrm{Var}\left(\max_{i\in[N]} f_i(\boldsymbol{x}_0 + \varepsilon)_{y_0}\right) = s_f^2.$$

As a result,

$$\mathrm{Var}\,\hat{X}_2(\varepsilon) \leq (1 + \lambda_2)^2 \cdot 4s_f^2.$$

$\square$

From Lemma D.3, now with Lemma D.1, we are ready to derive the statistical robustness lower bound for WE and MME.

**Theorem 6** (Certified Robustness for WE). *Let $\varepsilon$ be a random variable supported on $\mathbb{R}^d$. Let $\mathcal{M}_{\mathrm{WE}}$ be a Weighted Ensemble defined over $\{F_i\}_{i=1}^N$ with weights $\{w_i\}_{i=1}^N$. The $\mathcal{M}_{\mathrm{WE}}$ is $(\varepsilon, \lambda_1, p)$-WE confident. Let $\boldsymbol{x}_0 \in \mathbb{R}^d$ be the input with ground-truth label $y_0 \in [C]$. Assume $\{f_i(\boldsymbol{x}_0 + \varepsilon)_{y_0}\}_{i=1}^N$, the confidence scores across base models for label $y_0$, are i.i.d. and follow symmetric distribution with mean $\mu$ and variance $s^2$, where $\mu > (1 + \lambda_1^{-1})^{-1}$. We have*

$$\Pr_\varepsilon(\mathcal{M}_{\mathrm{WE}}(\boldsymbol{x}_0 + \varepsilon) = y_0) \geq 1 - p - \frac{\|\boldsymbol{w}\|_2^2}{\|\boldsymbol{w}\|_1^2} \cdot \frac{s^2}{2\left(\mu - \left(1 + \lambda_1^{-1}\right)^{-1}\right)^2}. \tag{44}$$

**Theorem 7** (Certified Robustness for MME). *Let $\varepsilon$ be a random variable. Let $\mathcal{M}_{\mathrm{MME}}$ be a Max-Margin Ensemble defined over $\{F_i\}_{i=1}^N$. The $\mathcal{M}_{\mathrm{MME}}$ is $(\varepsilon, \lambda_2, p)$-MME confident. Let $\boldsymbol{x}_0 \in \mathbb{R}^d$ be the input with ground-truth label $y_0 \in [C]$. Assume $\{f_i(\boldsymbol{x}_0 + \varepsilon)_{y_0}\}_{i=1}^N$, the confidence scores across base models for label $y_0$, are i.i.d. and follow symmetric distribution with mean $\mu$ where $\mu > (1 + \lambda_2^{-1})^{-1}$. Define $s_f^2 = \mathrm{Var}(\min_{i\in[N]} f_i(\boldsymbol{x}_0 + \varepsilon)_{y_0})$. We have*

$$\Pr_\varepsilon(\mathcal{M}_{\mathrm{MME}}(\boldsymbol{x}_0 + \varepsilon) = y_0) \geq 1 - p - \frac{s_f^2}{2\left(\mu - \left(1 + \lambda_2^{-1}\right)^{-1}\right)^2}. \tag{45}$$

*Proof of Theorems 6 and 7.* Combining Lemmas D.1 to D.3, we get the theorem. $\square$

*Remark.* Theorems 6 and 7 provide two statistical robustness lower bounds for both types of ensembles, which is shown to be able to translate to certified robustness.

For the Weighted Ensemble, noticing that $\hat{X}_1$ is the weighted sum of several independent variables, we can further apply McDiarmid's Inequality to get another bound

$$\Pr_\varepsilon(\mathcal{M}_{\mathrm{WE}}(\boldsymbol{x}_0 + \varepsilon) = y_0) \geq 1 - p - \exp\left(-2\frac{\|\boldsymbol{w}\|_1^2}{\|\boldsymbol{w}\|_2^2}\left(\mu - \left(1 + \lambda_1^{-1}\right)^{-1}\right)^2\right),$$

which is tighter than Equation (44) when $\|\boldsymbol{w}\|_1^2/\|\boldsymbol{w}\|_2^2$ is large. For average weighted ensemble, $\|\boldsymbol{w}\|_1^2/\|\boldsymbol{w}\|_2^2 = N$. Thus, when $N$ is large, this bound is tighter.

Both theorems are applicable under the i.i.d. assumption of confidence scores. The another assumption $\mu > \max\{(1 + \lambda_1^{-1})^{-1}, (1 + \lambda_2^{-1})^{-1}\}$ insures that both ensembles have higher probability of predicting the true class rather than other classes, i.e., the ensembles have non-trivial clean accuracy.

### D.3 COMPARISON OF CERTIFIED ROBUSTNESS

We first show and prove an important lemma. Then, based on the lemma and Theorems 6 and 7, we derive the comparison corollary.

**Lemma D.4.** *For $\mu, \lambda_1, \lambda_2, C > 0$, when $\max\{\lambda_1/(1 + \lambda_1), \lambda_2/(1 + \lambda_2)\} < \mu \leq 1$, and $C < 1$, we have*

$$\frac{\mu - (\lambda_2^{-1} + 1)^{-1}}{\mu - (\lambda_1^{-1} + 1)^{-1}} < C \iff \frac{\lambda_1}{\lambda_2} < \lambda_2^{-1}\left(\left(C^{-1}\left(\mu - \frac{\lambda_2}{1 + \lambda_2}\right) + 1 - \mu\right)^{-1} - 1\right). \quad (46)$$

*Proof of Lemma D.4.*

$$\frac{\mu - (\lambda_2^{-1} + 1)^{-1}}{\mu - (\lambda_1^{-1} + 1)^{-1}} < C$$

$$\iff \frac{1}{\lambda_2^{-1} + 1} - \frac{C}{\lambda_1^{-1} + 1} > \mu(1 - C)$$

$$\iff \frac{\lambda_1/\lambda_2}{\lambda_2^{-1} + \lambda_1/\lambda_2} < \frac{C^{-1}}{\lambda_2^{-1} + 1} - \mu(C^{-1} - 1)$$

$$\iff \frac{\lambda_1}{\lambda_2}\left(1 - \mu + C^{-1}\left(\mu - \frac{1}{\lambda_2^{-1} + 1}\right)\right) < \lambda_2^{-1}\left(C^{-1}\left(\frac{1}{\lambda_2^{-1} + 1} - \mu\right) + \mu\right)$$

$$\iff \frac{\lambda_1}{\lambda_2} < \lambda_2^{-1} \frac{C^{-1}\left(\frac{1}{\lambda_2^{-1} + 1} - \mu\right) + \mu}{C^{-1}\left(\mu - \frac{1}{\lambda_2^{-1} + 1}\right) + 1 - \mu}$$

$$\iff \frac{\lambda_1}{\lambda_2} < \lambda_2^{-1}\left(\left(C^{-1}\left(\mu - \frac{\lambda_2}{1 + \lambda_2}\right) + 1 - \mu\right)^{-1} - 1\right).$$

$\square$

Now we can show and prove the comparison corollary.

**Corollary 5** (Comparison of Certified Robustness). *Let $\varepsilon$ be a random variable supported on $\mathbb{R}^d$. Over base models $\{F_i\}_{i=1}^N$, let $\mathcal{M}_{\mathrm{MME}}$ be Max-Margin Ensemble, and $\mathcal{M}_{\mathrm{WE}}$ the Weighted Ensemble with weights $\{w_i\}_{i=1}^N$. Let $\boldsymbol{x}_0 \in \mathbb{R}^d$ be the input with ground-truth label $y_0 \in [C]$. Assume $\{f_i(\boldsymbol{x}_0 + \varepsilon)_{y_0}\}_{i=1}^N$, the confidence scores across base models for label $y_0$, are i.i.d, and follow symmetric distribution with mean $\mu$ and variance $s^2$, where $\mu > \max\{(1 + \lambda_1^{-1})^{-1}, (1 + \lambda_2^{-1})^{-1}\}$. Define $s_f^2 = \mathrm{Var}(\min_{i \in [N]} f_i(\boldsymbol{x}_0 + \varepsilon)_{y_0})$ and assume $s_f < s$.*

- *When*

$$\frac{\lambda_1}{\lambda_2} < \lambda_2^{-1}\left(\left(\frac{s}{s_f}\left(\mu - (1 + \lambda_2^{-1})^{-1}\right) + 1 - \mu\right)^{-1} - 1\right), \quad (47)$$

   *for any weights $\{w_i\}_{i=1}^N$, $\mathcal{M}_{\mathrm{WE}}$ has higher certified robustness than $\mathcal{M}_{\mathrm{MME}}$.*

- *When*

$$\frac{\lambda_1}{\lambda_2} > \lambda_2^{-1}\left(\left(\frac{s}{\sqrt{N}s_f}\left(\mu - (1 + \lambda_2^{-1})^{-1}\right) + 1 - \mu\right)^{-1} - 1\right), \quad (48)$$

   *for any weights $\{w_i\}_{i=1}^N$, $\mathcal{M}_{\mathrm{MME}}$ has higher certified robustness than $\mathcal{M}_{\mathrm{WE}}$.*

*Here, the certified robustness is given by Theorems 6 and 7.*

*Proof of Corollary 5.* (1) According to Lemma D.4, we have

$$\frac{\lambda_1}{\lambda_2} < \lambda_2^{-1} \left( \left( \frac{s}{s_f} \left( \mu - \left( 1 + \lambda_2^{-1} \right)^{-1} \right) + 1 - \mu \right)^{-1} - 1 \right)$$

$$\Longrightarrow \frac{\mu - (\lambda_2^{-1} + 1)^{-1}}{\mu - (\lambda_1^{-1} + 1)^{-1}} < \frac{s_f}{s}$$

$$\Longrightarrow \sqrt{\frac{\|\boldsymbol{w}\|_2^2}{\|\boldsymbol{w}\|_1^2}} \frac{\mu - (\lambda_2^{-1} + 1)^{-1}}{\mu - (\lambda_1^{-1} + 1)^{-1}} < \frac{s_f}{s}$$

$$\Longrightarrow \frac{\|\boldsymbol{w}\|_2^2}{\|\boldsymbol{w}\|_1^2} \cdot \frac{s^2}{2 \left( \mu - \left( 1 + \lambda_1^{-1} \right)^{-1} \right)^2} < \frac{s_f^2}{2 \left( \mu - \left( 1 + \lambda_2^{-1} \right)^{-1} \right)^2}.$$

According to Theorems 6 and 7, we know the RHS in Equation (44) is larger than the RHS in Equation (45), i.e., $\mathcal{M}_{\mathrm{WE}}$ has higher certified robustnesss than $\mathcal{M}_{\mathrm{MME}}$.

(2) According to Lemma D.4, we have

$$\frac{\lambda_1}{\lambda_2} > \lambda_2^{-1} \left( \left( \frac{s}{\sqrt{N} s_f} \left( \mu - \left( 1 + \lambda_2^{-1} \right)^{-1} \right) + 1 - \mu \right)^{-1} - 1 \right)$$

$$\Longrightarrow \frac{\mu - (\lambda_2^{-1} + 1)^{-1}}{\mu - (\lambda_1^{-1} + 1)^{-1}} > \frac{\sqrt{N} s_f}{s}$$

$$\Longrightarrow \sqrt{\frac{\|\boldsymbol{w}\|_2^2}{\|\boldsymbol{w}\|_1^2}} \frac{\mu - (\lambda_2^{-1} + 1)^{-1}}{\mu - (\lambda_1^{-1} + 1)^{-1}} > \frac{s_f}{s}$$

$$\Longrightarrow \frac{\|\boldsymbol{w}\|_2^2}{\|\boldsymbol{w}\|_1^2} \cdot \frac{s^2}{2 \left( \mu - \left( 1 + \lambda_1^{-1} \right)^{-1} \right)^2} > \frac{s_f^2}{2 \left( \mu - \left( 1 + \lambda_2^{-1} \right)^{-1} \right)^2}.$$

According to Theorems 6 and 7, we know the RHS in Equation (45) is larger than the RHS in Equation (44), i.e., $\mathcal{M}_{\mathrm{MME}}$ has higher certified robustnesss than $\mathcal{M}_{\mathrm{WE}}$. □

*Remark.* (1) Given that $\lambda_1/\lambda_2$ reflects the adversarial transferability among base models Appendix D.1, the corollary implies that, MME is more robust when the transferability is high; WE is more robust when the transferability is low.

(2) As we can observe in the proof, there is a gap between Equation (47) and Equation (48) — when $\lambda_1/\lambda_2$ lies in between RHS of Equation (47) and RHS of Equation (48), it is undetermined which ensemble protocol has higher robustness. Indeed, this uncertainty is caused by the adjustable weights $\{w_i\}_{i=1}^N$ of the Weighted Ensemble. If we only consider the average ensemble, then this gap is closed:

$$\frac{\lambda_1}{\lambda_2} \underset{\mathcal{M}_{\mathrm{WE}} \text{ more robust}}{\overset{\mathcal{M}_{\mathrm{MME}} \text{ more robust}}{\gtrless}} \lambda_2^{-1} \left( \left( \frac{s}{\sqrt{N} s_f} \left( \mu - \left( 1 + \lambda_2^{-1} \right)^{-1} \right) + 1 - \mu \right)^{-1} - 1 \right).$$

(3) Note that we assume that $s_f < s$, where $s^2$ is the variance of single variable and $s_f^2$ is the variance of minimum of $N$ i.i.d. variables. For common symmetry distributions, along with the increase of $N$, $s_f$ shrinks in the order of $O(1/N^B)$ where $B \in (0, 2]$. Thus, as long as $N$ is large, the assumption $s_f < s$ will always hold. An exception is that when these random variables follow the exponential distribution, where $s_f$ does not shrink along with the increase of $N$. However, since these random variables are confidence scores which are in $[0, 1]$, they cannot obey exponential distribution.

## D.4 A CONCRETE CASE: UNIFORM DISTRIBUTION

As shown by Saremi & Srivastava (2020) (Remark 2.1), when the input dimension $d$ is large, the Gaussian noise $\varepsilon \sim \mathcal{N}(0, \sigma^2 \boldsymbol{I}_d) \approx \mathrm{Unif}(\sigma \sqrt{d} S_{d-1})$, i.e., $\boldsymbol{x}_0 + \varepsilon$ is highly *uniformly distributed* on

the $(d-1)$-sphere centered at $\boldsymbol{x}_0$. Motivated by this, we study the case where the confidence scores $\{f_i(\boldsymbol{x}_0 + \varepsilon)_{y_0}\}_{i=1}^N$ are also uniformly distributed.

Under this additional assumption, we can further make the certified robustness for the single model and both ensembles more concrete.

### D.4.1 CERTIFIED ROBUSTNESS FOR SINGLE MODEL

**Proposition D.2** (Certified Robustness for Single Model under Uniform Distribution). *Let $\varepsilon$ be a random variable supported on $\mathbb{R}^d$. Let $F$ be a classification model, which is $(\varepsilon, \lambda_3, p)$-single confident. Let $\boldsymbol{x}_0 \in \mathbb{R}^d$ be the input with ground-truth $y_0 \in [C]$. Suppose $f(\boldsymbol{x}_0 + \varepsilon)_{y_0}$ is uniformly distributed in $[a, b]$. We have*

$$\Pr_\varepsilon(F(\mathrm{x}_0 + \varepsilon) = y_0) \geq 1 - p - \mathrm{clip}\left(\frac{1/(1 + \lambda_3^{-1}) - a}{b - a}\right),$$

*where* $\quad \mathrm{clip}(x) = \max(\min(x, 1), 0).$

*Proof of Proposition D.2.* We consider the distribution of quantity $Y := f(\boldsymbol{x}_0 + \varepsilon)_{y_0} - \lambda_3(1 - f(\boldsymbol{x}_0 + \varepsilon)_{y_0})$. Since the model $F$ is $(\varepsilon, \lambda_3, p)$-single confident, with probability $1 - p$, $Y \leq f(\boldsymbol{x}_0 + \varepsilon)_{y_0} - \max_{y_j \in [C]: y_j \neq y_0} f(\boldsymbol{x}_0 + \varepsilon)_{y_j}$. At the same time, because $f(\boldsymbol{x}_0 + \epsilon)_{y_0}$ follows the distribution $\mathcal{U}([a, b])$,

$$Y = (1 + \lambda_3) f(\boldsymbol{x}_0 + \varepsilon)_{y_0} - \lambda_3$$

follows the distribution $\mathcal{U}([(1 + \lambda_3)a - \lambda_3, (1 + \lambda_3)b - \lambda_3])$. Therefore,

$$\Pr(Y \leq 0) = \mathrm{clip}\left(\frac{\lambda_3 - (1 + \lambda_3)a}{(1 + \lambda_3)(b - a)}\right).$$

As the result,

$$\Pr\left(f(\boldsymbol{x}_0 + \varepsilon)_{y_0} - \max_{y_j \in [C]: y_j \neq y_0} f(\boldsymbol{x}_0 + \varepsilon)_{y_j} \leq 0\right) \leq p + \mathrm{clip}\left(\frac{\lambda_3 - (1 + \lambda_3)a}{(1 + \lambda_3)(b - a)}\right),$$

which is exactly

$$\Pr\left(F(\boldsymbol{x}_0 + \varepsilon) = y_0\right) \geq 1 - p - \mathrm{clip}\left(\frac{\lambda_3 - (1 + \lambda_3)a}{(1 + \lambda_3)(b - a)}\right) = 1 - p - \mathrm{clip}\left(\frac{1/(1 + \lambda_3^{-1}) - a}{b - a}\right).$$

$\square$

### D.4.2 CERTIFIED ROBUSTNESS FOR ENSEMBLES

Still, we define $\hat{X}_1(\varepsilon)$ and $\hat{X}_2(\varepsilon)$ according to Definitions D.1 and D.2. Under the uniform distribution assumption, we have the following lemma.

**Lemma D.5** (Expectation and Variance of $\hat{X}_1$ and $\hat{X}_2$ under Uniform Distribution). *Let $\hat{X}_1$ and $\hat{X}_2$ be defined by Definition D.1 and Definition D.2 respectively. Assume that under the distribution of $\varepsilon$, the base models' confidence scores for true class $\{f_i(\boldsymbol{x}_0 + \varepsilon)_{y_0}\}_{i=1}^N$ are pairwise i.i.d and uniformly distributed in range $[a, b]$. We have*

$$\mathbb{E}\,\hat{X}_1(\varepsilon) = \frac{1}{2}(1 + \lambda_1)\|\boldsymbol{w}\|_1(a + b) - \lambda_1\|\boldsymbol{w}\|_1, \quad \mathrm{Var}\,\hat{X}_1(\varepsilon) = \frac{1}{12}(1 + \lambda_1)^2\|\boldsymbol{w}\|_2^2(b - a)^2,$$

$$\mathbb{E}\,\hat{X}_2(\varepsilon) = (1 + \lambda_2)(a + b) - 2\lambda_2, \qquad \mathrm{Var}\,\hat{X}_2(\varepsilon) \leq (1 + \lambda_2)^2 \frac{4}{N + 1}\left(\frac{2}{N + 2} - \frac{1}{N + 1}\right)(b - a)^2.$$

*Proof of Lemma D.5.* We start from analyzing $\hat{X}_1$. From the definition

$$\hat{X}_1(\boldsymbol{\epsilon}) := (1 + \lambda_1)\sum_{j=1}^N w_j f_j(\boldsymbol{x}_0 + \boldsymbol{\epsilon})_{y_0} - \lambda_1\|\boldsymbol{w}\|_1 \tag{42}$$

where $\{f_i(\boldsymbol{x}_0 + \epsilon)_{y_0}\}_{i=1}^N$ are i.i.d. variables obeying uniform distribution $\mathcal{U}([a, b])$,

$$\mathbb{E}\,\hat{X}_1(\boldsymbol{\epsilon}) = (1 + \lambda_1)\|\boldsymbol{w}\|_1 \frac{a+b}{2} - \lambda_1 \|\boldsymbol{w}\|_1 = \frac{1}{2}(1 + \lambda_1)\|\boldsymbol{w}\|_1(a+b) - \lambda_1 \|\boldsymbol{w}\|_1,$$

$$\mathrm{Var}\,\hat{X}_1(\boldsymbol{\epsilon}) = (1 + \lambda_1)^2 \sum_{j=1}^N w_j^2 \frac{1}{12}(b-a)^2 = \frac{1}{12}(1 + \lambda_1)^2 \|\boldsymbol{w}\|_2^2 (b-a)^2.$$

Now analyze the expectation of $\hat{X}_2$. By the symmetry of uniform distribution, we know

$$\mathbb{E}\,\hat{X}_2(\boldsymbol{\epsilon}) = (1 + \lambda_2) \cdot 2\mathbb{E}\,f_i(\boldsymbol{x}_0 + \boldsymbol{\epsilon})_{y_0} - 2\lambda_2 = (1 + \lambda_2)(a+b) - 2\lambda_2.$$

To reason about the variance, we need the following fact:

**Fact D.1.** *Let $x_1, x_2, \ldots, x_n$ be uniformly distributed and independent random variables. Specifically, for each $1 \le i \le n$, $\boldsymbol{x}_i \sim \mathcal{U}([a, b])$. Then we have*

$$\mathrm{Var}\left(\min_{1 \le i \le n} x_i\right) = \mathrm{Var}\left(\max_{1 \le i \le n} x_i\right) = \frac{1}{n+1}\left(\frac{2}{n+2} - \frac{1}{n+1}\right)(b-a)^2.$$

Observing that each i.i.d. $f_i(\boldsymbol{x}_0 + \varepsilon)_{y_0}$ is exactly identical to $x_i$ in Fact D.1, we have

$$\mathrm{Var}\left(\max_{i \in [N]} f_i(\boldsymbol{x}_0 + \boldsymbol{\epsilon})_{y_0} + \min_{i \in [N]} f_i(\boldsymbol{x}_0 + \boldsymbol{\epsilon})_{y_0}\right) \le \frac{4}{N+1}\left(\frac{2}{N+2} - \frac{1}{N+1}\right)(b-a)^2.$$

Therefore,

$$\mathrm{Var}\,\hat{X}_2(\varepsilon) \le (1 + \lambda_2)^2 \frac{4}{N+1}\left(\frac{2}{N+2} - \frac{1}{N+1}\right)(b-a)^2.$$

$\square$

*Proof of Fact D.1.* From symmetry of uniform distribution, we know $\mathrm{Var}\left(\min_{1 \le i \le n} x_i\right) = \mathrm{Var}\left(\max_{1 \le i \le n} x_i\right)$. So here we only consider $Y := \min_{1 \le i \le n} x_i$. Its CDF $F$ and PDF $f$ can be easily computed:

$$F(y) = 1 - \Pr\left(\min_i x_i \ge y\right) = 1 - \left(\frac{b-y}{b-a}\right)^n, \; f(y) = F'(y) = n\frac{(b-y)^{n-1}}{(b-a)^n}, \text{ where } y \in [a, b].$$

Hence,

$$\mathbb{E}\,Y = \int_a^b yf(y)\mathrm{d}y = \frac{y(b-y)^n + (n+1)^{-1}(b-y)^{n+1}}{(b-a)^n}\bigg|_b^a = a + \frac{b-a}{n+1},$$

$$\begin{aligned}
\mathbb{E}\,Y^2 &= \int_a^b y^2 f(y)\mathrm{d}y = \int_a^b ny^2 \frac{(b-y)^{n-1}}{(b-a)^n}\mathrm{d}y \\
&= -\left(\frac{b-y}{b-a}\right)^n y^2 \bigg|_a^b + 2\int_a^b \left(\frac{b-y}{b-a}\right)^n y\mathrm{d}y \\
&= -\left(\frac{b-y}{b-a}\right)^n y^2 \bigg|_a^b + \frac{2}{n+1}\left(-\frac{(b-y)^{n+1}}{(b-a)^n}y + \int \frac{(b-y)^{n+1}}{(b-a)^n}\mathrm{d}y\right)\bigg|_a^b \\
&= -\left(\frac{b-y}{b-a}\right)^n y^2 \bigg|_a^b + \frac{2}{n+1}\left(-\frac{(b-y)^{n+1}}{(b-a)^n}y - \frac{1}{n+2}\frac{(b-y)^{n+2}}{(b-a)^n}\right)\bigg|_a^b \\
&= a^2 + \frac{2}{n+1}(b-a)a + \frac{2}{(n+1)(n+2)}(b-a)^2.
\end{aligned}$$

As the result, $\mathrm{Var}\,Y = \mathbb{E}\,Y^2 - (\mathbb{E}\,Y)^2 = \frac{1}{n+1}\left(\frac{2}{n+2} - \frac{1}{n+1}\right)(b-a)^2.$ $\square$

Now, similarly, we use Lemma D.1 to derive the statistical robustness lower bound for WE and MME. We omit the proofs since they are direct applications of Lemma D.5, Lemma D.1, and Lemma D.2.

**Theorem 8** (Certified Robustness for WE under Uniform Distribution). *Let $\mathcal{M}_{\mathrm{WE}}$ be a Weighted Ensemble defined over $\{F_i\}_{i=1}^N$ with weights $\{w_i\}_{i=1}^N$. Let $\boldsymbol{x}_0 \in \mathbb{R}^d$ be the input with ground-truth label $y_0 \in [C]$. Let $\varepsilon$ be a random variable supported on $\mathbb{R}^d$. Under the distribution of $\varepsilon$, suppose $\{f_i(\boldsymbol{x}_0 + \varepsilon)_{y_0}\}_{i=1}^N$ are i.i.d. and uniformly distributed in $[a, b]$. The $\mathcal{M}_{\mathrm{WE}}$ is $(\varepsilon, \lambda_1, p)$-WE confident. Assume $\frac{a+b}{2} > \frac{1}{1+\lambda_1^{-1}}$. We have*

$$
\Pr_{\varepsilon}(\mathcal{M}_{\mathrm{WE}}(\boldsymbol{x}_0 + \varepsilon) = y_0) \geq 1 - p - \frac{d_{\boldsymbol{w}} K_1^2}{12},
$$
$$
where \quad d_{\boldsymbol{w}} = \frac{\|\boldsymbol{w}\|_2^2}{\|\boldsymbol{w}\|_1^2}, \ K_1 = \frac{b - a}{\frac{a+b}{2} - \frac{1}{1+\lambda_1^{-1}}}.
\tag{49}
$$

**Theorem 9** (Certified Robustness for MME under Uniform Distribution). *Let $\mathcal{M}_{\mathrm{MME}}$ be a Max-Margin Ensemble over $\{F_i\}_{i=1}^N$. Let $\boldsymbol{x}_0 \in \mathbb{R}^d$ be the input with ground-truth label $y_0 \in [C]$. Let $\varepsilon$ be a random variable supported on $\mathbb{R}^d$. Under the distribution of $\varepsilon$, suppose $\{f_i(\boldsymbol{x}_0 + \varepsilon)_{y_0}\}_{i=1}^N$ are i.i.d. and uniformly distributed in $[a, b]$. $\mathcal{M}_{\mathrm{MME}}$ is $(\varepsilon, \lambda_2, p)$-MME confident. Assume $\frac{a+b}{2} > \frac{1}{1+\lambda_2^{-1}}$. We have*

$$
\Pr_{\varepsilon}(\mathcal{M}_{\mathrm{MME}}(\boldsymbol{x}_0 + \varepsilon) = y_0) \geq 1 - p - \frac{c_N K_2^2}{4},
$$
$$
where \quad c_N = \frac{2}{N + 1}\left(\frac{2}{N + 2} - \frac{1}{N + 1}\right), \ K_2 = \frac{b - a}{\frac{a+b}{2} - \frac{1}{1+\lambda_2^{-1}}}.
\tag{50}
$$

### D.4.3 COMPARISON OF CERTIFIED ROBUSTNESS FOR ENSEMBLES

Now under the uniform distribution, we can also have the certified robustness comparison.

**Corollary 6** (Comparison of Certified Robustness under Uniform Distribution). *Over base models $\{F_i\}_{i=1}^N$, let $\mathcal{M}_{\mathrm{MME}}$ be Max-Margin Ensemble, and $\mathcal{M}_{\mathrm{WE}}$ the Weighted Ensemble with weights $\{w_i\}_{i=1}^N$. Let $\boldsymbol{x}_0 \in \mathbb{R}^d$ be the input with ground-truth label $y_0 \in [C]$. Let $\varepsilon$ be a random variable supported on $\mathbb{R}^d$. Under the distribution of $\varepsilon$, suppose $\{f_i(\boldsymbol{x}_0 + \varepsilon)_{y_0}\}_{i=1}^N$ are i.i.d. and uniformly distributed with mean $\mu$. Suppose $\mathcal{M}_{\mathrm{WE}}$ is $(\varepsilon, \lambda_1, p)$-WE confident, and $\mathcal{M}_{\mathrm{MME}}$ is $(\varepsilon, \lambda_2, p)$-MME confident. Assume $\mu > \max\left\{\frac{1}{1+\lambda_1^{-1}}, \frac{1}{1+\lambda_2^{-1}}\right\}$.*

- *When*

$$
\frac{\lambda_1}{\lambda_2} < \lambda_2^{-1}\left(\left((N+1)\sqrt{\frac{N+2}{6N}}\left(\mu - \frac{1}{1+\lambda_2^{-1}}\right) + 1 - \mu\right)^{-1} - 1\right),
\tag{51}
$$

  $\mathcal{M}_{\mathrm{WE}}$ *has higher certified robustness than* $\mathcal{M}_{\mathrm{MME}}$.

- *When*

$$
\frac{\lambda_1}{\lambda_2} > \lambda_2^{-1}\left(\left(\frac{N+1}{N}\sqrt{\frac{N+2}{6}}\left(\mu - \frac{1}{1+\lambda_2^{-1}}\right) + 1 - \mu\right)^{-1} - 1\right),
\tag{52}
$$

  $\mathcal{M}_{\mathrm{MME}}$ *has higher certified robustness than* $\mathcal{M}_{\mathrm{WE}}$.

- *When*

$$
N > 6\left(1 - \frac{1}{\mu(1 + \lambda_2^{-1})}\right)^{-2} - 2,
\tag{53}
$$

  *for any* $\lambda_1$, $\mathcal{M}_{\mathrm{MME}}$ *has higher or equal certified robustness than* $\mathcal{M}_{\mathrm{WE}}$.

*Here, the certified robustness is given by Theorems 8 and 9.*

*Proof of Corollary 6.* First, we notice that a uniform distribution with mean $\mu$ can be any distribution $\mathcal{U}([a, b])$ where $(a + b)/2 = \mu$. We replace $\mu$ by $(a + b)/2$.

Then (1) and (2) follow from Lemma D.4 similar to the proof of Corollary 5.

(3) Since

$$N > 6\left(1 - \frac{1}{\mu(1 + \lambda_2^{-1})}\right)^{-2} - 2 \Longrightarrow \left(\sqrt{\frac{N+2}{6}}\left(\mu - \frac{1}{1 + \lambda_2^{-1}}\right) + 1 - \mu\right)^{-1} < 1$$

$$\Longrightarrow \left(\frac{N+1}{N}\sqrt{\frac{N+2}{6}}\left(\mu - \frac{1}{1 + \lambda_2^{-1}}\right) + 1 - \mu\right)^{-1} < 1,$$

the RHS of Equation (52) is smaller than 0. Thus, for any $\lambda_1$, since $\lambda_1/\lambda_2 > 0$, the Equation (48) is satisfied. According to (2), $\mathcal{M}_{\mathrm{MME}}$ has higher certified robustnesss than $\mathcal{M}_{\mathrm{WE}}$. $\square$

*Remark.* Comparing to the general corollary (Corollary 5), under the uniform distribution, we have an additional finding that when $N$ is sufficiently large, we will always have higher certified robustness for Max-Margin Ensemble than Weighted Ensemble. This is due to the more efficient variance reduction of Max-Margin Ensemble than Weighted Ensemble. As shown in Lemma D.5, the quantity $\mathrm{Var}\hat{X}(\varepsilon)/(\mathbb{E}\hat{X}(\varepsilon))^2$ for Weighted Ensemble is $\Omega(1/N)$, while for Max-Margin Ensemble is $O(1/N^2)$. As the result, when $N$ becomes larger, Max-Margin Ensemble has higher certified robustness.

We use uniform assumption here to give an illustration in a specific regime. We think it would be an interesting future direction to generalize the analysis to other distributions such as the Gaussian distribution that corresponds to locally linear classifiers. The result from these distribution may be derived from their specific concentration bound for maximum/minimum i.i.d. random variables as discussed at the end of Appendix D.3.

## D.5 NUMERICAL EXPERIMENTS

To validate and give more intuitive explanations for our theorems, we present some numerical experiments.

### D.5.1 ENSEMBLE COMPARISON FROM NUMERICAL SAMPLING

As discussed in Appendix D.1, $\lambda_1/\lambda_2$ reflects the transferability across base models. It is challenging to get enough amount of different ensembles of various transferability levels while keeping all other variables controlled. Therefore, we simulate the transferability of ensembles numerically by varying $\lambda_1/\lambda_2$ (see the definitions of $\lambda_1$ and $\lambda_2$ in Definitions 9 and 10), and sampling the confidence scores $\{f_i(\boldsymbol{x}_0 + \varepsilon)_{y_0}\}$ and $\{\max_{j\in[C]:j\neq y_0} f_i(\boldsymbol{x}_0 + \varepsilon)_j\}$ under determined $\lambda_1$ and $\lambda_2$. For each level of $\lambda_1/\lambda_2$, with the samples, we compute the certified robust radius $r$ using randomized smoothing (Lemma A.1) and compare the radius difference of Weighted Ensemble and Max-Margin Ensemble. According to Corollary 5, we should observe the tendency that along with the increase of transferability $\lambda_1/\lambda_2$, Max-Margin Ensemble would gradually become more certifiably robust than Weighted Ensemble.

Figure 4 verifies the trends: with the increase of $\lambda_1/\lambda_2$, MME model tends to achieve higher certified radius than WE model. Moreover, we notice that under the same $\lambda_1/\lambda_2$, with the larger number of base models $N$, the MME tends to be relatively more certifiably robust compared with WE. This is because we sample the confidence score uniformly and under the uniform distribution, MME tends to be more certifiably robust than WE when the number of base models $N$ becomes large, according to Corollary 6.

The concrete number settings of $\lambda_1, \lambda_2$, and the sampling interval of confidence scores are entailed in the caption of Figure 4.

### D.5.2 ENSEMBLE COMPARISON FROM CERTIFIED ROBUSTNESS PLOTTING

In Corollary 6, we derive the concrete certified robustness for both ensembles and the single model under i.i.d. and uniform distribution assumption. In fact, from the corollary, we can directly compute the certified robust radius without sampling, as long as we assume the added noise $\varepsilon$ is Gaussian. In Figure 5, we plot out such certified robust radius for the single model, the WE, and the MME.

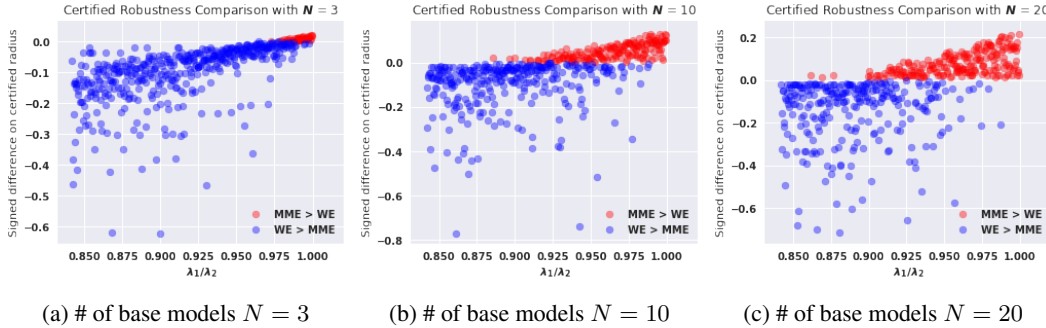

(a) # of base models $N = 3$   (b) # of base models $N = 10$   (c) # of base models $N = 20$

Figure 4: Signed *certified robust radius* difference between MME and WE by $\lambda_1/\lambda_2$ under different numbers of base models $N$. Here we fix $\lambda_2$ to be 0.95 and uniformly sample $\lambda_1 \in [0.8, 0.95)$. The confidence score for the true class on each base model is uniformly sampled from $[a, b]$, where $a$ is sampled from $[0.3, 1.0)$ and $b$ is sampled from $[a, 1.0)$ uniformly for each instance. **Blue** points correspond to the negative radius difference (i.e., WE has larger radius than MME) and **Red** points correspond to the positive radius difference (i.e., MME has larger radius than WE).

Concretely, in the figure, we assume that the true class confidence score for each base model is i.i.d. and *uniformly distributed* in $[a, b]$. The Weighted Ensemble is $(\varepsilon, \lambda_1, 0.01)$-WE confident; the Max-Margin Ensemble is $(\varepsilon, \lambda_2, 0.01)$-MME confident; and the single model is $(\varepsilon, \lambda_3, 0.01)$-MME confident. We guarantee that $\lambda_1 \leq \lambda_3 \leq \lambda_2$ to simulate the scenario that ensembles are based on the same set of base models to make a fair comparison. We directly apply the results from our analysis (Theorem 8, Theorem 9, Proposition D.2) to get the statistical robustness for single model and both ensembles. Then, we leverage Lemma A.1 to get the certified robust radius (with $\sigma = 1.0, N = 100000$ and failing probability $\alpha = 0.001$ which are aligned with realistic setting). The $x$-axis is the number of base models $N$ and the $y$-axis is the certified robustness. We note that $N$ is not applicable to the single model, so we plot the single model's curve by a horizontal red dashed line.

From the figure, we observe that when the number of base models $N$ becomes larger, both ensembles perform much better than the single model. We remark that when $N$ is small, the ensembles have 0 certified robustness mainly because our theoretical bounds for ensembles are not tight enough with the small $N$. Furthermore, we observe that the Max-Margin Ensemble gradually surpasses Weighted Ensemble when $N$ is large, which conforms to our Corollary 6. Note that the left sub-figure has smaller transferability $\lambda_1/\lambda_2$ and the right subfigure has larger transferability $\lambda_1/\lambda_2$, it again conforms to our Corollary 5 and its following remarks in Appendix D.3 that in the left subfigure the Weighted Ensemble is relatively more robust than the Max-Margin Ensemble.

### D.5.3 Ensemble Comparison from Realistic Data

We study the correlation between transferability $\lambda_1/\lambda_2$ and whether Weighted Ensemble or Max-Margin Ensemble is more certifiably robust using realistic data.

By varying the hyper-parameters of DRT, we find out a setting where over the same set of base models, Weighted Ensemble and Max-Margin Ensemble have similar certified robustness, i.e., for about half of the test set samples, WE is more robust; for another half, MME is more robust. We collect $1,000$ test set samples in total. Then, for each test set sample, we compute the transferability $\lambda_1/\lambda_2$ and whether WE or MME has the higher certified robust radius. We remark that $\lambda_1$ and $\lambda_2$ are difficult to be practically estimated so we use the average confidence portion as the proxy:

- For WE,

$$\lambda_1 = \mathbb{E}_\varepsilon \frac{\max_{y_j \in [C]: y_j \neq y_0} \sum_{i=1}^N w_i f_i(\boldsymbol{x}_0 + \varepsilon)_{y_j}}{\sum_{i=1}^N w_i (1 - f_i(\boldsymbol{x}_0 + \varepsilon)_{y_0})}.$$

- For MME,

$$\lambda_2 = \mathbb{E}_\varepsilon \max_{i \in [N]} \frac{\max_{y_j \in [C]: y_j \neq y_0} f_i(\boldsymbol{x}_0 + \varepsilon)_{y_j}}{(1 - f_i(\boldsymbol{x}_0 + \varepsilon)_{y_0})}.$$

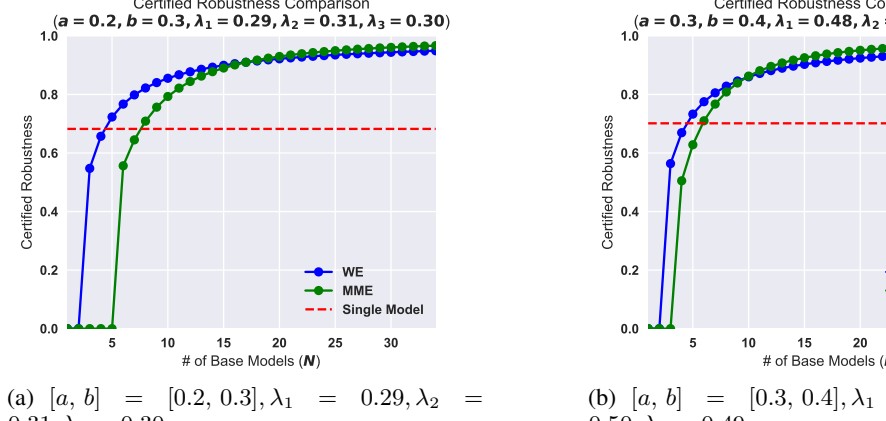

(a) $[a, b] = [0.2, 0.3], \lambda_1 = 0.29, \lambda_2 = 0.31, \lambda_3 = 0.30$.

(b) $[a, b] = [0.3, 0.4], \lambda_1 = 0.48, \lambda_2 = 0.50, \lambda_3 = 0.49$.

Figure 5: Comparison of certified robustness (in terms of certified robust radius) of Max-Margin Ensemble, Weighted Ensemble, and single model under concrete numerical settings. The $y$-axis is the certified robustness and the $x$-axis is the number of base models. The confidence score for the true class is uniformly distributed in $[a, b]$. The Weighted Ensemble (shown by **blue line**) is $(\varepsilon, \lambda_1, 0.01)$-WE confident; the Max-Margin Ensemble (shown by **green line**) is $(\varepsilon, \lambda_2, 0.01)$-MME confident; and the single model (shown by **red line**) is $(\varepsilon, \lambda_3, 0.01)$-MME confident.

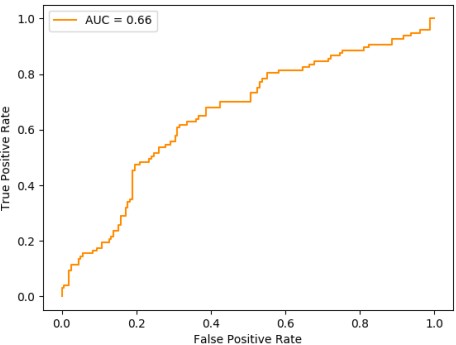

Figure 6: ROC curve of the $\mathbb{I}[\text{MME has higher certified robustness}]$ classification task with the threshold variable $X$.

Now we study the correlation between

$$X := \lambda_1/\lambda_2 - \text{RHS of Equation (48) and } Y := \mathbb{I}[\text{MME has higher certified robustness}].$$

To do so, we draw the ROC curve where the threshold on $X$ does binary classification on $Y$. The curve and the AUC score is shown in Figure 6. From the ROC curve, we find that $X$ and $Y$ are apparently positively correlated since $\text{AUC} = 0.66 > 0.5$, which again verifies Corollary 5. We remark that besides $X$, other factors such as non-symmetric or non-i.i.d. confidence score distribution may also play a role.

*Closing Remarks.* The analysis in this appendix mainly shows two major findings theoretically and empirically: (1) MME is more robust when the adversarial transferability is high; while WE is more robust when the adversarial transferability is low; (2) If each $f_i(x_0 + \varepsilon)_{y_0}$ follows uniform distribution, when number of base models $N$ is sufficiently large, the MME is always more certifiably robust. Our analysis does have limitations: we assume the symmetric and i.i.d. distribution of $f_i(x_0 + \varepsilon)_{y_0}$ or even more strict uniform distribution to derive these findings. Though they model the real-world scenario in some extent as our realistic data results show, they are not perfect considering the transferability among base models and boundedness of confidence scores. We hope current

analysis can open an angle of theoretical analysis of ensembles and leave a more general analysis as the future work.

## E    ANALYSIS OF ALTERNATIVE DESIGN OF DRT

In the main text, we design our DRT based on GD Loss

$$\mathcal{L}_{\text{GD}}(\boldsymbol{x}_0)_{ij} = \left\| \nabla_{\boldsymbol{x}} f_i^{y_0/y_i^{(2)}}(\boldsymbol{x}_0) + \nabla_{\boldsymbol{x}} f_j^{y_0/y_j^{(2)}}(\boldsymbol{x}_0) \right\|_2 \tag{9}$$

and CM Loss

$$\mathcal{L}_{\text{CM}}(\boldsymbol{x}_0)_{ij} = f_i^{y_i^{(2)}/y_0}(\boldsymbol{x}_0) + f_j^{y_j^{(2)}/y_0}(\boldsymbol{x}_0). \tag{10}$$

Following the convention, we apply Gaussian augmentation to train the models, i.e., replacing $\boldsymbol{x}_0$ by $\boldsymbol{x}_0 + \varepsilon$ where $\varepsilon \sim \mathcal{N}(0, \sigma^2 \boldsymbol{I}_d)$ in Equation (9) and Equation (10). We apply these two regularizers to every valid base model pair $(F_i, F_j)$, where the valid pair means both base models predict the ground truth label $y_0$: $F_i(\boldsymbol{x}_0 + \varepsilon) = F_j(\boldsymbol{x}_0 + \varepsilon) = y_0$.

One may concern that in the worst case, there could be $O(N^2)$ valid pairs, i.e., $O(N^2)$ regularization terms in the training loss. However, we should notice that each base model $F_i$ only appears in $O(N)$ valid pairs. Therefore, when $N$ is large, we can optimize DRT by training iteratively, i.e., by regularizing each base model one by one to save the computational cost.

An alternative design inspired from the theorems (e.g., Theorem 1) is to use overall summation instead of pairwise summation, which directly correlates with $I_{y_i}$ (Equation (6)):

$$\mathcal{L}'_{\text{GD}}(\boldsymbol{x}_0) = \left\| \sum_{i=1}^{N} \nabla_{\boldsymbol{x}} f_i^{y_0/y_i^{(2)}}(\boldsymbol{x}_0) \right\|_2, \tag{54}$$

$$\mathcal{L}'_{\text{CM}}(\boldsymbol{x}_0) = \sum_{i=1}^{N} f_i^{y_i^{(2)}/y_0}(\boldsymbol{x}_0). \tag{55}$$

Although this design appears to be more aligned with the theorem and more efficient with $O(N)$ regularization terms, it also requires all base models $F_i$ to have the same runner-up prediction $y_i^{(2)}$ as observed from both theorem and intuition (otherwise diversified gradients and confidence margins are for different and independent labels that are meaningless to jointly optimize). It is less likely to have all base models having the same runner-up prediction than a pair of base models having the same runner-up prediction especially in the initial training phase. Therefore, this alternative design will cause fewer chances of meaningful optimization than the previous design and we use the previous design for our DRT in practice.

## F    EXPERIMENT DETAILS

**Baselines.** We consider the following state-of-the-art baselines for certified robustness: (1) **Gaussian smoothing** (Cohen et al., 2019) trains a smoothed classifier by applying Gaussian augmentation. 2. MACER (Zhai et al., 2019): Adding the regularization term to maximize the certified radius $R = \frac{\sigma}{2}(p_A - p_B)$ on training instances. (2) **SmoothAdv** (Salman et al., 2019) combines adversarial training with Gaussian augmentation. (3) **MACER** (Zhai et al., 2019) improves a single model's certified robustness by adding regularization terms to minimize the Negative Log Likelihood (NLL) between smoothed classifier's output $g_F(\boldsymbol{x})$ and label $y$, and maximize the certified radius $R = \frac{\sigma}{2}(\Phi^{-1}(g_F^\varepsilon(\boldsymbol{x})_y) - \Phi^{-1}(\max_{y' \neq y} g_F^\varepsilon(\boldsymbol{x})_{y'}))$, where $\varepsilon \sim \mathcal{N}(0, \sigma^2 \boldsymbol{I}_d)$ and $g_F^\varepsilon$ is as defined in Equation (3). (4) **Stability** (Li et al., 2019) maintains the stability of the smoothed classifier $g_F$ by minimizing the Rényi Divergence between $g_F(\boldsymbol{x})$ and $g_F(\boldsymbol{x} + \varepsilon)$ where $\varepsilon \sim \mathcal{N}(0, \sigma^2 \boldsymbol{I}_d)$. (5) **SWEEN** (Liu et al., 2020) builds smoothed Weighted Ensemble (WE), which is the only prior work computing certified robustness for ensemble to our knowledge.

**Evaluation Metric.** We report the standard *certified accuracy* under different $L_2$ radii $r$'s as our evaluation metric following Cohen et al. (2019), which is defined as the fraction of the test set samples that the smoothed classifier can certify the robustness within the $L_2$ ball of radius $r$. Since the computation of the accurate value of this metric is intractable, we report the *approximate certified*

*test accuracy* (Cohen et al., 2019) sampled through the Monte Carlo procedure. For each sample, the robustness certification holds with probability at least $1 - \alpha$. Following the literature, we choose $\alpha = 0.001$, $n_0 = 100$ for Monte Carlo sampling during prediction phase, and $n = 10^5$ for Monte Carlo sampling during certification phase. On MNIST and CIFAR-10 we evaluated every 10-th image in the test set, for $1,000$ images total. On ImageNet we evaluated every 100-th image in the validation set, for 500 images total. This evaluation protocol is the same as prior work (Cohen et al., 2019; Salman et al., 2019).

## F.1 MNIST

**Baseline Configuration.** Following the literature (Salman et al., 2019; Jeong & Shin, 2020; Zhai et al., 2019), in each batch, each training sample is Gaussian augmented twice (augmenting more times yields negligible difference as Salman et al. (2019) show). We choose Gaussian smoothing variance $\sigma \in \{0.25, 0.5, 1.0\}$ for training and evaluation for all methods. For SmoothAdv, we consider the attack to be 10-step $L_2$ PGD attack with perturbation scale $\delta = 1.0$ without pretraining and unlabelled data augmentation. We reproduced results similar to their paper by using their open-sourced code[1].

**Training Details.** First, we use LeNet architecture and train each base model for 90 epochs. For the training optimizer, we use the SGD-momentum with the initial learning rate $\alpha = 0.01$. The learning rate is decayed for every 30 epochs with decay ratio $\gamma = 0.1$ and the batch size equals to 256. Then, we apply DRT to finetune our model with small learning rate $\alpha$ for another 90 epochs. We explore different DRT hyper-parameters $\rho_1, \rho_2$ together with the initial learning rate $\alpha$, and report the best certified accuracy on each radius $r$ among all the trained ensemble models.

Table 4: Certified accuracy of DRT-$(\rho_1, \rho_2)$ under different radii $r$ on MNIST dataset. Smoothing parameter $\sigma = 0.25$. The grey rows present the performance of the proposed DRT approach. The brackets show the base models we use.

| Radius $r$ | $\rho_1$ | $\rho_2$ | 0.00 | 0.25 | 0.50 | 0.75 |
|---|---|---|---|---|---|---|
| Gaussian (Cohen et al., 2019) | - | - | 99.1 | 97.9 | 96.6 | 93.0 |
| SmoothAdv (Salman et al., 2019) | - | - | 99.1 | 98.4 | 97.0 | 96.3 |
| MME (Gaussian) | - | - | 99.2 | 98.4 | 96.8 | 93.6 |
| | 0.1 | 0.2 | 99.4 | 98.3 | 97.5 | 95.1 |
| DRT + MME (Gaussian) | 0.1 | 0.5 | 99.5 | **98.6** | 97.1 | 94.8 |
| | 0.2 | 0.5 | **99.5** | 98.5 | 97.4 | 95.1 |
| MME (SmoothAdv) | - | - | 99.2 | 98.2 | 97.3 | 96.4 |
| | 0.1 | 0.2 | 99.1 | 98.4 | 97.5 | 96.4 |
| DRT + MME (SmoothAdv) | 0.1 | 0.5 | 99.1 | 98.3 | **97.6** | **96.7** |
| | 0.2 | 0.5 | 99.1 | 98.4 | 97.5 | 96.6 |
| WE (Gaussian) | - | - | 99.2 | 98.4 | 96.9 | 93.7 |
| | 0.1 | 0.2 | 99.5 | 98.4 | 97.3 | 95.1 |
| DRT + WE (Gaussian) | 0.1 | 0.5 | **99.5** | **98.6** | 97.1 | 94.9 |
| | 0.2 | 0.5 | **99.5** | 98.5 | 97.3 | 95.3 |
| WE (SmoothAdv) | - | - | 99.2 | 98.2 | 97.4 | 96.4 |
| | 0.1 | 0.2 | 99.1 | 98.4 | 97.5 | 96.5 |
| DRT + WE (SmoothAdv) | 0.1 | 0.5 | 99.1 | 98.2 | **97.6** | 96.6 |
| | 0.2 | 0.5 | 99.0 | 98.4 | 97.5 | **96.7** |

**Trend of Certified Accuracy with Perturbation Radius.** We visualize the trend of certified accuracy along with different perturbation radii on different smoothing parameters separately in Figure 7 and Figure 8. For each radius $r$, we present the best certified accuracy among all the trained models. We can notice that while simply applying MME or WE protocol could slightly improve the certified accuracy, DRT could significantly boost the certified accuracy on different radii.

**Average Certified Radius**. We report the Average Certified Radius (ACR) (Zhai et al., 2019): ACR $= \frac{1}{|\mathcal{S}_{\text{test}}|} \sum_{(x,y) \in \mathcal{S}_{\text{test}}} R(x, y)$, where $\mathcal{S}_{\text{test}}$ refers to the test set and $R(x, y)$ the certifed radius on testing

---
[1] https://github.com/Hadisalman/smoothing-adversarial/

Table 5: Certified accuracy of DRT-$(\rho_1, \rho_2)$ under different radii $r$ on MNIST dataset. Smoothing parameter $\sigma = 0.50$. The grey rows present the performance of the proposed DRT approach. The brackets show the base models we use.

| Radius $r$ | $\rho_1$ | $\rho_2$ | 0.00 | 0.25 | 0.50 | 0.75 | 1.00 | 1.25 | 1.50 | 1.75 |
|---|---|---|---|---|---|---|---|---|---|---|
| Gaussian (Cohen et al., 2019) | - | - | 99.0 | 97.7 | 96.4 | 94.7 | 90.0 | 83.0 | 68.2 | 43.5 |
| SmoothAdv (Salman et al., 2019) | - | - | 98.6 | 98.0 | 97.0 | 95.4 | 93.0 | 87.7 | 80.2 | 66.3 |
| MME (Gaussian) | - | - | 99.0 | 97.7 | 96.8 | 94.9 | 90.5 | 84.3 | 69.8 | 48.5 |
| DRT + MME (Gaussian) | 0.2 | 2.0 | 99.1 | 98.4 | 97.2 | 95.2 | 92.6 | 86.5 | 74.3 | 54.1 |
|  |  | 5.0 | 99.1 | **98.6** | 97.1 | 95.3 | 92.6 | 86.2 | 74.0 | 54.3 |
|  | 0.5 | 2.0 | **99.2** | 98.3 | **97.4** | **95.5** | 92.1 | 86.4 | 74.7 | 55.6 |
|  |  | 5.0 | 99.0 | 98.2 | 97.3 | 95.1 | 91.6 | 84.8 | 73.7 | 52.4 |
|  |  | 10.0 | 99.1 | 98.1 | 97.1 | 95.0 | 91.8 | 85.7 | 73.3 | 51.4 |
|  | 1.0 | 5.0 | 99.1 | 98.2 | 97.2 | 95.2 | 92.2 | 85.8 | 74.4 | 54.4 |
|  | 10.0 | 0.1 | 98.8 | 98.0 | 96.8 | 94.7 | 91.5 | 86.5 | 75.5 | 59.1 |
|  |  | 0.2 | 98.9 | 98.1 | 96.9 | 95.1 | 92.1 | 85.8 | 76.1 | 56.4 |
|  |  | 0.5 | 98.7 | 98.1 | 96.8 | 95.2 | 92.1 | 85.8 | 76.0 | 56.9 |
|  |  | 2.5 | 99.0 | 98.3 | 97.0 | 95.1 | 92.4 | 85.8 | 75.7 | 57.0 |
|  |  | 5.0 | 99.0 | 98.1 | 96.8 | 95.0 | 91.9 | 85.5 | 74.4 | 54.6 |
|  |  | 10.0 | 99.0 | 98.2 | 96.9 | 95.1 | 91.9 | 85.5 | 74.6 | 54.5 |
|  | 80.0 | 2.5 | 98.7 | 98.0 | 96.7 | 95.1 | 91.7 | 86.4 | 75.6 | 59.8 |
|  |  | 5.0 | 98.5 | 97.7 | 96.5 | 94.9 | 91.9 | 85.9 | 76.1 | 59.3 |
|  |  | 25.0 | 98.9 | 98.0 | 96.9 | 94.9 | 92.2 | 85.7 | 76.5 | 58.3 |
| MME (SmoothAdv) | - | - | 98.6 | 98.0 | 97.0 | **95.5** | 93.2 | 88.1 | 80.6 | 67.8 |
| DRT + MME (SmoothAdv) | 0.1 | 0.5 | 98.4 | 97.8 | 97.0 | **95.5** | 92.7 | 87.7 | 80.9 | 67.9 |
|  |  | 1.0 | 98.4 | 97.9 | 97.0 | **95.5** | 92.9 | 88.1 | 80.8 | 67.2 |
|  |  | 5.0 | 98.5 | 98.2 | 97.0 | 95.4 | 93.1 | 88.4 | **81.2** | 68.3 |
|  | 0.2 | 0.5 | 98.4 | 97.7 | 97.2 | 95.3 | 92.3 | 87.7 | 79.3 | 68.4 |
|  |  | 2.0 | 98.4 | 97.6 | 97.1 | 95.3 | 92.3 | 87.8 | 80.2 | 67.7 |
|  |  | 5.0 | 98.4 | 97.8 | 97.1 | 95.2 | 93.0 | 87.9 | 80.3 | 68.3 |
|  |  | 10.0 | 98.4 | 97.8 | 97.1 | 95.3 | 92.9 | **88.5** | 81.0 | 67.6 |
|  | 0.3 | 5.0 | 98.4 | 97.5 | 97.1 | 95.0 | 92.4 | 87.7 | 79.7 | 68.3 |
|  |  | 10.0 | 98.5 | 97.7 | 97.0 | 95.2 | 92.6 | **88.5** | 81.1 | 68.1 |
|  | 0.5 | 2.0 | 98.5 | 97.3 | 96.6 | 94.3 | 91.6 | 86.7 | 79.5 | **68.6** |
|  |  | 5.0 | 98.4 | 97.5 | 96.9 | 94.6 | 92.0 | 87.5 | 80.1 | 67.8 |
|  | 1.0 | 0.5 | 97.7 | 96.8 | 95.5 | 92.3 | 89.6 | 84.1 | 76.7 | 66.3 |
|  |  | 1.0 | 97.9 | 96.6 | 95.7 | 92.6 | 89.7 | 84.6 | 77.5 | 66.2 |
|  | 10.0 | 0.1 | 95.4 | 93.3 | 91.2 | 88.1 | 83.8 | 76.8 | 68.3 | 59.9 |
|  |  | 0.2 | 95.5 | 93.7 | 90.9 | 87.7 | 82.0 | 75.7 | 68.7 | 59.6 |
|  |  | 0.5 | 95.0 | 93.3 | 91.1 | 87.8 | 82.6 | 76.3 | 68.2 | 59.7 |
|  |  | 2.5 | 94.6 | 92.9 | 90.1 | 86.3 | 81.6 | 76.0 | 69.6 | 62.5 |
|  |  | 5.0 | 94.3 | 93.1 | 90.0 | 86.1 | 81.9 | 76.3 | 70.0 | 63.6 |
|  |  | 10.0 | 94.9 | 93.4 | 91.3 | 87.3 | 83.2 | 78.2 | 71.8 | 65.9 |
|  | 80.0 | 2.5 | 87.7 | 84.4 | 79.9 | 75.0 | 70.5 | 65.5 | 58.9 | 50.5 |
|  |  | 5.0 | 88.5 | 85.1 | 81.0 | 76.8 | 71.4 | 67.4 | 60.6 | 52.1 |
| WE (Gaussian) | - | - | 99.0 | 97.8 | 96.8 | 94.9 | 90.6 | 84.5 | 70.4 | 48.2 |
| DRT + WE (Gaussian) | 0.2 | 2.0 | **99.2** | 98.4 | 97.2 | 95.2 | 92.5 | 86.2 | 74.3 | 53.5 |
|  |  | 5.0 | 99.1 | **98.6** | 97.1 | 95.3 | 92.6 | 86.4 | 74.2 | 54.4 |
|  | 0.5 | 2.0 | **99.2** | 98.3 | **97.4** | 95.6 | 92.1 | 86.5 | 74.7 | 55.3 |
|  |  | 5.0 | 99.0 | 98.1 | **97.4** | 95.1 | 91.4 | 84.8 | 73.7 | 52.5 |
|  |  | 10.0 | 99.1 | 98.2 | 97.1 | 95.1 | 91.7 | 85.4 | 73.5 | 51.0 |
|  | 1.0 | 5.0 | 99.1 | 98.2 | 97.2 | 95.2 | 92.2 | 85.9 | 75.1 | 55.3 |
|  | 10.0 | 0.1 | 98.8 | 98.0 | 96.8 | 94.8 | 91.6 | 86.7 | 76.3 | 59.0 |
|  |  | 0.2 | 98.8 | 98.1 | 97.0 | 95.0 | 92.1 | 86.0 | 75.7 | 56.8 |
|  |  | 0.5 | 98.8 | 98.1 | 96.9 | 95.2 | 92.2 | 86.0 | 76.2 | 57.0 |
|  |  | 2.5 | 98.9 | 98.3 | 97.0 | 95.1 | 92.4 | 85.9 | 76.2 | 56.3 |
|  |  | 5.0 | 99.0 | 98.1 | 96.9 | 95.0 | 91.8 | 85.5 | 74.5 | 55.0 |
|  |  | 10.0 | 99.0 | 98.1 | 96.9 | 95.1 | 91.9 | 85.7 | 74.3 | 54.4 |
|  | 80.0 | 2.5 | 98.7 | 97.9 | 96.7 | 95.1 | 91.8 | 86.2 | 75.5 | 60.1 |
|  |  | 5.0 | 98.4 | 97.8 | 96.8 | 95.0 | 91.9 | 86.2 | 75.6 | 60.2 |
|  |  | 25.0 | 99.0 | 98.1 | 96.9 | 94.9 | 92.1 | 85.9 | 76.7 | 58.4 |
| WE (SmoothAdv) | - | - | 98.7 | 98.0 | 97.0 | **95.5** | **93.4** | 88.2 | 81.1 | 67.9 |
| DRT + WE (SmoothAdv) | 0.1 | 0.5 | 98.4 | 97.8 | 97.0 | **95.5** | 92.7 | 87.8 | 80.6 | 68.1 |
|  |  | 1.0 | 98.5 | 97.9 | 97.0 | **95.5** | 93.1 | 88.0 | **81.2** | 67.7 |
|  |  | 5.0 | 98.5 | 98.2 | 97.0 | 95.4 | 93.3 | **88.5** | 81.4 | **68.6** |
|  | 0.2 | 0.5 | 98.4 | 97.7 | 97.2 | 95.4 | 92.3 | 87.6 | 79.7 | 68.0 |
|  |  | 2.0 | 98.4 | 97.6 | 97.1 | 95.3 | 92.3 | 87.8 | 80.6 | 68.1 |
|  |  | 5.0 | 98.4 | 97.9 | 97.1 | 95.1 | 93.0 | 88.2 | 80.4 | 69.1 |
|  |  | 10.0 | 98.3 | 97.8 | 97.1 | 95.3 | 92.9 | 88.4 | 80.7 | 68.1 |
|  | 0.3 | 5.0 | 98.4 | 97.5 | 97.1 | 95.0 | 92.4 | 87.9 | 79.9 | 69.3 |
|  |  | 10.0 | 98.4 | 97.7 | 97.0 | 95.2 | 92.6 | 88.4 | 81.1 | 68.2 |
|  | 0.5 | 2.0 | 98.4 | 97.3 | 96.6 | 94.3 | 91.8 | 86.7 | 79.6 | 68.1 |
|  |  | 5.0 | 98.4 | 97.5 | 96.9 | 94.7 | 92.0 | 87.7 | 79.7 | 67.7 |
|  | 1.0 | 0.5 | 97.8 | 96.8 | 95.4 | 92.3 | 89.7 | 84.1 | 77.0 | 65.9 |
|  |  | 1.0 | 97.9 | 96.6 | 95.6 | 92.7 | 89.8 | 84.4 | 77.4 | 66.2 |
|  | 10.0 | 0.1 | 95.3 | 93.5 | 91.2 | 88.7 | 83.8 | 76.8 | 68.9 | 60.1 |
|  |  | 0.2 | 95.4 | 93.8 | 90.9 | 88.1 | 83.2 | 76.6 | 69.1 | 59.9 |
|  |  | 0.5 | 95.1 | 93.5 | 90.9 | 87.7 | 83.6 | 76.6 | 69.1 | 59.8 |
|  |  | 2.5 | 94.8 | 93.0 | 90.5 | 86.8 | 82.1 | 75.1 | 69.1 | 62.0 |
|  |  | 5.0 | 94.4 | 93.3 | 90.1 | 86.6 | 82.0 | 75.8 | 70.0 | 63.2 |
|  |  | 10.0 | 94.7 | 93.3 | 90.5 | 86.8 | 82.5 | 77.2 | 71.8 | 65.6 |
|  | 80.0 | 2.5 | 87.8 | 83.1 | 78.5 | 74.0 | 67.7 | 62.3 | 54.9 | 47.0 |
|  |  | 5.0 | 88.4 | 84.2 | 79.9 | 75.3 | 69.3 | 63.7 | 56.5 | 48.7 |

Table 6: Certified accuracy of DRT-$(\rho_1, \rho_2)$ under different radii $r$ on MNIST dataset. Smoothing parameter $\sigma = 1.00$. The grey rows present the performance of the proposed DRT approach. The brackets show the base models we use.

| Radius $r$ | $\rho_1$ | $\rho_2$ | 0.00 | 0.25 | 0.50 | 0.75 | 1.00 | 1.25 | 1.50 | 1.75 | 2.00 | 2.25 | 2.50 |
|---|---|---|---|---|---|---|---|---|---|---|---|---|---|
| Gaussian (Cohen et al., 2019) | - | - | **96.5** | 94.3 | 91.1 | 87.0 | 80.2 | 71.8 | 60.1 | 46.6 | 33.0 | 20.5 | 11.5 |
| SmoothAdv (Salman et al., 2019) | - | - | 95.3 | 93.5 | 89.3 | 85.6 | 80.4 | 72.8 | 63.9 | 54.6 | 43.2 | 34.3 | 24.0 |
| MME (Gaussian) | - | - | 96.4 | 94.8 | **91.3** | **87.7** | **80.8** | **73.5** | 61.0 | 48.8 | 34.7 | 23.4 | 12.7 |
| DRT + MME (Gaussian) | 0.5 | 2.0 | 96.0 | 93.9 | 90.1 | 86.3 | 80.7 | 73.2 | 63.0 | 52.0 | 38.9 | 26.9 | 15.6 |
| | 0.5 | 5.0 | 95.8 | 94.1 | 90.0 | 86.6 | 80.4 | 72.9 | 62.4 | 51.3 | 40.0 | 27.8 | 16.5 |
| | 1.0 | 5.0 | 95.3 | 93.1 | 89.7 | 85.8 | 80.0 | 72.7 | 62.9 | 52.0 | 39.8 | 28.5 | 17.6 |
| | 5.0 | 0.5 | 91.3 | 89.7 | 85.6 | 78.8 | 73.3 | 65.8 | 59.1 | 52.2 | 43.9 | 36.0 | 29.1 |
| | 5.0 | 2.5 | 92.5 | 90.2 | 87.7 | 82.0 | 76.3 | 69.6 | 60.7 | 52.8 | 43.4 | 35.4 | 26.0 |
| | 5.0 | 5.0 | 93.2 | 90.6 | 88.1 | 82.9 | 78.1 | 70.6 | 62.3 | 52.5 | 43.3 | 34.4 | 23.8 |
| MME (SmoothAdv) | - | - | 95.4 | 93.4 | 89.3 | 86.1 | 80.7 | 73.1 | **65.0** | 55.0 | 44.8 | 35.0 | 25.2 |
| DRT + MME (SmoothAdv) | 0.2 | 2.0 | 94.1 | 91.9 | 88.6 | 84.5 | 79.4 | 72.4 | 63.4 | 54.0 | 45.0 | 36.6 | 27.3 |
| | 0.2 | 5.0 | 94.1 | 91.6 | 88.9 | 84.4 | 79.3 | 72.3 | 63.2 | 54.2 | 46.1 | 36.9 | 28.5 |
| | 0.5 | 2.0 | 92.8 | 91.3 | 87.7 | 83.2 | 77.3 | 71.2 | 62.2 | 53.3 | 45.5 | 37.0 | 29.7 |
| | 0.5 | 5.0 | 92.5 | 91.2 | 88.0 | 83.5 | 78.5 | 71.2 | 62.2 | 53.8 | 45.2 | 37.7 | 29.2 |
| | 1.0 | 5.0 | 92.1 | 90.0 | 86.3 | 81.3 | 76.2 | 69.4 | 61.1 | 54.0 | 46.4 | 38.6 | 31.1 |
| | 5.0 | 1.0 | 89.3 | 86.5 | 82.2 | 76.5 | 70.5 | 62.8 | 54.6 | 48.5 | 41.4 | 35.2 | 29.2 |
| | 5.0 | 5.0 | 87.6 | 83.3 | 78.8 | 73.1 | 67.4 | 61.8 | 56.2 | 50.5 | 44.9 | 38.4 | 32.8 |
| | 10.0 | 20.0 | 82.7 | 79.6 | 75.3 | 72.0 | 67.9 | 63.3 | 58.6 | 51.1 | 46.6 | **40.3** | **34.7** |
| WE (Gaussian) | - | - | 96.3 | **94.9** | **91.3** | **87.7** | 80.7 | **73.5** | 61.1 | 49.0 | 35.2 | 23.7 | 12.9 |
| DRT + WE (Gaussian) | 0.5 | 2.0 | 95.9 | 93.9 | 90.2 | 86.3 | 80.7 | 73.2 | 63.2 | 51.9 | 38.6 | 27.0 | 15.5 |
| | 0.5 | 5.0 | 95.9 | 94.1 | 90.0 | 86.4 | 80.4 | 73.1 | 62.3 | 51.7 | 39.8 | 27.5 | 16.4 |
| | 1.0 | 5.0 | 95.4 | 93.1 | 89.7 | 85.8 | 80.0 | 72.7 | 62.9 | 52.1 | 39.9 | 28.5 | 17.8 |
| | 5.0 | 0.5 | 91.3 | 89.8 | 85.9 | 79.0 | 73.4 | 65.5 | 59.2 | 52.2 | 43.9 | 35.4 | 28.8 |
| | 5.0 | 2.5 | 92.4 | 90.2 | 87.8 | 81.7 | 76.2 | 69.5 | 60.5 | 52.5 | 43.5 | 35.8 | 26.8 |
| | 5.0 | 5.0 | 92.9 | 90.7 | 88.0 | 82.7 | 78.1 | 70.5 | 62.3 | 52.6 | 43.1 | 34.5 | 24.4 |
| WE (SmoothAdv) | - | - | 95.2 | 93.4 | 89.4 | 86.2 | **80.8** | 73.3 | 64.8 | **55.1** | 44.7 | 35.2 | 24.9 |
| DRT + WE (SmoothAdv) | 0.2 | 2.0 | 94.2 | 91.9 | 88.6 | 84.5 | 79.6 | 72.5 | 63.7 | 53.9 | 44.9 | 36.4 | 27.3 |
| | 0.2 | 5.0 | 94.2 | 91.6 | 88.9 | 84.4 | 79.3 | 72.5 | 63.3 | 54.3 | 45.9 | 36.9 | 28.7 |
| | 0.5 | 2.0 | 92.6 | 91.3 | 87.7 | 83.1 | 77.5 | 71.1 | 62.4 | 53.3 | 45.3 | 36.7 | 29.3 |
| | 0.5 | 5.0 | 92.5 | 91.2 | 88.0 | 83.4 | 78.5 | 71.1 | 62.3 | 53.7 | 45.3 | 37.8 | 29.5 |
| | 1.0 | 5.0 | 92.1 | 90.0 | 86.4 | 81.4 | 76.3 | 69.7 | 61.1 | 54.0 | 46.4 | 38.4 | 31.0 |
| | 5.0 | 1.0 | 89.1 | 86.5 | 82.5 | 76.7 | 70.5 | 63.0 | 54.8 | 48.4 | 41.5 | 35.3 | 29.1 |
| | 5.0 | 5.0 | 87.9 | 83.4 | 78.8 | 73.0 | 67.5 | 61.6 | 56.2 | 50.4 | 44.8 | 38.5 | 32.7 |
| | 10.0 | 20.0 | 82.0 | 79.1 | 75.2 | 71.8 | 67.6 | 63.4 | 58.6 | 51.2 | **46.7** | 40.2 | **34.7** |

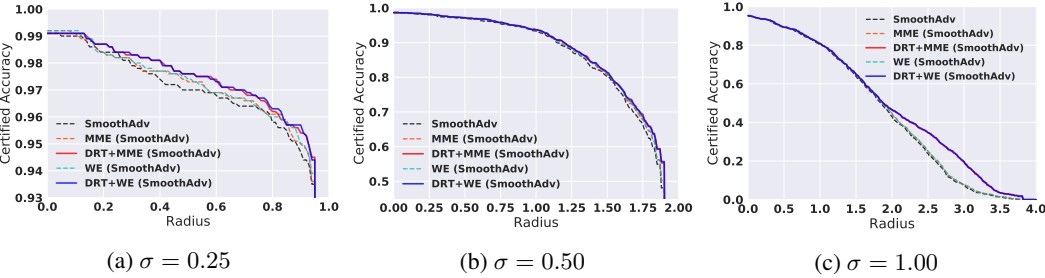

(a) $\sigma = 0.25$     (b) $\sigma = 0.50$     (c) $\sigma = 1.00$

Figure 7: Certified accuracy for ML ensembles with Gaussian smoothed base models, under smoothing parameter $\sigma \in \{0.25, 0.50, 1.00\}$ separately on MNIST.

(a) $\sigma = 0.25$     (b) $\sigma = 0.50$     (c) $\sigma = 1.00$

Figure 8: Certified accuracy for ML ensembles with SmoothAdv base models, under smoothing parameter $\sigma \in \{0.25, 0.50, 1.00\}$ separately on MNIST.

sample $(x, y)$. We evaluate ACR of our DRT-trained ensemble trained with $\sigma \in \{0.25, 0.5, 1.0\}$ smoothing parameter and compare it with other baselines. Results are shown in Table 7.

We can clearly see that our DRT-trained ensemble could still achieve the highest ACR on all the smoothing parameter settings. Especially on $\sigma = 1.00$, our improvement is significant.

Table 7: Average Certified Radius (ACR) of DRT-trained ensemble trained with different smoothing parameter $\sigma \in \{0.25, 0.50, 1.00\}$ on MNIST dataset, compared with other baselines. The grey rows present the performance of the proposed DRT approach. The brackets shows the base models we use.

| Radius $r$ | $\sigma = 0.25$ | $\sigma = 0.50$ | $\sigma = 1.00$ |
|---|---|---|---|
| Gaussian (Cohen et al., 2019) | 0.912 | 1.565 | 1.633 |
| SmoothAdv (Salman et al., 2019) | 0.920 | 1.629 | 1.734 |
| MACER (Zhai et al., 2019) | 0.918 | 1.583 | 1.520 |
| MME / WE (Gaussian) | 0.915 | 1.585 | 1.669 |
| DRT + MME / WE (Gaussian) | 0.923 | 1.637 | 1.745 |
| MME / WE (SmoothAdv) | 0.926 | 1.678 | 1.765 |
| DRT + MME / WE (SmoothAdv) | **0.929** | **1.689** | **1.812** |

**Effects of $\rho_1$ and $\rho_2$.** We investigate the DRT hyper-parameters $\rho_1$ and $\rho_2$ corresponding to different smoothing parameter $\sigma \in \{0.25, 0.5, 1.0\}$. Here we put the detailed results for various hyper-parameter settings in Tables 4 to 6 and bold the numbers with the highest certified accuracy on each radius $r$. From the experiments, we find that the GD loss's weight $\rho_1$ can have the major influence on the ensemble model's functionality: if we choose larger $\rho_1$, the model will achieve slightly lower certified accuracy on small radii, but higher certified accuracy on large radii. We also can not choose too large $\rho_1$ on small $\sigma$ cases (e.g., $\sigma = 0.25$). Otherwise, model's functionality will collapse. Here we show DRT-based models' *certified accuracy* by applying different $\rho_1$ in Figure 9.

Alternatively, we find that the CM loss's weight $\rho_2$ can also have positive influence on model's performance: the larger $\rho_2$ we choose, the higher certified accuracy we could get. Choosing larger and larger $\rho_2$ does not harm model's functionality too much, but the improvement on certified accuracy will become more and more marginal.

**Efficiency Analysis.** We regard the *execution time per mini-batch* as our efficiency criterion. For MNIST with batch size equals to 256, DRT with the Gaussian smoothing base model only requires 1.04s to finish one mini-batch training to achieve the comparable results to the SmoothAdv method which requires 1.86s. Moreover, DRT with the SmoothAdv base model requires 2.52s per training batch but achieves much better results. The evaluation is on single NVIDIA GeForce GTX 1080 Ti GPU.

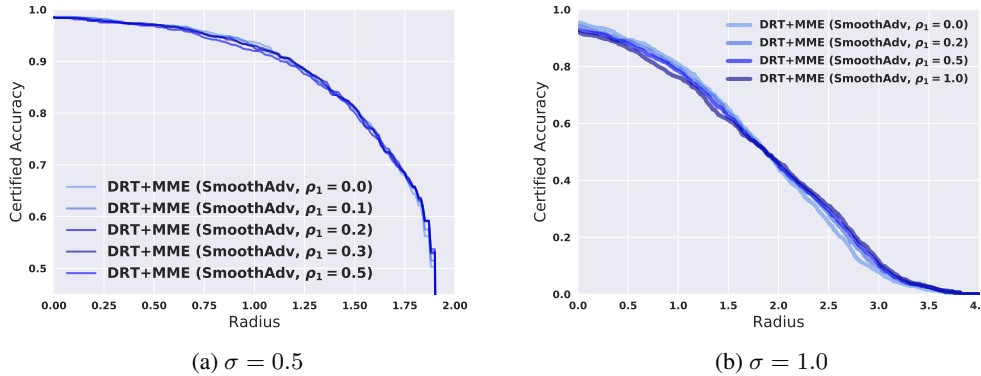

(a) $\sigma = 0.5$         (b) $\sigma = 1.0$

Figure 9: **Effect of $\rho_1$:** Certified accuracy of DRT-based models with MME protocol trained by different GD Loss's weight $\rho_1$ on MNIST. Smoothing parameter $\sigma \in \{0.50, 1.00\}$. Training with large $\rho_1$ will lead to lower certified accuracy on small radii but higher certified accuracy on large radii.

Table 8: Certified accuracy of DRT-$(\rho_1, \rho_2)$ under different radii $r$ on CIFAR-10 dataset. Smoothing parameter $\sigma = 0.25$. The grey rows present the performance of the proposed DRT approach. The brackets show the base models we use.

| Radius $r$ | $\rho_1$ | $\rho_2$ | 0.00 | 0.25 | 0.50 | 0.75 |
|---|---|---|---|---|---|---|
| Gaussian (Cohen et al., 2019) | - | - | 78.9 | 64.4 | 47.4 | 30.6 |
| SmoothAdv (Salman et al., 2019) | - | - | 68.9 | 61.0 | 54.4 | 45.7 |
| MME (Gaussian) | - | - | 80.8 | 68.2 | 53.4 | 37.4 |
| | 0.1 | 0.5 | 81.4 | **70.4** | 57.6 | 43.4 |
| DRT + MME (Gaussian) | 0.2 | 0.5 | 78.8 | 69.2 | 57.8 | 43.8 |
| | 0.5 | 2.0 | 73.3 | 61.7 | 51.0 | 39.3 |
| | 0.5 | 5.0 | 66.2 | 57.1 | 46.2 | 34.4 |
| MME (SmoothAdv) | - | - | 71.4 | 64.5 | 57.6 | 48.4 |
| | 0.1 | 0.5 | 72.6 | 67.2 | **60.2** | 50.3 |
| DRT + MME (SmoothAdv) | 0.2 | 0.5 | 71.8 | 66.5 | 59.3 | 50.4 |
| | 0.5 | 0.5 | 68.2 | 64.3 | 58.2 | 48.9 |
| WE (Gaussian) | - | - | 80.7 | 68.3 | 53.6 | 37.5 |
| | 0.1 | 0.5 | **81.5** | **70.4** | 57.7 | 43.4 |
| DRT + WE (Gaussian) | 0.2 | 0.5 | 78.8 | 69.3 | 57.9 | 44.0 |
| | 0.5 | 2.0 | 73.4 | 61.7 | 51.0 | 39.2 |
| | 0.5 | 5.0 | 66.2 | 57.1 | 46.1 | 34.5 |
| WE (SmoothAdv) | - | - | 71.8 | 64.6 | 57.8 | 48.5 |
| | 0.1 | 0.5 | 72.6 | 67.0 | **60.2** | 50.3 |
| DRT + WE (SmoothAdv) | 0.2 | 0.5 | 71.9 | 66.5 | 59.4 | **50.5** |
| | 0.5 | 0.5 | 68.2 | 64.3 | 58.4 | 49.1 |

## F.2 CIFAR-10

**Baseline Configuration.** Following the literature (Salman et al., 2019; Jeong & Shin, 2020; Zhai et al., 2019), in each batch, each training sample is Gaussian augmented twice (augmenting more times yields negligible difference as Salman et al. (2019) show). We choose Gaussian smoothing variance $\sigma \in \{0.25, 0.5, 1.0\}$ for training and evaluation for all methods. For SmoothAdv, we consider the attack to be 10-step $L_2$ PGD attack with perturbation scale $\delta = 1.0$ without pretraining and unlabelled data augmentation. We also reproduced the similar results mentioned in baseline papers.

**Training Details.** First, we use ResNet-110 architecture and train each base model for 150 epochs. For the training optimizer, we use the SGD-momentum with the initial learning rate $\alpha = 0.1$. The learning rate is decayed for every 50-epochs with decay ratio $\gamma = 0.1$. Then, we use DRT to finetune our model with small learning rate $\alpha$ for another 150 epochs. We also explore different DRT hyper-parameters $\rho_1, \rho_2$ together with the initial learning rate $\alpha$, and report the best certified accuracy on each radius $r$ among all the trained ensemble models.

**Trend of Certified Accuracy with Perturbation Radius.** We visualize the trend of certified accuracy along with different perturbation radii on different smoothing parameters separately in Figure 10 and Figure 11. For each radius $r$, we present the best certified accuracy among all the trained models. We can see the similar trends: Applying either MME or WE ensemble protocol will only give slight improvement while DRT can help make this improvement significant.

**Average Certified Radius**. We report the Average Certified Radius (ACR) (Zhai et al., 2019): ACR $= \frac{1}{|\mathcal{S}_{\text{test}}|} \sum_{(x,y) \in \mathcal{S}_{\text{test}}} R(x, y)$, where $\mathcal{S}_{\text{test}}$ refers to the test set and $R(x, y)$ the certifed radius on testing sample $(x, y)$. We evaluate ACR of our DRT-trained ensemble trained with $\sigma \in \{0.25, 0.5, 1.0\}$ smoothing parameter and compare it with other baselines. Results are shown in Table 11.

Results shows that, DRT-trained ensemble has the highest ACR on almost all the settings. Especially on $\sigma = 1.00$, our improvement is significant.

**Effects of $\rho_1$ and $\rho_2$.** We study the DRT hyper-parameter $\rho_1$ and $\rho_2$ corresponding to different smoothing parameters $\sigma \in \{0.25, 0.5, 1.0\}$ and put the detailed results in Tables 8 to 10. We bold the

Table 9: Certified accuracy of DRT-$(\rho_1, \rho_2)$ under different radii $r$ on CIFAR-10 dataset. Smoothing parameter $\sigma = 0.50$. The grey rows present the performance of the proposed DRT approach. The brackets show the base models we use.

| Radius $r$ | $\rho_1$ | $\rho_2$ | 0.00 | 0.25 | 0.50 | 0.75 | 1.00 | 1.25 | 1.50 | 1.75 |
|---|---|---|---|---|---|---|---|---|---|---|
| Gaussian (Cohen et al., 2019) | - | - | 68.2 | 57.1 | 44.9 | 33.7 | 23.1 | 16.3 | 10.0 | 5.4 |
| SmoothAdv (Salman et al., 2019) | - | - | 60.6 | 54.2 | 47.9 | 41.2 | 34.8 | 28.5 | 21.9 | 17.1 |
| MME (Gaussian) | - | - | 69.5 | 59.6 | 47.3 | 38.4 | 29.0 | 19.6 | 13.3 | 7.6 |
| DRT + MME (Gaussian) | 0.2 | 2.0 | **69.7** | 61.0 | **50.9** | 40.3 | 30.8 | 22.5 | 15.8 | 10.0 |
| | | 5.0 | 68.0 | 59.9 | 50.0 | 40.8 | 30.1 | 22.1 | 15.2 | 9.6 |
| | 0.5 | 2.0 | 67.8 | 58.5 | 49.0 | 39.9 | 31.6 | 23.4 | 16.1 | 10.2 |
| | | 5.0 | 65.5 | 58.4 | 49.0 | 40.1 | 31.2 | 23.6 | 16.5 | 10.2 |
| | 1.0 | 2.0 | 64.5 | 55.8 | 47.5 | 39.4 | 31.1 | 23.6 | 14.8 | 9.3 |
| | | 5.0 | 62.2 | 54.1 | 46.5 | 38.8 | 29.7 | 22.8 | 16.6 | 11.0 |
| | 1.5 | 5.0 | 59.2 | 52.8 | 44.1 | 35.6 | 27.8 | 22.3 | 15.0 | 10.2 |
| | 5.0 | 2.5 | 58.4 | 51.0 | 44.2 | 39.2 | 33.4 | 27.6 | 23.4 | 20.6 |
| | | 5.0 | 56.2 | 49.6 | 45.8 | 40.4 | 34.4 | 29.6 | 24.4 | 20.8 |
| | 10.0 | 2.5 | 52.0 | 46.8 | 42.0 | 36.2 | 32.4 | 27.8 | 23.4 | 19.7 |
| | | 5.0 | 51.2 | 47.5 | 42.5 | 38.1 | 33.7 | 28.9 | 24.9 | 20.9 |
| | 15.0 | 20.0 | 54.5 | 49.8 | 44.7 | 34.9 | 30.2 | 23.0 | 18.7 | 11.1 |
| | 20.0 | 30.0 | 52.2 | 46.2 | 40.2 | 34.4 | 29.4 | 22.6 | 17.8 | 12.8 |
| MME (SmoothAdv) | - | - | 61.0 | 54.8 | 48.7 | 42.2 | 36.2 | 29.8 | 23.9 | 19.1 |
| DRT + MME (SmoothAdv) | 0.2 | 5.0 | 62.2 | 56.4 | 50.3 | 43.4 | 37.5 | 26.7 | 24.6 | 19.4 |
| | 0.5 | 5.0 | 61.9 | 56.2 | 50.3 | 43.5 | 37.6 | 31.8 | 24.8 | 19.6 |
| | 1.0 | 5.0 | 56.4 | 52.6 | 48.2 | **44.4** | 39.6 | 35.8 | **30.4** | 23.6 |
| | 1.5 | 5.0 | 56.0 | 50.8 | 47.2 | 44.2 | **39.8** | 35.0 | 29.4 | 24.0 |
| WE (Gaussian) | - | - | 69.4 | 59.7 | 47.5 | 38.4 | 29.2 | 19.7 | 13.3 | 7.5 |
| DRT + WE (Gaussian) | 0.2 | 2.0 | **69.7** | **61.2** | 50.8 | 40.2 | 30.8 | 22.4 | 15.9 | 10.0 |
| | | 5.0 | 68.0 | 59.9 | 50.1 | 40.8 | 30.1 | 22.1 | 15.4 | 9.7 |
| | 0.5 | 2.0 | 67.8 | 58.5 | 49.2 | 39.8 | 31.7 | 23.5 | 16.2 | 10.4 |
| | | 5.0 | 65.5 | 58.4 | 49.1 | 40.3 | 31.3 | 24.2 | 16.4 | 10.3 |
| | 1.0 | 2.0 | 64.6 | 55.9 | 47.5 | 39.6 | 31.0 | 24.0 | 14.8 | 9.4 |
| | | 5.0 | 62.3 | 54.2 | 46.6 | 38.8 | 29.8 | 22.9 | 16.6 | 10.9 |
| | 1.5 | 5.0 | 59.2 | 52.8 | 44.2 | 35.8 | 27.8 | 22.4 | 15.0 | 10.3 |
| | 5.0 | 2.5 | 58.4 | 51.1 | 44.2 | 39.2 | 33.3 | 27.8 | 23.2 | 20.6 |
| | | 5.0 | 56.2 | 49.7 | 45.8 | 40.3 | 34.2 | 29.6 | 24.5 | 20.8 |
| | 10.0 | 2.5 | 52.0 | 46.9 | 42.0 | 36.4 | 32.5 | 27.8 | 23.5 | 19.7 |
| | | 5.0 | 51.2 | 47.6 | 42.4 | 38.1 | 33.6 | 28.9 | 24.9 | 20.8 |
| | 15.0 | 20.0 | 54.3 | 49.8 | 44.6 | 35.0 | 30.3 | 23.0 | 18.8 | 11.3 |
| | 20.0 | 30.0 | 52.2 | 46.2 | 40.2 | 34.5 | 29.2 | 22.6 | 17.9 | 12.8 |
| WE (SmoothAdv) | - | - | 61.1 | 54.8 | 48.8 | 42.3 | 36.2 | 29.6 | 24.2 | 19.0 |
| DRT + WE (SmoothAdv) | 0.2 | 5.0 | 62.2 | 56.3 | 50.3 | 43.4 | 37.5 | 26.9 | 24.7 | 19.3 |
| | 0.5 | 5.0 | 61.9 | 56.2 | 50.2 | 43.4 | 37.9 | 31.8 | 25.0 | 19.6 |
| | 1.0 | 5.0 | 56.4 | 52.6 | 48.2 | **44.4** | 39.5 | **36.0** | 30.3 | 23.6 |
| | 1.5 | 5.0 | 56.1 | 50.9 | 47.2 | 44.1 | **39.8** | 35.1 | 29.4 | **24.1** |

numbers with the highest certified accuracy on each radius $r$. The results show similar conclusion to our understanding from MNIST.

**Efficiency Analysis.** We also use the *execution time per mini-batch* as our efficiency criterion. For CIFAR-10 with batch size equals to 256, DRT with the Gaussian smoothing base model requires 3.82s to finish one mini-batch training to achieve the competitive results to 10-step PGD attack based SmoothAdv method which requires 6.39s. All the models are trained in parallel on 4 NVIDIA GeForce GTX 1080 Ti GPUs.

## F.3 IMAGENET

For ImageNet, we utilize ResNet-50 architecture and train each base model for 90 epochs using SGD-momentum optimizer. The initial learning rate $\alpha$ is set to 0.1. During training, the learning rate is decayed for every 30-epochs with decay ratio $\gamma = 0.1$. We tried different Gaussian smoothing parameter $\sigma \in \{0.50, 1.00\}$, and consider the best hyper-parameter configuration for each $\sigma$. Then, we use DRT to finetune base models with the learning rate $\alpha = 5 \times 10^{-3}$ for another 90 epochs. Due

Table 10: DRT-$(\rho_1, \rho_2)$ model's certified accuracy under different radii $r$ on CIFAR-10 dataset. Smoothing parameter $\sigma = 1.00$. The grey rows present the performance of the proposed DRT approach. The brackets show the base models we use.

| Radius $r$ | $\rho_1$ | $\rho_2$ | 0.00 | 0.25 | 0.50 | 0.75 | 1.00 | 1.25 | 1.50 | 1.75 | 2.00 |
|---|---|---|---|---|---|---|---|---|---|---|---|
| Gaussian (Cohen et al., 2019) | - | - | 48.9 | 42.7 | 35.4 | 28.7 | 22.8 | 18.3 | 13.6 | 10.5 | 7.3 |
| SmoothAdv (Salman et al., 2019) | - | - | 47.8 | 43.3 | 39.5 | 34.6 | 30.3 | 25.0 | 21.2 | 18.2 | 15.7 |
| MME (Gaussian) | - | - | 50.2 | 44.0 | 37.5 | 30.9 | 24.1 | 19.3 | 15.6 | 11.6 | 8.8 |
| | 0.5 | 5.0 | 49.4 | 44.2 | 37.8 | 31.6 | 25.4 | 22.6 | 18.2 | 14.4 | 12.4 |
| | 1.0 | 5.0 | 49.8 | **44.4** | 39.0 | 31.6 | 25.6 | 22.6 | 18.2 | 15.0 | 12.0 |
| | 1.5 | 5.0 | 48.0 | 42.4 | 36.4 | 30.4 | 26.2 | 22.0 | 18.4 | 15.4 | 12.8 |
| | | 0.5 | 44.6 | 38.6 | 34.6 | 29.2 | 25.6 | 21.8 | 19.4 | 17.0 | 15.6 |
| DRT + MME (Gaussian) | 5.0 | 2.5 | 44.8 | 39.6 | 35.2 | 31.0 | 27.8 | 23.4 | 20.6 | 18.2 | 16.6 |
| | | 10.0 | 45.4 | 40.4 | 36.8 | 30.4 | 26.0 | 21.8 | 19.0 | 15.8 | 13.6 |
| | 10.0 | 20.0 | 44.4 | 40.8 | 36.2 | 31.2 | 27.4 | 21.2 | 18.8 | 17.2 | 13.6 |
| | 15.0 | 20.0 | 42.2 | 39.6 | 34.8 | 30.8 | 26.2 | 22.4 | 18.0 | 16.6 | 15.4 |
| | 20.0 | 30.0 | 33.8 | 30.2 | 26.8 | 22.6 | 18.6 | 16.8 | 15.0 | 12.8 | 11.4 |
| MME (SmoothAdv) | - | - | 48.2 | 43.7 | 40.1 | 35.4 | 31.3 | 26.2 | 22.6 | 19.5 | 16.2 |
| | 0.2 | 5.0 | 48.2 | 43.9 | 40.1 | 35.4 | 31.5 | 26.7 | 22.9 | 19.8 | 16.8 |
| DRT + MME (SmoothAdv) | 0.5 | 5.0 | 48.1 | 43.8 | 40.3 | 35.7 | 31.8 | 26.9 | 23.1 | 20.1 | 17.5 |
| | 1.0 | 5.0 | 46.2 | 43.4 | **40.8** | **37.0** | **34.2** | 30.0 | **26.8** | 23.8 | 20.1 |
| | 1.5 | 5.0 | 47.8 | 43.4 | 39.5 | 35.4 | 31.6 | 26.7 | 23.1 | 20.4 | 18.1 |
| WE (Gaussian) | - | - | **50.4** | 44.1 | 37.5 | 30.9 | 24.2 | 19.2 | 15.9 | 11.8 | 8.9 |
| | 0.5 | 5.0 | 49.5 | 44.3 | 37.8 | 31.8 | 25.6 | 22.5 | 18.2 | 14.4 | 12.3 |
| | 1.0 | 5.0 | 49.8 | **44.4** | 39.1 | 31.7 | 25.6 | 22.8 | 18.4 | 15.1 | 12.1 |
| | 1.5 | 5.0 | 48.2 | 42.5 | 36.6 | 30.4 | 26.1 | 22.1 | 18.2 | 15.7 | 12.6 |
| | | 0.5 | 44.6 | 38.6 | 34.7 | 29.1 | 25.8 | 21.8 | 19.6 | 17.1 | 15.6 |
| DRT + WE (Gaussian) | 5.0 | 2.5 | 44.8 | 39.6 | 35.4 | 31.0 | 27.9 | 23.4 | 20.6 | 18.1 | 16.4 |
| | | 10.0 | 45.4 | 40.3 | 36.8 | 30.4 | 26.2 | 21.8 | 19.1 | 15.8 | 13.6 |
| | 10.0 | 20.0 | 44.5 | 40.8 | 36.2 | 31.3 | 27.4 | 21.2 | 18.9 | 17.2 | 13.5 |
| | 15.0 | 20.0 | 42.2 | 39.7 | 34.8 | 30.8 | 26.1 | 22.4 | 18.0 | 16.7 | 15.4 |
| | 20.0 | 30.0 | 33.8 | 30.4 | 26.8 | 22.8 | 18.6 | 16.9 | 15.0 | 12.7 | 11.2 |
| WE (SmoothAdv) | - | - | 48.2 | 43.7 | 40.2 | 35.4 | 31.5 | 26.2 | 22.7 | 19.6 | 16.0 |
| | 0.2 | 5.0 | 48.2 | 43.8 | 40.2 | 35.4 | 31.5 | 26.8 | 23.0 | 19.9 | 16.7 |
| DRT + WE (SmoothAdv) | 0.5 | 5.0 | 48.2 | 43.8 | 40.5 | 35.7 | 31.9 | 26.8 | 23.3 | 20.2 | 17.5 |
| | 1.0 | 5.0 | 46.2 | 43.4 | 40.6 | **37.0** | **34.2** | 30.1 | **26.8** | **23.9** | **20.3** |
| | 1.5 | 5.0 | 47.8 | 43.4 | 39.6 | 35.4 | 31.4 | 26.7 | 23.0 | 20.4 | 18.1 |

Table 11: Average Certified Radius (ACR) of DRT-trained ensemble trained with different smoothing parameter $\sigma \in \{0.25, 0.50, 1.00\}$ on CIFAR-10 dataset, compared with other baselines. The grey rows present the performance of the proposed DRT approach. The brackets shows the base models we use.

| Radius $r$ | $\sigma = 0.25$ | $\sigma = 0.50$ | $\sigma = 1.00$ |
|---|---|---|---|
| Gaussian | 0.484 | 0.595 | 0.559 |
| SmoothAdv | 0.539 | 0.662 | 0.730 |
| MACER | **0.556** | 0.726 | 0.792 |
| MME / WE (Gaussian) | 0.513 | 0.621 | 0.579 |
| DRT + MME / WE (Gaussian) | 0.551 | 0.687 | 0.744 |
| MME / WE (SmoothAdv) | 0.542 | 0.692 | 0.689 |
| DRT + MME / WE (SmoothAdv) | 0.545 | **0.760** | **0.868** |

to the consideration of achieving high certified accuracy on large radii, we choose large DRT training hyper-parameter $\rho_1$ and $\rho_2$ in practice, which lead to relatively low benign accuracy.

# G ABLATION STUDIES

## G.1 THE EFFECTS OF GRADIENT DIVERSITY LOSS AND CONFIDENCE MARGIN LOSS

To explore the effects of individual Gradient Diversity and Confidence Margin Losses in DRT, we set $\rho_1$ or $\rho_2$ to 0 and tune the other for evaluation on MNIST and CIFAR-10. The results are shown in Table 12 and 13. We observe that both GD and CM losses have positive effects on improving the

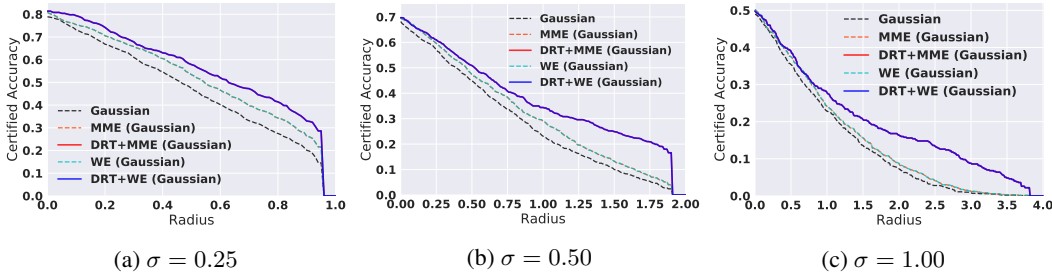

Figure 10: Certified accuracy for ML ensembles with Gaussian smoothed base models, under smoothing parameter $\sigma \in \{0.25, 0.50, 1.00\}$ separately on CIFAR-10.

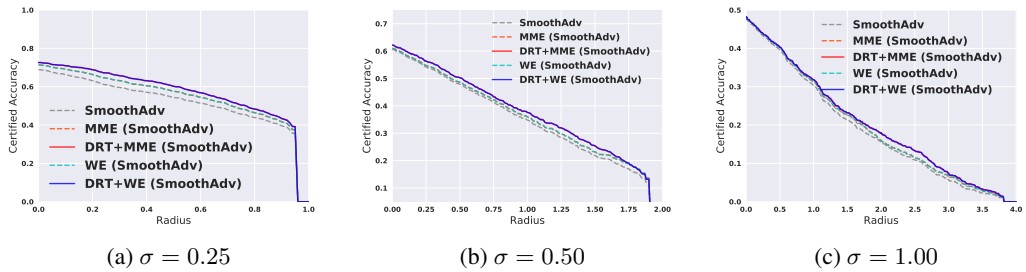

Figure 11: Certified accuracy for ML ensembles with SmoothAdv base models, under smoothing parameter $\sigma \in \{0.25, 0.50, 1.00\}$ separately on CIFAR-10.

certified accuracy while GD plays a major role on larger radii. By combining these two regularization losses together in DRT, the ensemble model achieves the highest certified accuracy under all radii.

Table 12: Certified accuracy achieved by training with GD Loss (GDL) or Confidence Margin Loss (CML) only on MNIST dataset.

| Radius $r$ | 0.00 | 0.25 | 0.50 | 0.75 | 1.00 | 1.25 | 1.50 | 1.75 | 2.00 | 2.25 | 2.50 |
|---|---|---|---|---|---|---|---|---|---|---|---|
| MME (Gaussian) | 99.2 | 98.4 | 96.8 | 94.9 | 90.5 | 84.3 | 69.8 | 48.8 | 34.7 | 23.4 | 12.7 |
| GDL + MME (Gaussian) | 99.2 | 98.4 | 96.9 | 95.3 | 92.3 | 86.2 | 76.0 | **60.2** | 43.3 | 35.5 | 28.7 |
| CML + MME (Gaussian) | 99.3 | 98.4 | 97.0 | 95.0 | 90.8 | 84.0 | 71.1 | 50.0 | 36.7 | 24.5 | 13.7 |
| DRT + MME (Gaussian) | **99.5** | **98.6** | **97.5** | **95.5** | **92.6** | **86.8** | **76.5** | **60.2** | **43.9** | **36.0** | **29.1** |
| WE (Gaussian) | 99.2 | 98.4 | 96.9 | 94.9 | 90.6 | 84.5 | 70.4 | 49.0 | 35.2 | 23.7 | 12.9 |
| GDL + WE (Gaussian) | 99.3 | 98.5 | 97.1 | 95.3 | 92.3 | 86.3 | 76.3 | 43.4 | 35.1 | **29.0** |  |
| CML + WE (Gaussian) | 99.3 | 98.4 | 97.0 | 95.0 | 90.8 | 84.1 | 71.1 | 50.3 | 37.0 | 24.6 | 13.7 |
| DRT + WE (Gaussian) | **99.5** | **98.6** | **97.4** | **95.6** | **92.6** | **86.7** | **76.7** | **60.2** | **43.9** | **35.8** | **29.0** |

## G.2 CERTIFIED ROBUSTNESS OF SINGLE BASE MODEL WITHIN DRT-TRAINED ENSEMBLE

We also conduct ablation study on how the single base models' certified accuracy can be improved after applying DRT to the whole ensemble for both MNIST and CIFAR-10 datasets. Results are shown in in Table 14 and 15. We are surprised to find that the single base model within our DRT-trained ensemble are more robust compared to single base model within other baseline ensembles. Also, integrating them together could achieve higher robustness.

## G.3 OPTIMIZING THE WEIGHTS OF DRT-TRAINED ENSEMBLE

While we adapt average weights in our WE ensemble protocol in our experiments, we are also interested in how tuning the optimal weights could further improve the certified accuracy of our DRT-trained ensemble. We conduct this ablation study on both MNIST and CIFAR-10 datasets by grid-searching all the possible weights combination with step size as $0.1$. Results are shown in Table 16 and 17. (**AE** here refers to the average ensemble protocol and **WE** the weighted ensemble protocol by adapting the tuned optimal weights)

Table 13: Certified accuracy achieved by training with GD Loss (GDL) or Confidence Margin Loss (CML) only on CIFAR-10 dataset.

| Radius $r$ | 0.00 | 0.25 | 0.50 | 0.75 | 1.00 | 1.25 | 1.50 | 1.75 | 2.00 |
|---|---|---|---|---|---|---|---|---|---|
| MME (Gaussian) | 80.8 | 68.2 | 53.4 | 38.4 | 29.0 | 19.6 | 15.6 | 11.6 | 8.8 |
| GDL + MME (Gaussian) | 81.0 | 69.0 | 55.6 | 41.9 | 30.4 | 24.8 | 20.1 | 16.9 | 14.7 |
| CML + MME (Gaussian) | 81.2 | 69.4 | 54.4 | 39.6 | 29.2 | 21.6 | 17.0 | 13.1 | 12.8 |
| DRT + MME (Gaussian) | **81.4** | **70.4** | **57.8** | **43.8** | **34.4** | **29.6** | **24.9** | **20.9** | **16.6** |
| WE (Gaussian) | 80.8 | 68.4 | 53.6 | 38.4 | 29.2 | 19.7 | 15.9 | 11.8 | 8.9 |
| GDL + WE (Gaussian) | 81.0 | 69.1 | 55.6 | 41.8 | 30.6 | 25.2 | 20.2 | 16.9 | 14.9 |
| CML + WE (Gaussian) | 81.1 | 69.4 | 54.6 | 39.7 | 29.4 | 21.7 | 17.2 | 13.2 | 12.8 |
| DRT + WE (Gaussian) | **81.5** | **70.4** | **57.9** | **44.0** | **34.2** | **29.6** | **24.9** | **20.8** | **16.4** |

Table 14: Certified accuracy of single base model within DRT-trained ensemble on MNIST dataset.

| Radius $r$ | 0.00 | 0.25 | 0.50 | 0.75 | 1.00 | 1.25 | 1.50 | 1.75 | 2.00 | 2.25 | 2.50 |
|---|---|---|---|---|---|---|---|---|---|---|---|
| Single (Gaussian) | 99.1 | 97.9 | 96.6 | 94.7 | 90.0 | 83.0 | 68.2 | 46.6 | 33.0 | 20.5 | 11.5 |
| DRT Single (Gaussian) | 99.0 | **98.6** | 97.2 | 95.4 | 92.0 | 85.6 | 74.9 | 59.8 | 43.4 | 35.2 | 28.6 |
| DRT + MME (Gaussian) | **99.5** | **98.6** | **97.5** | 95.5 | **92.6** | **86.8** | 76.5 | **60.2** | **43.9** | **36.0** | **29.1** |
| DRT + WE (Gaussian) | **99.5** | **98.6** | 97.4 | **95.6** | **92.6** | 86.7 | **76.7** | **60.2** | **43.9** | 35.8 | 29.0 |
| Single (SmoothAdv) | 99.1 | 98.4 | 97.0 | 96.3 | 93.0 | 87.7 | 80.2 | 66.3 | 43.2 | 34.3 | 24.0 |
| DRT Single (SmoothAdv) | **99.2** | **98.4** | **97.6** | 96.6 | 92.9 | 88.1 | 80.4 | 68.0 | 46.4 | 39.2 | 34.1 |
| DRT + MME (SmoothAdv) | **99.2** | **98.4** | **97.6** | **96.7** | 93.1 | **88.5** | 83.2 | **68.9** | 48.2 | **40.3** | 34.7 |
| DRT + WE (SmoothAdv) | 99.1 | **98.4** | **97.6** | **96.7** | **93.4** | **88.5** | **83.3** | 69.6 | **48.3** | 40.2 | **34.8** |

Table 15: Certified accuracy of single base model within DRT-trained ensemble on CIFAR-10 dataset.

| Radius $r$ | 0.00 | 0.25 | 0.50 | 0.75 | 1.00 | 1.25 | 1.50 | 1.75 | 2.00 |
|---|---|---|---|---|---|---|---|---|---|
| Single (Gaussian) | 78.9 | 64.4 | 47.4 | 33.7 | 23.1 | 18.3 | 13.6 | 10.5 | 7.3 |
| DRT Single (Gaussian) | 81.4 | 69.8 | 56.2 | 42.5 | 33.6 | 27.6 | 24.2 | 20.4 | 15.4 |
| DRT + MME (Gaussian) | 81.4 | **70.4** | 57.8 | 43.8 | **34.4** | **29.6** | **24.9** | **20.9** | **16.6** |
| DRT + WE (Gaussian) | **81.5** | **70.4** | **57.9** | **44.0** | 34.2 | **29.6** | **24.9** | 20.8 | 16.4 |
| Single (SmoothAdv) | 68.9 | 61.0 | 54.4 | 45.7 | 34.8 | 28.5 | 21.9 | 18.2 | 15.7 |
| DRT Single (SmoothAdv) | 72.4 | 66.8 | 57.8 | 48.2 | 38.1 | 33.4 | 28.6 | 22.2 | 19.6 |
| DRT + MME (SmoothAdv) | **72.6** | **67.2** | **60.2** | 50.4 | 39.4 | 35.8 | **30.4** | 24.0 | 20.1 |
| DRT + WE (SmoothAdv) | **72.6** | 67.0 | **60.2** | 50.5 | 39.5 | 36.0 | 30.3 | **24.1** | 20.3 |

We can see that, by learning the optimal weights, the certified accuracy could be only slightly improved compared to the average weights setting, which indicates that, average weights can be a good choice in practice.

Table 16: Comparison of the certified accuracy between Average Ensemble (AE) protocol and Weighted Ensemble (WE) protocol on MNIST dataset. Cells with **bold** numbers indicate learning optimal weights could achieve higher certified accuracy on corresponding radius $r$.

| Radius $r$ | 0.00 | 0.25 | 0.50 | 0.75 | 1.00 | 1.25 | 1.50 | 1.75 | 2.00 | 2.25 | 2.50 |
|---|---|---|---|---|---|---|---|---|---|---|---|
| DRT + AE (Gaussian) | 99.5 | 98.6 | 97.4 | 95.6 | 92.6 | 86.7 | 76.7 | 60.2 | 43.9 | 35.8 | 29.0 |
| DRT + WE (Gaussian) | 99.5 | 98.6 | **97.6** | 95.6 | **92.7** | 86.8 | 76.7 | **60.3** | **44.0** | 36.0 | 29.3 |
| DRT + AE (SmoothAdv) | 99.1 | 98.4 | 97.6 | 96.7 | 93.4 | 88.5 | 83.3 | 69.6 | 48.3 | 40.2 | 34.8 |
| DRT + WE (SmoothAdv) | **99.1** | 98.4 | 97.6 | **96.8** | **93.5** | 88.5 | 83.3 | **69.7** | **48.5** | 40.2 | 34.8 |

### G.4 COMPARISON WITH OTHER GRADIENT DIVERSITY REGULARIZERS

We notice that out of the *certifiably* robust ensemble field, there exist two representatives of gradient diversity promoting regularizers: ADP (Pang et al., 2019) and GAL (Kariyappa & Qureshi, 2019). They achieved notable improvements on *empirical* ensemble robustness. For an ensemble consisting of base models $\{F_i\}_{i=1}^N$ and input $\boldsymbol{x}$ and ground truth label $y$, the ADP regularizer is defined as

$$\mathcal{L}_{\mathrm{ADP}}(\boldsymbol{x}, y) = \alpha \cdot \sum_{i=1}^{N} H(\mathrm{mean}(\{f_i(\boldsymbol{x})\})) + \beta \cdot \log(\mathbb{ED})$$

Table 17: Comparison of the certified accuracy between Average Ensemble (AE) protocol and Weighted Ensemble (WE) protocol on CIFAR-10 dataset. Cells with **bold** numbers indicate learning optimal weights could achieve higher certified accuracy on corresponding radius $r$.

| Radius $r$ | 0.00 | 0.25 | 0.50 | 0.75 | 1.00 | 1.25 | 1.50 | 1.75 | 2.00 |
|---|---|---|---|---|---|---|---|---|---|
| DRT + AE (Gaussian) | 81.5 | 70.4 | 57.9 | 44.0 | 34.2 | 29.6 | 24.9 | 20.8 | 16.4 |
| DRT + WE (Gaussian) | 81.5 | 70.4 | 57.9 | 44.0 | **34.3** | 29.6 | **25.0** | **20.9** | **16.5** |
| DRT + AE (SmoothAdv) | 72.6 | 67.0 | 60.2 | 50.5 | 39.5 | 36.0 | 30.3 | 24.1 | 20.3 |
| DRT + WE (SmoothAdv) | 72.6 | **67.1** | 60.2 | 50.5 | 39.5 | **36.1** | 30.3 | 24.1 | **20.4** |

Table 18: Certified accuracy of {ADP, GAL, DRT}-based Gaussian smoothed ensemble under different radii with WE protocol.

| **MNIST** $r$ | 0.00 | 0.25 | 0.50 | 0.75 | 1.00 | 1.25 | 1.50 | 1.75 | 2.00 | 2.25 | 2.50 |
|---|---|---|---|---|---|---|---|---|---|---|---|
| ADP | **99.5** | 98.2 | 97.2 | 95.2 | 92.2 | 85.8 | 73.4 | 53.2 | 36.9 | 24.7 | 13.3 |
| GAL | **99.5** | 98.3 | 97.2 | 95.1 | 92.4 | 86.1 | 73.2 | 54.4 | 36.2 | 24.7 | 13.9 |
| DRT | **99.5** | **98.6** | **97.4** | **95.6** | **92.6** | **86.7** | **76.7** | **60.2** | **43.9** | **35.8** | **29.0** |

| **CIFAR-10** $r$ | 0.00 | 0.25 | 0.50 | 0.75 | 1.00 | 1.25 | 1.50 | 1.75 | 2.00 |
|---|---|---|---|---|---|---|---|---|---|
| ADP | **83.0** | 68.0 | 52.2 | 38.2 | 28.8 | 20.0 | 16.8 | 14.2 | 11.0 |
| GAL | 82.2 | 67.6 | 53.6 | 38.8 | 27.6 | 20.2 | 15.4 | 13.6 | 10.6 |
| DRT | 81.5 | **70.4** | **57.9** | **44.0** | **34.2** | **29.6** | **24.9** | **20.8** | **16.4** |

where $H(\cdot)$ refers to the Shannon Entropy Loss function and $\mathbb{ED}$ the square of the spanned volume of base models' logit vectors.

GAL regularizer minimizes the cosine similarity value between base models' loss gradient vectors, which is defined as:

$$\mathcal{L}_{\text{GAL}}(\boldsymbol{x}, y) = \log \left( \sum_{1 \leq i < j \leq N} \exp \left( \cos \langle \nabla_{\boldsymbol{x}} \ell_{F_i}, \nabla_{\boldsymbol{x}} \ell_{F_j} \rangle \right) \right).$$

Under the smoothed ensemble training setting, the final training loss is represented by

$$\mathcal{L}_{\text{train}}(\boldsymbol{x}, y) = \sum_{i \in [N]} \mathcal{L}_{\text{std}}(\boldsymbol{x} + \varepsilon, y)_i + \{\mathcal{L}_{\text{ADP}}(\boldsymbol{x} + \varepsilon, y) \text{ or } \mathcal{L}_{\text{GAL}}(\boldsymbol{x} + \varepsilon, y)\}$$

where we consider standard training loss $\mathcal{L}_{\text{std}}(\boldsymbol{x}_0 + \varepsilon, y_0)_i$ of each base model $F_i$ to be the standard cross-entropy loss.

Table 18 shows the certified accuracy of {ADP, GAL, DRT}-trained ensemble under different radii with WE protocol on MNIST and CIFAR-10 dataset. We notice that DRT outperforms both ADP and GAL significantly in terms of the *certified* accuracy on different datasets.

