# OpenReview forum: "On the Certified Robustness for Ensemble Models and Beyond"
_ICLR.cc/2022/Conference — ICLR 2022 Poster_

### Official Review · Reviewer_VR8o · 2021-11-01

**Correctness:** 4
**Technical Novelty And Significance:** 3
**Empirical Novelty And Significance:** 2
**Recommendation:** 6
**Confidence:** 3

**Main Review:**

The ensemble model performs well in empirical defense, so its extension to provable robustness is essential. Some interesting insights are presented, but they also seem to be readily available from the empirical ensemble model defense.

In definition 1, To be consistent with randomized smoothing, should ${\left\|\delta\right\|}_2 \leqslant r$ be ${\left\|\delta\right\|}_2 < r$?

Why is the empirical robust accuracy of DRT $(r=0)$ lower than the baseline on large-scale datasets?

What is the motivation for Ensemble-before-Smoothing? Is it driven by the property of Theorem II or is it due to empirical observation? The EBS here seems trivial.

I will consider raising my score based on the other reviewers' comments and the authors' responses.



**Summary Of The Paper:**

The paper analyzed the certified robustness for ensemble ML models. It is proven that diversified gradient and large confidence margin are sufficient and necessary conditions for certifiably robust ensemble models under the model-smoothness assumption. The proposed Diversity Regularized Training (DRT) method performs better than current provable methods.



**Summary Of The Review:**

The paper has a lot of proof, which is great, but there are some questions about the experimental results and motivation that need to be clarified.

---

> ### Author Response · Authors · 2021-11-20
> **Response to Reviewer VR8o**
>
> We thank the reviewer for the valuable comments, and we address your questions below.
>
> > “In definition 1, To be consistent with randomized smoothing, should $|\delta|_2\le r$ be $|\delta|_2<r$?”
>
> Thanks for pointing this out. This is indeed a typo and we have fixed this in our revision (see Definition 1 on Page 3).
>
> > “Why is the empirical robust accuracy of DRT $(r=0)$ lower than the baseline on large-scale datasets?”
>
> Sorry for the confusion. $r=0$ column represents the accuracy on clean data (benign accuracy) and we did not evaluate the empirical robust accuracy following the prior works. The decrease of benign accuracy is due to the regularization on base model gradients vectors which aim to promote diversity during the DRT training. We remark that such decrease of benign accuracy shows the standard tradeoff between robustness and benign performance, which is also demonstrated in other certified robust training methods (e.g., SmoothAdv) and empirical robust training methods (e.g., TRS [1]).
>
> > “What is the motivation for Ensemble-before-Smoothing? Is it driven by the property of Theorem II or is it due to empirical observation? The EBS here seems trivial.”
>
> Thanks for the insightful question. At high level, typically neural networks are usually nonsmooth or have only coarse smoothness bounds, i.e., $\beta$ is large. Therefore, directly applying Theorem 1 for normal nonsmooth models would lead to near-zero certified radius. As a result, we first propose soft smoothing to enforce the smoothness of base models as shown in Theorem 2. Based on soft smoothing, in Equation 7, we can directly prove that GD and CM losses boost certified robustness. (We do not consider “hard smoothing”, i.e., directly applying randomized smoothing on base models, because that does not reveal a differentiable objective as Eqn. (7) on which we can apply regularization.) However, with such soft smoothed base models, directly applying Theorem 1 to certify robustness is still practically challenging, since the LHS of Equation (4) and (5) involves gradient of the soft smoothed confidence. A precise computation of such gradient requires high-confidence estimation of high-dimensional vectors via sampling, which requires linear number of samples with respect to input dimension and is thus too expensive in practice. To solve this issue, we then propose EBS as the practical smoothing protocol, which serves as an approximation of soft smoothing to realize the effectiveness of GD and CM losses and enable the randomized smoothing based techniques to certify the robustness.
>
> Though the EBS protocol itself seems trivial, it serves as a very important strategy driven by our theoretical analysis to approximate the ensemble smoothness and therefore enable the certification of the ensemble via randomized smoothing based techniques.
> In our revision, we have rewritten Section 2.3 thoroughly to make this logic clear and motivate the EBS explicitly.
>
> We sincerely thank the reviewer for the insightful and helpful comments, and if the reviewer is satisfied with our answers please help to increase the score. Please also let us know if there are other questions, and we really look forward to the discussion with the reviewer to further improve our paper. Thank you!!
>
> [1] Yang, Zhuolin, Linyi Li, Xiaojun Xu, Shiliang Zuo, Qian Chen, Benjamin Rubinstein, Ce Zhang, and Bo Li. TRS: Transferability Reduced Ensemble via Encouraging Gradient Diversity and Model Smoothness. Advances in Neural Information Processing Systems, 2021.
>
> [2] Hadi Salman, Jerry Li, Ilya Razenshteyn, Pengchuan Zhang, Huan Zhang, Sebastien Bubeck, and Greg Yang.   Provably robust deep learning via adversarially trained smoothed classifiers.  Advances in Neural Information Processing Systems, pp. 11292–11303, 2019.
>
> [3] Runtian Zhai, Chen Dan, Di He, Huan Zhang, Boqing Gong, Pradeep Ravikumar, Cho-Jui Hsieh, and Liwei Wang. Macer: Attack-free and scalable robust training via maximizing certified radius. International Conference on Learning Representations, 2019.
>
> [4] Aounon Kumar, Alexander Levine, Soheil Feizi, and Tom Goldstein.  Certifying confidence via randomized smoothing. Advances in Neural Information Processing Systems, pp. 5165–5177, 2020.

---

> > ### Comment · Reviewer_VR8o · 2021-11-21
> > **Questions about experiments on ImageNet.**
> >
> > The response addresses most of my concerns.
> >
> > The second question is actually trying to convey that the performance of DRT in Table 3 is inconsistent with that of Table 1 and Table 2. I would like the authors to discuss this.

---

> > > ### Author Response · Authors · 2021-11-22
> > > **Response for Remaining Concerns**
> > >
> > > We thank the reviewer for the prompt reply and the valuable following question!
> > >
> > > Indeed we believe there is still a tradeoff between the benign accuracy and certified robustness on DRT, which is mainly controlled by the weights of the robustness regularizations $\rho_1$ and $\rho_2$. In particular, large $\rho_1$ and $\rho_2$ would lead to higher certified accuracy on large radii but lower benign accuracy, since large $\rho_1$ and $\rho_2$ would emphasize the robustness regularization more (which is also mentioned in Appendix F "Effects of $\rho_1$ and $\rho_2$").
> > >
> > > In Table 1 and 2, we choose relatively small $\rho_1$ and $\rho_2$ given the less challenging MNIST and CIFAR-10 datasets, and therefore the benign accuracy is preserved and even slightly improved potentially due to the large margin regularization. However for Table 3, on the more challenging ImageNet dataset, we have to select large $\rho_1$ and $\rho_2$ to achieve high certified robustness on large radii, which therefore hurts the benign performance more obviously. We have included such discussion in Appendix F.3 in our revision, and thanks for the helpful suggestions!

---

### Official Review · Reviewer_f6eQ · 2021-11-01

**Correctness:** 4
**Technical Novelty And Significance:** 4
**Empirical Novelty And Significance:** 4
**Recommendation:** 8
**Confidence:** 3

**Main Review:**

### Strengths
* The empirical results are strong, i.e., the proposed model outperforms baselines consistently.
* The theoretical contributions appear sound and meaningful.
* The paper is well-written and easy to follow.
### Weaknesses
* The authors only study a single setting of $N=3$. An interesting study would have been to see how increasing $N$ changes the certified robustness.
* The authors do not make use of the fact that their theory allows to certify when the ensemble *cannot* be robust w.r.t. a sample.
* The proposed method has the apparent drawback that the ensemble models cannot be trained in isolation.

### Detailed comments
* An interesting finding is the importance of gradient diversity for the ensemble robustness. This seems to align well with the fact that we favor diverse ensembles in practice. I wonder whether there is a deeper connection of gradient diversity for robustness to the diversity of the ensemble.
* Out of curiosity, is there any theoretical indication that larger $N$ leads to more robustness? Intuitively, large $N$ should lead to a smoother ensemble function, but $N$ does not appear in the computation of the smoothness $\beta$.
* How are the $w_i$ optimized in the proposed approach?
* An interesting aspect of the theoretical contribution is that Theorem 1 can also prove when a sample is *not* robust. It would have been interesting to show the share of *certifiably nonrobust* samples in addition to the certified accuracy in the figures and tables. This way we could also see how large the share of samples is for which we cannot certify (non-)robustness.

**Summary Of The Paper:**

The authors present theoretical and empirical findings on how ensembles can improve certifiable robustness of classifiers. The authors show that increased robustness requires diverse gradients among the ensemble members (as well as large confidence margins). The authors incorporate these findings into two regularization terms for training and show that these lead to improved certifiable robustness on MNIST, CIFAR10, and ImageNet.

**Summary Of The Review:**

The paper is well-written and easy to follow, and the theoretical and empirical contributions are interesting and appear meaningful. I would have liked to see different values of $N$ and a study of the *certifiable non-robustness* of the ensemble models.

Edit: I have increased my score based on the authors' response.

---

> ### Author Response · Authors · 2021-11-20
> **Response to Reviewer f6eQ (1/2)**
>
> Thanks for your inspiring comments and appreciation of our work. We answer the questions below.
>
> > “I wonder whether there is a deeper connection of gradient diversity for robustness to the diversity of the ensemble.”
>
> Thanks for the insightful question. We believe there is indeed a deep connection between gradient diversity for robustness to the diversity of the ensemble. For instance, our work promotes gradient diversity together with enforcing large margins to prove the sufficient / necessary conditions of ensemble robustness.
>
> Prior works [1] proves that gradient diversity together with smoothed base models would lead to diverse ensemble (aka, low adversarial transferability between base models). Other related work [2,3] empirically verifies that enforcing gradient diversity would lead to more diverse ensemble which achieves low adversarial transferability between base models.
>
> Thus, we believe gradient diversity serves as one necessary condition for diversity of the ensemble, but there are other important necessary conditions as well such as base model smoothness and large margin.
>
> [1] Yang, Zhuolin, Linyi Li, Xiaojun Xu, Shiliang Zuo, Qian Chen, Benjamin Rubinstein, Ce Zhang, and Bo Li. TRS: Transferability Reduced Ensemble via Encouraging Gradient Diversity and Model Smoothness. Advances in Neural Information Processing Systems, 2021.
>
> [2] Huanrui Yang, Jingyang Zhang, Hongliang Dong, Nathan Inkawhich, Andrew Gardner, Andrew Touchet, Wesley Wilkes, Heath Berry and Hai Li. DVERGE: Diversifying Vulnerabilities for Enhanced Robust Generation of Ensembles Advances in Neural Information Processing Systems, 2020.
>
> [3] Sanjay Kariyappa, Moinuddin K. Qureshi. Improving Adversarial Robustness of Ensembles with Diversity Training. arXiv:1901.09981
>
> > “An interesting study would have been to see how increasing $N$ changes the certified robustness.”
>
> Thanks for the valuable suggestion. We follow the suggestion and evaluated the certified accuracy of our DRT-enhanced ensemble with $N=4, 5$ on MNIST and the results are shown below.
>
>
> | Radius $r$ / MNIST               | 0.00 | 0.25 |   0.50   | 0.75 |   1.00   |   1.25   |   1.50   |   1.75   |   2.00   |   2.25   |   2.50   |
> | -------------------------------- | :--: | :--: | :------: | :--: | :------: | :------: | :------: | :------: | :------: | :------: | :------: |
> | DRT + MME / WE ($n=3$, Gaussian) | 99.5 | 98.6 |   97.5   | 95.6 |   92.6   | **86.8** | **76.7** |   60.2   |   43.9   |   36.0   |   29.1   |
> | DRT + MME / WE ($n=4$, Gaussian) | 99.5 | 98.6 | **97.6** | 95.6 | **92.7** |   86.7   |   76.4   |   60.5   |   44.1   |   36.0   |   29.2   |
> | DRT + MME / WE ($n=5$, Gaussian) | 99.5 | 98.6 |   97.5   | 95.4 |   92.4   |   86.5   |   76.5   | **60.8** | **44.3** | **36.2** | **29.5** |
>
> We can see that increasing $N$ will improve the certified accuracy slightly on some radii (especially on larger $r$). Considering the training time would increase in $N$, we view $N=3$ as a good balance between the certified robustness and computational cost.
>
> > “Is there any theoretical indication that larger $N$ leads to more robustness? Intuitively, large $N$ should lead to a smoother ensemble function, but $N$ does not appear in the computation of the smoothness $\beta$.”
>
> Thanks for the insightful comment and yes the reviewer’s intuition is correct that large $N$ leads to more robustness.
>
> Inspired by the reviewer, in Appendix B.1, we add a new proposition (Proposition B.1) that theoretically shows larger $N$ leads to better certified robustness by making the gradients “more diverse”, i.e., the joint gradient magnitude in LHS of Equation 4 is relatively smaller than the RHS when $N$ grows larger and thus the sufficient condition of robustness under larger $r$ becomes easier to be satisfied.
>
> Note that such theoretical results do not require to show that larger $N$ leads to a smoother ensemble, though it is intuitively true. Concretely, it would require additional assumptions on the second-order curvature for diversity of base models to analyze the smoothness changes; while in Appendix B.1, in our revision, we directly prove that larger $N$ leads to more robustness.

---

> > ### Comment · Reviewer_f6eQ · 2021-11-24
> > **Thank you**
> >
> > Thank you for your response. It addresses most of my concerns. I have adapted my score accordingly.
> >
> > Some follow-up thoughts:
> > * I would have expected to see stronger effect of robustness by increasing $N$. Do you have an explanation why the effect appears to be so small?
> > * It would be great to see a small note in the final version that mentions the (theoretical) possibility of certifying non-robustness, as this could inspire future research.

---

> > > ### Author Response · Authors · 2021-11-24
> > > **Response for follow-up concerns**
> > >
> > > Thanks for your insightful follow-up concerns. We reply to your further suggestions here.
> > >
> > > > "I would have expected to see stronger effect of robustness by increasing $N$. Do you have an explanation why the effect appears to be so small?"
> > >
> > > Thanks for your insightful comment! We believe the reason why the effects of increasing $N$ is not significant is the decrement of marginal utilities on promoting diversity along with the increase of $N$. If we have already achieved low LHS value (norm of the aggregated gradient vector) in Eqn. (4) (5) by promoting gradient diversity when $N=3$, larger $N = 4, 5$ may not significantly help to decrease the LHS value further, and thus lead to marginal improvement on robustness.
> > >
> > > Another possible reason could be that more base models will lead to more model training parameters. We may need longer training epochs or better training scheduler to find a better local optima which may or may not exist. We will add these discussions in our revision.
> > >
> > > > "It would be great to see a small note in the final version that mentions the (theoretical) possibility of certifying non-robustness, as this could inspire future research."
> > >
> > > Thanks for the very interesting and valuable suggestion. Indeed, there is no certified non-robustness work yet to our best knowledge and it would be a very interesting direction. Currently, we provide both sufficient and necessary conditions for the certified robustness of the ensemble, which shows a possible way to further explore the non-robustness certification. We will definitely add related discussions in our final version and thank you for such a brilliant suggestion!!

---

> ### Author Response · Authors · 2021-11-20
> **Response to Reviewer f6eQ (2/2)**
>
> > “The ensemble models cannot be trained in isolation.”
>
> Thanks for pointing this out, and we will note this in our paper as well. In particular, since the gradient loss term needs to be computed across every pair of base models, it could be challenging to decompose it, but the confidence margin loss term for every base model can be separated and trained in isolation in practice.
>
> > “How are the $w_i$ optimized in the proposed approach?”
>
> To make more fair comparison with other baseline ensemble approaches, in this paper’s results, we view all the $w_i$​ to be the same in our experiments (i.e., average aggregation protocol)​. We have also made this clear as highlighted in Section 4.1 in our revision.
> Following the suggestion, we also conducted additional experiments to study how different weights could affect the robustness by learning the optimal weights of the trained base models of our DRT ensemble. Results are shown in the following tables (**AE** refers to the average ensemble protocol and **WE** the weighted ensemble protocol), and we have also included these results in Appendix G.2 in our revision.
>
>
>
> |  Radius $r$ / MNIST  | 0.00 | 0.25 |   0.50   |   0.75   |   1.00   |   1.25   | 1.50 |   1.75   |   2.00   |   2.25   |   2.50   |
> | :------------------: | :--: | :--: | :------: | :------: | :------: | :------: | :--: | :------: | :------: | :------: | :------: |
> | DRT + AE (Gaussian)  | 99.5 | 98.6 |   97.4   |   95.6   |   92.6   |   86.7   | 76.7 |   60.2   |   43.9   |   35.8   |   29.0   |
> | DRT + WE (Gaussian)  | 99.5 | 98.6 | **97.6** |   95.6   | **92.7** | **86.8** | 76.7 | **60.3** | **44.0** | **36.0** | **29.3** |
> |                      |      |      |          |          |          |          |      |          |          |          |          |
> | DRT + AE (SmoothAdv) | 99.1 | 98.4 |   97.6   |   96.7   |   93.4   |   88.5   | 83.3 |   69.6   |   48.3   |   40.2   |   34.8   |
> | DRT + WE (SmoothAdv) | 99.1 | 98.4 |   97.6   | **96.8** | **93.5** |   88.5   | 83.3 | **69.7** | **48.5** |   40.2   |   34.8   |
>
> | Radius $r$ / CIFAR-10 | 0.00 |   0.25   | 0.50 | 0.75 |   1.00   |   1.25   |   1.50   |   1.75   |   2.00   |
> | --------------------- | :--: | :------: | :--: | :--: | :------: | :------: | :------: | :------: | :------: |
> | DRT + AE (Gaussian)   | 81.5 |   70.4   | 57.9 | 44.0 |   34.2   |   29.6   |   24.9   |   20.8   |   16.4   |
> | DRT + WE (Gaussian)   | 81.5 |   70.4   | 57.9 | 44.0 | **34.3** |   29.6   | **25.0** | **20.9** | **16.5** |
> |                       |      |          |      |      |          |          |          |          |          |
> | DRT + AE (SmoothAdv)  | 72.6 |   67.0   | 60.2 | 50.5 |   39.5   |   36.0   |   30.3   |   24.1   |   20.3   |
> | DRT + WE (SmoothAdv)  | 72.6 | **67.1** | 60.2 | 50.5 |   39.5   | **36.1** |   30.3   |   24.1   | **20.4** |
>
> We can see that, by learning the optimal weights, the certified accuracy could be slightly improved compared to the equal-weighted setting in some cases and we have added related discussion in our revision.
>
> > “It would have been interesting to show the share of certifiably nonrobust samples in addition to the certified accuracy in the figures and tables.”
>
> Thanks to the reviewer for raising this interesting point! Yes, it is theoretically possible to compute whether the sample is certifiably nonrobust. However, in practice, to leverage Theorem 1 for certification, we need to have a high-confidence estimation of LHS in Equations (4) and (5) which is the $L_2$ norm of the gradient vector. This reduces to the problem of high-confidence estimation of high-dimensional vectors via sampling since it is the gradient vector of Gaussian smoothed model. Such estimation requires a linear number of samples w.r.t. input dimension, which hinders the practicality of certifying nonrobust samples in a short time. We have added these interesting discussions in our revision (see the first paragraph of Section 2.3). Therefore, to efficiently certify the nonrobustness based on our analysis would be an interesting future work.

---

### Official Review · Reviewer_AZhe · 2021-11-01

**Correctness:** 3
**Technical Novelty And Significance:** 3
**Empirical Novelty And Significance:** 4
**Recommendation:** 6
**Confidence:** 4

**Main Review:**

- The paper is clearly written and well-motivated. The motivation is theoretically supported, and the strong experimental results also confirm the effectiveness. I agree that exploring on the ensemble of smoothed classifiers is an important yet under-explored topic.
- Although the paper motivates the method from Theorem 1, it was quite unclear to me when the paper proceeds from Theorem 1 to Ensemble-before-Smoothing (EBS): I can see that Theorem 1 holds only when all the base classifiers ("before" ensemble) are $\beta$-smooth, but then why Theorem 2 shows the $\beta$-smoothness of the classifier "after" ensemble? Does EBS guarantee that all the base classifiers before ensemble are $\beta$-smooth? If not, how can one apply Theorem 1 to justify GD and CM losses?
- I feel the paper could benefit from comparing Average Certified Radius (ACR) [Zhai et al., 2020] as well as the certified accuracies: overall, I could see that the proposed method improves certified accuracy, but generally the gains are not uniform (especially for the ImageNet results) so that the readers may be confused on interpreting them.
- It would be helpful for the readers that the Table 1 and 2 could also present the single-model performance of DRT-trained models.
- The paper should include a discussion about the computational costs of the proposed method - e.g., the increase in training time of DRT compared to the Gaussian training (or standard ensemble). Also, I would like to see the overheads in certification time, as the certification pipeline now includes an ensemble during smoothing.
- I feel the results on ImageNet are relatively less significant, especially given the decrease in the clean accuracy of DRT+* models. The paper could include more discussion on it.

**Summary Of The Paper:**

The paper proposes to train diverse classifiers to improve certified robustness of ensemble classifiers. More specifically, it proposes two regularization terms in training: (a) the Gradient Diversity (GD) loss, and (b) the Confidence Margin (CM) loss. This is motivated by a theoretical observation that one needs both (a) and (b) to achieve a robust ensemble model given that all the base classifiers are smooth. Experimental results on MNIST, CIFAR-10 and ImageNet show that the proposed training can consistently improve the certified accuracies of ensemble models compared to the ensemble of classifiers from the standard training.

**Summary Of The Review:**

Overall, I can see that the paper is clearly written and addresses an important research direction that explores a better ensemble scheme of smoothed classifiers. Although I feel a slight logical gap in Theorem 1 → EBS, the paper presents a novel and theoretical justification to motivate the new training method, and it is further supported with a strong empirical results.

---

> ### Author Response · Authors · 2021-11-20
> **Response to Reviewer AZhe (1/3)**
>
> Thanks for your constructive comments on this work. Here are the answers to your major concerns.
>
> > “Unclear to me when the paper proceeds from Theorem 1 to Ensemble-before-Smoothing (EBS):  I can see that Theorem 1 holds only when all the base classifiers ("before" ensemble) are $\beta$-smooth, but then why Theorem 2 shows the $\beta$-smoothness of the classifier "after" ensemble? Does EBS guarantee that all the base classifiers before ensemble are $\beta$-smooth? If not, how can one apply Theorem 1 to justify GD and CM losses?”
>
> Thanks for the insightful comment! Theorem 2 shows the $\beta$-smoothness of the classifier “before” ensemble, i.e., smoothness of $\bar g^\varepsilon_f$ where $f$ is a base model, instead of smoothness of “after” ensemble. EBS approximately guarantees that all the base classifiers before ensemble are $\beta$-smooth, since EBS is an effective approximation for soft smoothing, and soft smoothing ensures that base models (“before” ensemble) are $\beta$-smooth.
>
> We now explain in detail about our logic for Theorem 1 -> EBS: Typically neural networks are nonsmooth or only have coarse smoothness bounds, i.e., $\beta$ is large. Therefore, directly applying Theorem 1 for normal nonsmooth models would lead to near-zero certified radius. As a result, we first propose soft smoothing to enforce the smoothness of base models as shown in the second paragraph of Section 2.3. We then prove that regularizing Eqn. (7) can improve the certified robustness of soft smoothed ensembles. (We cannot directly apply “hard smoothing”, i.e., directly applying randomized smoothing on base models, since it does not reveal a differentiable objective as Eqn. (7) to which we can apply the regularization.) However, with such soft smoothed base models, applying Theorem 1 to certify its robustness is still *practically* challenging, since the LHS of Equation (4) and (5) involves gradient of the soft smoothed confidence. A precise computation of such gradient requires high-confidence estimation of high-dimensional vectors via sampling, which requires linear number of samples with respect to the input dimension and is thus too expensive in practice. To solve this issue, we then propose to apply EBS as the practical smoothing protocol, which serves as an approximation of soft smoothing to realize the effectiveness of GD and CM losses and enable the randomized smoothing based techniques to certify the robustness.
>
> In our revision, we have rewritten Section 2.3 thoroughly to make this logic clear and fill the gap between Theorem 1 and EBS. Such approximation is also adapted by previous work [1,2,3] and shown effective in analyzing smoothed models.
>
> [1] Hadi Salman, Jerry Li, Ilya Razenshteyn, Pengchuan Zhang, Huan Zhang, Sebastien Bubeck, and Greg Yang.   Provably robust deep learning via adversarially trained smoothed classifiers.   In Advances in Neural Information Processing Systems, pp. 11292–11303, 2019.
>
> [2] Runtian Zhai, Chen Dan, Di He, Huan Zhang, Boqing Gong, Pradeep Ravikumar, Cho-Jui Hsieh, and Liwei Wang. Macer: Attack-free and scalable robust training via maximizing certified radius. In International Conference on Learning Representations, 2019.
>
> [3] Aounon Kumar, Alexander Levine, Soheil Feizi, and Tom Goldstein.  Certifying confidence via randomized smoothing. Advances in Neural Information Processing Systems, volume 33, pp. 5165–5177.

---

> ### Author Response · Authors · 2021-11-20
> **Response to Reviewer AZhe (2/3)**
>
>
> > “The paper could benefit from comparing Average Certified Radius (ACR) [Zhai et al., 2020] as well as the certified accuracies.”
>
> Thanks for the valuable suggestion! We evaluate the ACR on all our models following the suggestions and the results with brief summary are shown below.
>
> | MNIST                      | $\sigma=0.25$ | $\sigma=0.50$ | $\sigma=1.00$ |
> | -------------------------- | :-----------: | :-----------: | :-----------: |
> | Gaussian                   |     0.912     |     1.565     |     1.633     |
> | SmoothAdv                  |     0.920     |     1.629     |     1.734     |
> | MACER                      |     0.918     |     1.583     |     1.520     |
> | MME / WE (Gaussian)        |     0.915     |     1.585     |     1.669     |
> | DRT + MME / WE (Gaussian)  |     0.923     |     1.637     |     1.745     |
> | MME / WE (SmoothAdv)       |     0.926     |     1.678     |     1.765     |
> | DRT + MME / WE (SmoothAdv) |   **0.929**   |   **1.689**   |   **1.812**   |
>
> | CIFAR-10                   | $\sigma=0.25$ | $\sigma=0.50$ | $\sigma=1.00$ |
> | -------------------------- | :-----------: | :-----------: | :-----------: |
> | Gaussian                   |     0.484     |     0.595     |     0.559     |
> | SmoothAdv                  |     0.539     |     0.662     |     0.730     |
> | MACER                      |   **0.556**   |     0.726     |     0.792     |
> | MME / WE (Gaussian)        |     0.513     |     0.621     |     0.579     |
> | DRT + MME / WE (Gaussian)  |     0.551     |     0.687     |     0.744     |
> | MME / WE (SmoothAdv)       |     0.542     |     0.692     |     0.689     |
> | DRT + MME / WE (SmoothAdv) |     0.545     |   **0.760**   |   **0.868**   |
>
> We can see that our DRT-enhanced ensemble could still achieve the highest ACR on almost all the settings. Especially on $\sigma = 1.00$, our improvement is significant. We have also updated ACR as our new evaluation metric in Appendix F in our version.
>
> > “It would be helpful for the readers that the Table 1 and 2 could also present the single-model performance of DRT-trained models.”
>
> Thanks for the valuable suggestion! We evaluated the single-model performance of our DRT-trained models on MNIST and CIFAR-10 datasets. Results are shown below.
>
>
>
> | Radius $r$ / MNIST     | 0.00 | 0.25 | 0.50 | 0.75 | 1.00 | 1.25 | 1.50 | 1.75 | 2.00 | 2.25 | 2.50 |
> | ---------------------- | :--: | :--: | :--: | :--: | :--: | :--: | :--: | :--: | :--: | :--: | :--: |
> | Single (Gaussian)      | 99.1 | 97.9 | 96.6 | 94.7 | 90.0 | 83.0 | 68.2 | 46.6 | 33.0 | 20.5 | 11.5 |
> | DRT Single (Gaussian)  | 99.0 | 98.6 | 97.2 | 95.4 | 92.0 | 85.6 | 74.9 | 59.8 | 43.4 | 35.2 | 28.6 |
> | DRT + MME (Gaussian)   | 99.5 | 98.6 | 97.5 | 95.5 | 92.6 | 86.8 | 76.5 | 60.2 | 43.9 | 36.0 | 29.1 |
> | DRT + WE (Gaussian)    | 99.5 | 98.6 | 97.4 | 95.6 | 92.6 | 86.7 | 76.7 | 60.2 | 43.9 | 35.8 | 29.0 |
> | Single (SmoothAdv)     | 99.1 | 98.4 | 97.0 | 96.3 | 93.0 | 87.7 | 80.2 | 66.3 | 43.2 | 34.3 | 24.0 |
> | DRT Single (SmoothAdv) | 99.2 | 98.4 | 97.6 | 96.6 | 92.9 | 88.1 | 80.4 | 68.0 | 46.4 | 39.2 | 34.1 |
> | DRT + MME (SmoothAdv)  | 99.2 | 98.4 | 97.6 | 96.7 | 93.1 | 88.5 | 83.2 | 68.9 | 48.2 | 40.3 | 34.7 |
> | DRT + WE (SmoothAdv)   | 99.1 | 98.4 | 97.6 | 96.7 | 93.4 | 88.5 | 83.3 | 69.6 | 48.3 | 40.2 | 34.8 |
>
> | Radius $r$ / CIFAR-10  | 0.00 | 0.25 | 0.50 | 0.75 | 1.00 | 1.25 | 1.50 | 1.75 | 2.00 |
> | ---------------------- | :--: | :--: | :--: | :--: | :--: | :--: | :--: | :--: | :--: |
> | Single (Gaussian)      | 78.9 | 64.4 | 47.4 | 33.7 | 23.1 | 18.3 | 13.6 | 10.5 | 7.3  |
> | DRT Single (Gaussian)  | 81.4 | 69.8 | 56.2 | 42.5 | 33.6 | 27.6 | 24.2 | 20.4 | 15.4 |
> | DRT + MME (Gaussian)   | 81.4 | 70.4 | 57.8 | 43.8 | 34.4 | 29.6 | 24.9 | 20.9 | 16.6 |
> | DRT + WE (Gaussian)    | 81.5 | 70.4 | 57.9 | 44.0 | 34.2 | 29.6 | 24.9 | 20.8 | 16.4 |
> | Single (SmoothAdv)     | 68.9 | 61.0 | 54.4 | 45.7 | 34.8 | 28.5 | 21.9 | 18.2 | 15.7 |
> | DRT Single (SmoothAdv) | 72.4 | 66.8 | 57.8 | 48.2 | 38.1 | 33.4 | 28.6 | 22.2 | 19.6 |
> | DRT + MME (SmoothAdv)  | 72.6 | 67.2 | 60.2 | 50.4 | 39.4 | 35.8 | 30.4 | 24.0 | 20.1 |
> | DRT + WE (SmoothAdv)   | 72.6 | 67.0 | 60.2 | 50.5 | 39.5 | 36.0 | 30.3 | 24.1 | 20.3 |
>
> We can see that the single base models within our DRT-trained ensemble are also more robust compared to the base models in other baselines. We view this as an interesting byproduct of the proposed DRT training approach. We have also included these results in our revision in Appendix G.3.

---

> ### Author Response · Authors · 2021-11-20
> **Response to Reviewer AZhe (3/3)**
>
> > “The paper should include a discussion about the computational costs of the proposed method - e.g., the increase in training time of DRT compared to the Gaussian training (or standard ensemble). Also, I would like to see the overheads in certification time, as the certification pipeline now includes an ensemble during smoothing.”
>
> Thanks for the suggestion. In our paper, we made a brief discussion on the computational costs of our DRT during the training in Appendix E: Given $N$ as the number of base models, the DRT training would need to compute $O(N^2)​$ training loss terms (iterating all valid pairs) while the standard Gaussian training would have $O(N)$ training loss terms. We also evaluate the training time per minibatch in subsections in Appendix F with paragraph “Efficiency Analysis”.  We have rewritten the sentence on the top of Page 9 to make such computational cost discussion more clear.
>
> For certification time, both MME and WE protocols need $O(N)$ certification times per sample, which is the same as other ensemble-based methods.
>
> > “I feel the results on ImageNet are relatively less significant, especially given the decrease in the clean accuracy of DRT+* models.”
>
> Thanks for the nice observation. We should note that our DRT-enhanced models could achieve significant improvement on large radii. On small radii, DRT achieves relatively less significant improvements, which we believe is mainly due to the limited hyper-parameter tuning since such cases are less useful (adversarial) than those under large radii and thus we did not perform hyper-parameter tuning for such cases. We believe that the supreme performance of DRT under large radii would demonstrate its effectiveness.

---

### Official Review · Reviewer_ug5z · 2021-11-03

**Correctness:** 3
**Technical Novelty And Significance:** 3
**Empirical Novelty And Significance:** 4
**Recommendation:** 6
**Confidence:** 4

**Main Review:**

The theoretical analysis in this paper is correct, although a bit difficult to follow due to the redundant and non-traditional notations. The experiment results are sufficient to support the claim of this paper.

One question I have is how the weights of ensembles could affect the results. I didn’t find any explicit values for these weights, which should be included in the experiment section as what are the values and how these are chosen.

**Summary Of The Paper:**

This paper proposes a new analysis for randomized smoothing based certified robustness of ensemble models. The analysis shows that the robustness radius depends on the l2 norm of the weighted sum of the gradients of the confidence margin of base models. A training strategy is then proposed to regularize this term by increasing the divergence of gradient vectors.

**Summary Of The Review:**

See comments above.

---

> ### Author Response · Authors · 2021-11-20
> **Response to Reviewer ug5z**
>
> We thank the reviewer’s appreciation for our work, and we address the questions below.
>
> > “How the weights of ensembles could affect the results?”
>
> Thanks for the insightful question and suggestion. In the paper, to directly demonstrate the effectiveness of the proposed training method, we use the same weights for different base models (i.e., use average ensemble). We also add a sentence in our revision in Section 4.1 to make this clear following the suggestion.
>
> In addition, based on the comment, we conducted more experiments to study how weights affect the results by learning the optimal weights of the trained base models of DRT enhanced ensemble. Results are shown in the following tables (**AE** refers to the equal-weighted average ensemble protocol used by default in other experiments; and **WE** the weighted ensemble protocol), and we have also included these two tables in Appendix G.3.
>
>
>
> |  Radius $r$ / MNIST  | 0.00 | 0.25 |   0.50   |   0.75   |   1.00   |   1.25   | 1.50 |   1.75   |   2.00   |   2.25   |   2.50   |
> | :------------------: | :--: | :--: | :------: | :------: | :------: | :------: | :--: | :------: | :------: | :------: | :------: |
> | DRT + AE (Gaussian)  | 99.5 | 98.6 |   97.4   |   95.6   |   92.6   |   86.7   | 76.7 |   60.2   |   43.9   |   35.8   |   29.0   |
> | DRT + WE (Gaussian)  | 99.5 | 98.6 | **97.6** |   95.6   | **92.7** | **86.8** | 76.7 | **60.3** | **44.0** | **36.0** | **29.3** |
> |                      |      |      |          |          |          |          |      |          |          |          |          |
> | DRT + AE (SmoothAdv) | 99.1 | 98.4 |   97.6   |   96.7   |   93.4   |   88.5   | 83.3 |   69.6   |   48.3   |   40.2   |   34.8   |
> | DRT + WE (SmoothAdv) | 99.1 | 98.4 |   97.6   | **96.8** | **93.5** |   88.5   | 83.3 | **69.7** | **48.5** |   40.2   |   34.8   |
>
> | Radius $r$ / CIFAR-10 | 0.00 |   0.25   | 0.50 | 0.75 |   1.00   |   1.25   |   1.50   |   1.75   |   2.00   |
> | --------------------- | :--: | :------: | :--: | :--: | :------: | :------: | :------: | :------: | :------: |
> | DRT + AE (Gaussian)   | 81.5 |   70.4   | 57.9 | 44.0 |   34.2   |   29.6   |   24.9   |   20.8   |   16.4   |
> | DRT + WE (Gaussian)   | 81.5 |   70.4   | 57.9 | 44.0 | **34.3** |   29.6   | **25.0** | **20.9** | **16.5** |
> |                       |      |          |      |      |          |          |          |          |          |
> | DRT + AE (SmoothAdv)  | 72.6 |   67.0   | 60.2 | 50.5 |   39.5   |   36.0   |   30.3   |   24.1   |   20.3   |
> | DRT + WE (SmoothAdv)  | 72.6 | **67.1** | 60.2 | 50.5 |   39.5   | **36.1** |   30.3   |   24.1   | **20.4** |
>
> Bold numbers means WE achieves higher certified accuracy than AE under the same radii.
>
> We can see that, by learning the optimal weights, the certified accuracy could be slightly improved compared to the equal-weighted setting in some cases and we have added related discussion in our revision.

---

### Official Review · Reviewer_mt7k · 2021-11-06

**Correctness:** 4
**Technical Novelty And Significance:** 3
**Empirical Novelty And Significance:** 3
**Recommendation:** 8
**Confidence:** 4

**Main Review:**

**Strengths of the paper**
- This is a rigorous and solid paper in which the practical approach is developed based on theoretical findings.
- The writing and presentation are generally good and motivating.
- Convincing experimental results.

**Weaknesses of the paper**
- The findings of the paper are not really new. The usefulness of gradient diversity in improving the robustness of an ensemble model and the capability of a large confident margin in enhancing the robustness are well-known. Furthermore, randomized smoothing has been proposed before. However, this paper is good in the sense of putting together them in the theory development of an ensemble approach.

**Summary Of The Paper:**

This paper proposes using an ensemble model to improve certified robustness. Based on the developed theory, it proposes Diversity-Regularized Training (DRT), a lightweight regularization-based ensemble training approach which composes of two simple yet effective and general regularizers to promote the diversified gradients and large confidence margins respectively. The experiments were conducted to show the merit of the proposed approach.

**Summary Of The Review:**

It would be great if the authors offers more discussion about the variation of the upper bounds in equations (4,5) if we vary the robust radius $r$. The current analysis is done with a fixed $r$.

---

> ### Author Response · Authors · 2021-11-20
> **Response to Reviewer mt7k**
>
> Thanks for your appreciation and thoughtful suggestions. We checked your concerns carefully and answered them as follows.
>
> > “It would be great if the authors offer more discussion about the variation of the upper bounds in equations (4,5) if we vary the robust radius r. The current analysis is done with a fixed $r$.”
>
> We thank the reviewer for the insightful comments. We have added the discussion in Section 2.2 (highlighted) following the suggestion as: We can directly analyze the robustness under different attack radius $r$ by varying $r$ in Theorem 1. When $r$ becomes larger, the gap between the RHS of two inequalities ($2\beta r \sum_{j=1}^N w_j$) also becomes larger, and thus it becomes relatively hard to tightly bound the robustness via Theorem 1. This is because the first-order condition implied by the theorem becomes coarse when $r$ is large.  However, due to the bounded $\beta$ after randomized smoothing (Theorem 2), the proposed training approach motivated by the theorem is able to ensure the robustness empirically even under large $r$.

---

> > ### Comment · Reviewer_mt7k · 2021-11-25
> > **About the analysis when varying r**
> >
> > Thanks for your feedback and adding this further clarification to the main paper.

---

### Author Response · Authors · 2021-11-20
**Revision Summary according to the Feedbacks from Reviewers**

We sincerely thank all the reviewers for the valuable comments and we have revised our paper accordingly. Specifically, we made the following updates and highlighted the changes in blue in our revision:

1. We added brief discussion about how changing robust radius $r$ would affect the upper bound on Page 4-5, following **Reviewer mt7k**’s suggestion.

2. We rewritten Section 2.3 thoroughly to strengthen the logic chain: Theorem 1 => EBS, following **Reviewer AZhe** and **Reviewer VR8o**’s suggestion.

3. We provided theoretical evidence and related discussion on how larger number of base models ($N$) could lead to better certified robustness of the ensemble in Proposition B.1 in Appendix B.1, following **Reviewer f6eQ**’s suggestion.

4. We clarified our same weights setting for WE protocol in Section 4.1 and added ablation study on how learning the optimal weights could further improve certified robustness in Appendix G.3, following **Reviewer ug5z** and **Reviewer f6eQ**’s suggestion.

5. We updated the performance of DRT-trained ensemble under Average Certified Radius (ACR) metric for both MNIST and CIFAR-10 datasets in Appendix F, following **Reviewer AZhe**’s suggestion.

6. We updated the certified accuracy of single base model within DRT-trained ensemble in Appendix G.2, following **Reviewer AZhe**’s suggestion.

7. We added discussion on the top of Page 9, linking to the analysis of computational cost of DRT, following **Reviewer AZhe**’s suggestion.

8. We fixed the typo $|\delta|_2\le r$ in Definition 1 on Page 3, following **Reviewer VR8o**’s suggestion.

Please also let us know if there are other questions, and we really look forward to the discussion with the reviewers to further improve our paper. Thank you!!

---

### Decision · Program_Chairs · 2022-01-20

**Decision:**

Accept (Poster)

**Comment:**

The paper proposes Diversity-Regularized Training (DRT), a new training method for an ensemble classifier to improve its certified robustness when randomized smoothing is applied. Specifically, it trains a set of base classifiers to diversify their input gradients while maximizing the confidence margin of each. The method is backed up with a theoretical observation on robustness of ensembles of smooth classifiers.

After the discussion phase, the reviewers unanimously ended up with supporting acceptance of this paper, and the authors were quite responsive to address the reviewers' concerns. Overall, the reviewers appreciated its strong empirical results with a theoretical support - AC also agrees on that, and thinks the paper presents a promising and under-explored direction to boost certified robustness of randomized smoothing.